# Neuronal temperature perception induces specific defenses that enable *C. elegans* to cope with the enhanced reactivity of hydrogen peroxide at high temperature

Francesco A Servello[1], Rute Fernandes[2,3], Matthias Eder[2,3], Nathan Harris[4], Olivier MF Martin[2,3], Natasha Oswal[2,3], Anders Lindberg[1], Nohelly Derosiers[1], Piali Sengupta[4], Nicholas Stroustrup[2,3], Javier Apfeld[1,5]*

[1]Biology Department, Northeastern University, Boston, United States; [2]Centre for Genomic Regulation (CRG), The Barcelona Institute of Science and Technology, Barcelona, Spain; [3]Universitat Pompeu Fabra (UPF), Barcelona, Spain; [4]Department of Biology, Brandeis University, Waltham, United States; [5]Bioengineering Department, Northeastern University, Boston, United States

**Abstract** Hydrogen peroxide is the most common reactive chemical that organisms face on the microbial battlefield. The rate with which hydrogen peroxide damages biomolecules required for life increases with temperature, yet little is known about how organisms cope with this temperature-dependent threat. Here, we show that *Caenorhabditis elegans* nematodes use temperature information perceived by sensory neurons to cope with the temperature-dependent threat of hydrogen peroxide produced by the pathogenic bacterium *Enterococcus faecium*. These nematodes preemptively induce the expression of specific hydrogen peroxide defenses in response to perception of high temperature by a pair of sensory neurons. These neurons communicate temperature information to target tissues expressing those defenses via an insulin/IGF1 hormone. This is the first example of a multicellular organism inducing their defenses to a chemical when they sense an inherent enhancer of the reactivity of that chemical.

*For correspondence:
j.apfeld@northeastern.edu

## Editor's evaluation

For any organism, tailoring defenses to the most pressing threats has high adaptive value – this paper makes the important finding that the nematode *C. elegans* pre-emptively augments its defenses against hydrogen peroxide when temperature increases, a condition that enhances the damage this compound causes. The authors describe this strategy as 'enhancer sensing,' whereby the perception of an environmental stimulus leads to the induction of defenses against a distinct (but mechanistically linked) threat. Convincing mechanistic studies reveal a role for a key thermosensory neuron and insulin-like signaling in this phenomenon. Because of its interdisciplinary outlook, this work will be of interest to readers in the fields of sensory neuroscience, stress physiology, and evolutionary biology.

## Introduction

Reactive chemicals in the environment pose a lethal threat to organisms by changing the chemical composition of their macromolecules. But organisms are not passive chemical substrates, they have sophisticated defense mechanisms that deal with the threat posed by those chemicals. That threat is

**eLife digest** The Earth's environment is full of reactive chemicals that can cause harm to organisms. One of the most common is hydrogen peroxide, which is produced by several bacteria in concentrations high enough to kill small animals, such as the roundworm *Caenorhabditis elegans*. Forced to live in close proximity to such perils, *C. elegans* have evolved defenses to ensure their survival, such as producing enzymes that can break down hydrogen peroxide.

However, this battle is compounded by other factors. For instance, rising temperatures can increase the rate at which the hydrogen peroxide produced by bacteria reacts with the molecules and proteins of *C. elegans*. In 2020, a group of researchers found that roundworms sense these temperature changes through special cells called sensory neurons and use this information to control the generation of enzymes that break down hydrogen peroxide. This suggests that *C. elegans* may pre-emptively prepare their defenses against hydrogen peroxide in response to higher temperatures so they are better equipped to shield themselves from this harmful chemical.

To test this theory, Servello et al. – including some of the authors involved in the 2020 study – exposed *C. elegans* to a species of bacteria that produces hydrogen peroxide. This revealed that the roundworms were better at dealing with the threat of hydrogen peroxide when growing in warmer temperatures. Experiments done in *C. elegans* lacking a class of sensory cells, the AFD neurons, showed that these neurons increased the roundworms' resistance to the chemical when temperatures increase. They do this by repressing the activity of INS-39, a hormone that stops *C. elegans* from switching on their defense mechanism against peroxides.

This is the first example of a multicellular organism preparing its defenses to a chemical after sensing something (such as temperature) that enhances its reactivity. It is possible that other animals may also use this 'enhancer sensing' strategy to anticipate and shield themselves from hydrogen peroxide and potentially other external threats.

inherently temperature dependent because chemical reactions occur at faster rates at higher temperatures (*Arrhenius, 1889*; *Evans and Polanyi, 1935*; *Eyring, 1935*). However, the extent to which the defense mechanisms protecting the organism from reactive chemicals are adjusted to balance the temperature-dependent threat posed by those chemicals remains poorly understood. In the present study, we used the nematode *Caenorhabditis elegans* as a model system to explore the extent to which temperature regulates how multicellular organisms deal with the threat of hydrogen peroxide.

Hydrogen peroxide ($H_2O_2$) is the most common reactive chemical that organisms face on the microbial battlefield (*Mishra and Imlay, 2012*). Bacteria, fungi, plants, and animal cells have long been known to use $H_2O_2$ as an offensive weapon that damages the nucleic acids, proteins, and lipids of their targets (*Avery and Morgan, 1924*; *Imlay, 2018*). *C. elegans* encounter a wide variety of bacteria in their ecological setting (*Samuel et al., 2016*; *Schiffer et al., 2021*), including many genera known to produce $H_2O_2$ (*Passardi et al., 2007*). $H_2O_2$ produced by a bacterium from the *C. elegans* microbiome, *Neorhizobium sp.*, causes DNA damage to the nematodes (*Kniazeva and Ruvkun, 2019*) and many bacteria—including *Enterococcus faecium, Streptococcus pyogenes, Streptococcus pneumoniae*, and *Streptococcus oralis*—kill *C. elegans* by producing millimolar concentrations of hydrogen peroxide (*Bolm et al., 2004*; *Jansen et al., 2002*; *Moy et al., 2004*).

Prevention and repair of the damage that hydrogen peroxide inflicts on macromolecules are critical for cellular health and survival (*Chance et al., 1979*). To avoid damage from $H_2O_2$, *C. elegans* rely on conserved cellular defenses, including $H_2O_2$-degrading catalases (*Chávez et al., 2007*; *Schiffer et al., 2020*). However, inducing those defenses at inappropriate times can cause undesirable side effects, including developmental defects (*Doonan et al., 2008*; *Kramer-Drauberg et al., 2020*), because $H_2O_2$ modulates the function of proteins involved in a wide variety of cellular processes, including signal transduction and differentiation (*Hourihan et al., 2016*; *Kramer-Drauberg et al., 2020*; *Meng et al., 2021*; *Veal et al., 2007*). We recently found that 10 classes of sensory neurons in the brain of *C. elegans* manage the challenge of deciding when the nematode's tissues induce $H_2O_2$ defenses (*Schiffer et al., 2020*). Sensory neurons might be able to integrate a wider variety of inputs than the individual tissues expressing those defenses could integrate, enabling a better assessment of the threat of hydrogen peroxide.

In their habitat, *C. elegans* face daily and seasonal variations in temperature, which can affect a wide variety of processes, including development, reproduction, and lifespan (*Golden and Riddle, 1984*; *Klass, 1977*). Temperature also affects the growth of bacteria (*Barber, 1908*; *Rosso et al., 1993*) and, therefore, likely affects the interactions of *C. elegans* with beneficial and pathogenic bacteria (*Samuel et al., 2016*; *Zhang et al., 2017*). While nematodes do not regulate their own body temperature, they adjust their behavior and physiology in response to the perception of temperature by sensory neurons, enabling them to seek temperatures conducive to survival, avoid noxious temperature ranges, and induce heat defenses (*Goodman and Sengupta, 2019*; *Hedge-cock and Russell, 1975*; *Prahlad et al., 2008*). Nematodes perceive temperature, in part, via seven classes of sensory neurons (*Beverly et al., 2011*; *Biron et al., 2008*; *Chatzigeorgiou et al., 2010*; *Kuhara et al., 2008*; *Liu et al., 2012*; *Mori and Ohshima, 1995*; *Schild et al., 2014*). Previously, we found that four of those classes of neurons regulate *C. elegans* peroxide resistance (*Schiffer et al., 2020*), suggesting that nematodes might adjust their peroxide defenses in response to temperature perception.

Here, we show that the lethality of hydrogen peroxide to *C. elegans* increases with temperature. Nematodes partially compensate for this by preemptively inducing their hydrogen peroxide defenses at high temperature. This adaptive response to temperature enables the nematodes to better cope with $H_2O_2$ produced by the pathogenic bacterium *E. faecium*. The temperature-dependent regulation of peroxide defenses is directed by the AFD sensory neurons. At high temperature, the AFD neurons repress the expression of the INS-39 insulin/IGF1 hormone and thereby alleviate inhibition by insulin/IGF1 signaling of the nematodes' peroxide defenses. The insulin/IGF1 effector DAF-16/FOXO functions in intestinal cells to determine the size of the gene-expression changes induced by the absence of signals from the AFD neurons. By coupling the induction of $H_2O_2$ defenses to the perception of high temperature—an inherent enhancer of the reactivity of $H_2O_2$—the nematodes are assessing faithfully the threat that $H_2O_2$ poses.

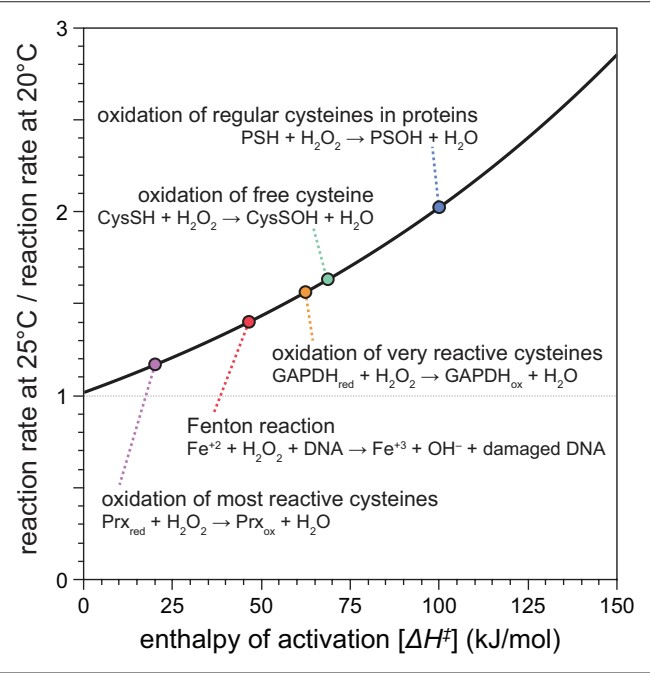

**Figure 1.** Temperature-dependent enhancement of $H_2O_2$ reactivity toward biologically important molecules. Using chemical kinetics, we modeled how much the rates of chemical reactions of $H_2O_2$ change between 20 and 25°C. This plot illustrates how the enthalpy of activation of a specific chemical reaction influences its rate at 25°C compared to 20°C. The dotted arrows point to the relative reaction rate values for specific reactions of $H_2O_2$ with biologically important molecules.

The online version of this article includes the following source data for figure 1:

**Source data 1.** Kinetic modeling data.

## Results

### *C. elegans* induces long-lasting peroxide defenses in response to high temperature

In their natural habitat, *C. elegans* nematodes encounter many threats that can shorten their lifespan. A major chemical threat that *C. elegans* face is hydrogen peroxide ($H_2O_2$). Bacteria can produce millimolar concentrations of $H_2O_2$ (*Bolm et al., 2004*; *Jansen et al., 2002*; *Moy et al., 2004*), shortening *C. elegans* lifespan over tenfold (*Chávez et al., 2007*). A major physical threat that *C. elegans* face is high temperature. Even a small increase in temperature from 20 to 25°C—within the range of ambient temperatures that *C. elegans* prefers in nature (*Crombie et al., 2019*)—shortens *C. elegans* lifespan from 19 to 15 days (*Klass, 1977*; *Lee and Kenyon, 2009*; *Stroustrup et al., 2016*). We set out to investigate the extent to which $H_2O_2$ and temperature acted together to determine *C. elegans* survival.

Previous studies showed that $H_2O_2$ kills *C. elegans* in a dose-dependent manner at environmental concentrations above 0.1 mM (*Bolm et al., 2004*; *Jansen et al., 2002*; *Moy et al., 2004*). We expected that higher temperatures would make the same concentration of $H_2O_2$ more lethal to *C. elegans*, because the reaction rates of the chemical reactions of $H_2O_2$ increase exponentially with temperature (*Arrhenius, 1889*; *Evans and Polanyi, 1935*; *Eyring, 1935*). The exact molecular mechanisms by which $H_2O_2$ kills *C. elegans*, or any organism, remain unknown but are thought to involve the reactions of $H_2O_2$ with biologically important molecules, including proteins and DNA (*Khademian and Imlay, 2021*). Using chemical kinetics, we modeled how an increase in temperature from 20 to 25°C would affect the rates of the chemical reactions of $H_2O_2$ with those biomolecules (*Figure 1*). Because these rate differences depend on the enthalpy of activation of the specific chemical reaction, they can vary widely between reactions. The Fenton reaction of $H_2O_2$ with DNA-bound Fe(II), which leads to DNA damage, was predicted to be 40% faster at 25°C than at 20°C (*Figure 1*). For the oxidation of the thiol groups of cysteines, reaction rates with $H_2O_2$ were predicted to be more than twofold faster for regular cysteines in proteins, 62% faster for free cysteines, up to 56% faster for very reactive cysteines such as the redox-sensitive cysteine residue of GAPDH, and 17% faster for the most reactive cysteines of hydroperoxidases (*Figure 1*). These predicted increases in $H_2O_2$'s reactivity toward specific biomolecules at 25°C, compared to 20°C, are similar to the ones that would occur at 20°C if $H_2O_2$ concentration were increased substantially—from 17% to more than 100%, depending on the specific reaction.

To investigate the extent to which cultivation temperature might influence *C. elegans* survival in the presence of environmental peroxides, we measured the peroxide resistance of nematodes cultured within their preferred temperature range (*Crombie et al., 2019*). We cultured the nematodes at either 20 or 25°C until the second day of adulthood, and then determined their subsequent survival at those temperatures in the presence of a peroxide in their environment. We used tert-butyl hydroperoxide (tBuOOH) because this peroxide, unlike $H_2O_2$, is not degraded efficiently by *Escherichia coli*—the nematodes' conventional food in the laboratory (*Brenner, 1974*). Previously, we found that when tBuOOH concentration exceeded 0.75 mM, *C. elegans* lifespan was shortened by 50% by each additional 45% increase in tBuOOH concentration (*Stroustrup et al., 2016*). In the presence of 6 mM tBuOOH, nematodes grown at 20°C survived an average of 1.6 days at 20°C, while those grown and assayed at 25°C survived 30% shorter (*Figure 2A*). Therefore, *C. elegans* peroxide resistance was temperature dependent.

We speculated that *C. elegans* survival to bacterially produced $H_2O_2$ would, likewise, be shorter at 25°C than at 20°C. $H_2O_2$ produced by the pathogenic bacterium *E. faecium* is lethal to *C. elegans* (*Chávez et al., 2007*; *Moy et al., 2004*). We exposed day 2 adult nematodes that fed on *E. coli* JI377—a *katG katE ahpCF* triple null mutant strain which cannot degrade environmental $H_2O_2$ (*Seaver and Imlay, 2001*)—to the supernatant of an *E. faecium* liquid culture and, after 16 hr, determined the proportion of nematodes that survived. Compared to nematodes grown and assayed at 20°C, those grown and assayed at 25°C were less likely to survive the *E. faecium* supernatant (*Figure 2B*), indicating that $H_2O_2$ was more lethal to *C. elegans* at the higher temperature. Together, these observations indicated that at the upper end of the natural temperature range of *C. elegans*, two types of peroxides were more lethal to the nematodes.

We expected that increasing temperature would make peroxides more lethal to *C. elegans* because temperature increases the rate of chemical reactions, including those that mediate peroxide-dependent killing. If this was the only mechanism by which temperature affected *C. elegans*' peroxide

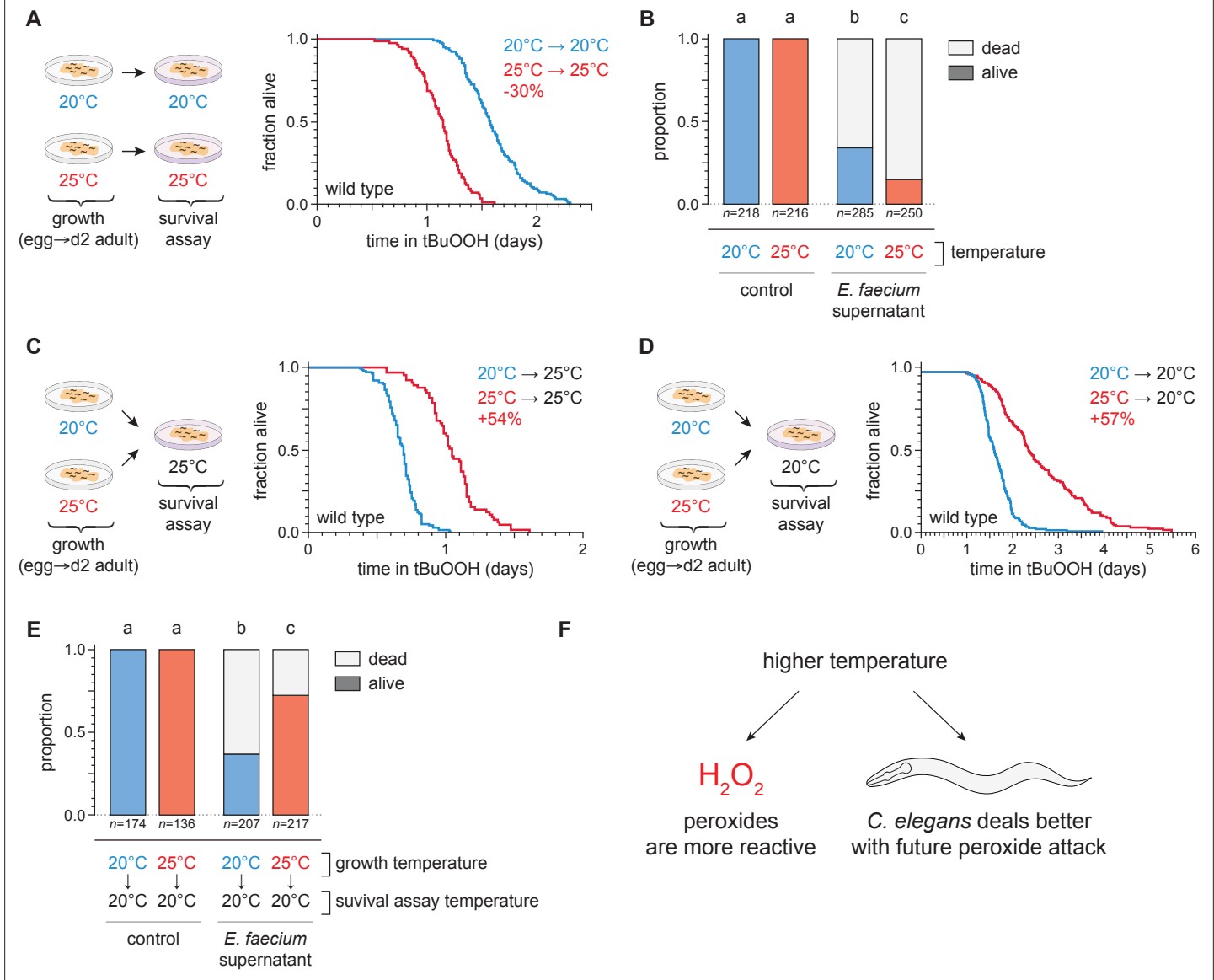

**Figure 2.** Temperature regulates the peroxide resistance of *C. elegans*. (**A**) Peroxide resistance of wild-type *C. elegans* grown and assayed at 20 or 25°C. The fraction of nematodes remaining alive in the presence of 6 mM tert-butyl hydroperoxide (tBuOOH) is plotted against time. (**B**) Survival of wild-type *C. elegans* 16 hr after exposure to *E. faecium* E007 liquid-culture supernatant. Nematodes were grown and assayed at 20 or 25°C. Groups labeled with different letters exhibited significant survival differences (p < 0.001, ordinal logistic regression) otherwise (p > 0.05). (**C**) Peroxide resistance at 25°C of wild-type *C. elegans* grown at 20 or 25°C. (**D**) Peroxide resistance at 20°C of wild-type *C. elegans* grown at 20 or 25°C. (**E**) Survival of wild-type *C. elegans* 16 hr after exposure to *E. faecium* E007 liquid-culture supernatant. Nematodes were grown at 20 or 25°C and assayed at 20°C. Groups labeled with different letters exhibited significant survival differences (p < 0.001, ordinal logistic regression) otherwise (p > 0.05). (**F**) Peroxides killed *C. elegans* more quickly at 25°C than at 20°C, but nematodes grown at 25°C could survive a subsequent peroxide exposure better than those grown at 20°C. Statistical analyses for panels (A, C, and D) are in *Supplementary file 1*.

The online version of this article includes the following source data and figure supplement(s) for figure 2:

**Source data 1.** Survival data for panels A, C, D.

**Source data 2.** Survival data for panels B, E.

**Figure supplement 1.** *C.elegans* can induce lasting peroxide defenses in response to high temperature.

**Figure supplement 1—source data 1.** Survival data for panels A, B.

resistance, then peroxide resistance should have been determined by the temperature the nematodes experienced during the peroxide resistance assay and not by the temperature they experienced before they were exposed to peroxide. Alternatively, the temperature *C. elegans* experienced before encountering peroxides in the environment may have influenced their subsequent sensitivity to peroxide. For example, a high cultivation temperature may have irreversibly damaged the nematodes, thus rendering them more sensitive to peroxide-dependent killing.

To distinguish between these possibilities, we measured the effects of the nematodes' growth-temperature history (before peroxide exposure) on their subsequent peroxide resistance by performing temperature-shift experiments where nematode populations grown at 20 or 25°C were transferred to assay plates containing 6 mM tBuOOH at either 20 or 25°C. To our surprise, we found that in survival assays performed at 25°C the nematodes grown at 25°C lived 54% longer than those grown at 20°C (*Figure 2C*). Similarly, in assays performed at 20°C, the nematodes grown at 25°C lived 57% longer than those grown at 20°C (*Figure 2D*). We also found that, compared with nematodes grown at 20°C, a higher proportion of nematodes grown at 25°C survived exposure to *E. faecium* liquid-culture supernatant at 20°C (*Figure 2E*). Therefore, nematodes grown at 25°C were more peroxide resistant than those grown at 20°C.

Our findings contradicted a model where temperature affected how quickly the nematodes were killed by peroxides only by influencing the reactivity of peroxides. In addition, those findings contradicted a prediction that high temperature would irreversibly render the nematodes more sensitive to peroxide-dependent killing. Instead, we conclude that even though peroxides killed *C. elegans* more quickly at 25°C than at 20°C, nematodes grown at 25°C could better survive a subsequent peroxide exposure than those grown at 20°C. Based on these findings, we speculated that *C. elegans* nematodes induced their peroxide defenses when grown at the higher temperature to prepare for the increased lethal threat posed by peroxides at high temperature (*Figure 2F*).

To determine the extent to which these differences in the nematodes' growth temperature had lasting effects on their subsequent peroxide resistance, we repeated the temperature-shift experiments, but this time we transferred the nematodes to the higher or lower temperature 1 day before the peroxide survival assay (on day 1 of adulthood), and 2 days before (at the onset of adulthood). Shifting from 20 to 25°C for 2 days was sufficient to improve peroxide survival at 25°C, but shifting only 1 day before the assay was not sufficient (*Figure 2—figure supplement 1A*). Therefore, nematodes grown at 20°C could increase their peroxide resistance in response to a temperature increase during adulthood. Nematodes down-shifted from 25 to 20°C for 2 days, 1 day, or immediately before the assay were all more peroxide resistant at 20°C than those grown continuously at 20°C (*Figure 2—figure supplement 1B*). Therefore, growth at 25°C could increase the nematodes' peroxide resistance even days after they had been transferred to 20°C. Together, these observations suggested that *C. elegans* can slowly induce long-lasting peroxide defenses in response to the higher cultivation temperature.

## AFD sensory neurons are required for the temperature dependence of *C. elegans* peroxide resistance

We recently found that sensory neurons regulate *C. elegans* sensitivity to peroxides in the environment (*Schiffer et al., 2020*). To investigate whether temperature might regulate *C. elegans* peroxide defenses via sensory neurons, we determined whether mutations that cause defects in the transduction of sensory information within neurons affected the extent to which temperature influenced the nematodes' peroxide resistance. We examined *tax-4* cyclic GMP-gated channel mutants, which are defective in the transduction of several sensory stimuli, including temperature (*Coburn and Bargmann, 1996*; *Komatsu et al., 1996*). When grown at 20°C, *tax-4* mutants exhibited an over twofold increase in peroxide resistance at 20°C relative to wild-type controls (*Figure 3A, B*). In contrast, when grown and assayed at 25°C, *tax-4* mutants exhibited a smaller increase in peroxide resistance, 49% (*Figure 3C*). These findings suggested that neuronal sensory transduction by TAX-4 channels normally lowers *C. elegans*' peroxide resistance to a lesser extent at high cultivation temperature.

To identify specific neurons that regulate *C. elegans* peroxide defenses in response to temperature, we focused on a single pair of neurons, the AFD neurons, chosen from the small subset of sensory neurons in which TAX-4 channels are expressed (*Coburn and Bargmann, 1996*; *Komatsu et al., 1996*). Previously, we found that genetic ablation of the AFD neurons via neuron-specific expression

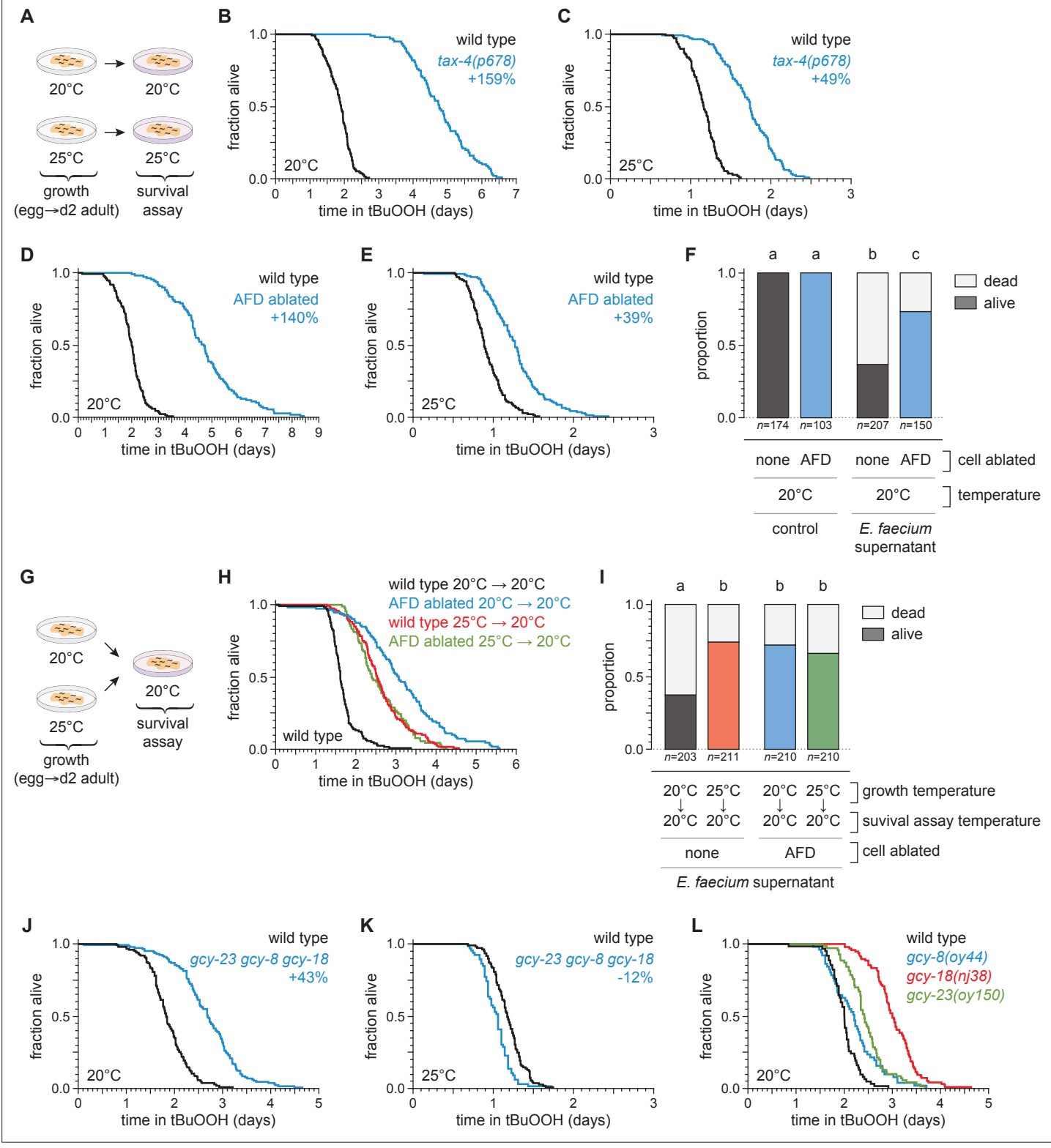

**Figure 3.** The AFD sensory neurons are required for the temperature dependence of *C. elegans* peroxide resistance. (**A**) Diagram summarizing the experimental strategy for panels (**B–F and J–L**). The *tax-4(p678)* mutation increased peroxide resistance by a greater factor at 20°C (**B**) than at 25°C (**C**). Genetic ablation of AFD increased peroxide resistance by a greater factor at 20°C (**D**) than at 25°C (**E**). (**F**) Genetic ablation of AFD increased the proportion of nematodes that survived 16 hr after exposure to *E. faecium* E007 liquid-culture supernatant. Nematodes were grown and assayed at 20°C. Groups labeled with different letters exhibited significant differences (p < 0.001, ordinal logistic regression) otherwise (p > 0.05). (**G**) Diagram

*Figure 3 continued on next page*

*Figure 3 continued*

summarizing the experimental strategy for panels (**H, I**). (**H**) AFD-ablated nematodes grown at 25°C did not exhibit a further increase in peroxide resistance at 20°C, unlike wild-type (unablated) nematodes. (**I**) Growth at 25°C did not further increase the proportion of AFD-ablated nematodes that survived after 16-hr exposure to *E. faecium* E007 liquid-culture supernatant at 20°C, unlike in wild-type (unablated) nematodes. Groups labeled with different letters exhibited significant differences (p < 0.001, ordinal logistic regression) otherwise (p > 0.05). (**J–K**) *gcy-23(oy150) gcy-8(oy44) gcy-18(nj38)* triple null mutants show increased peroxide resistance at 20°C (**J**) and decreased peroxide resistance at 25°C (**J**). (**L**) Peroxide resistance of *gcy-8(oy44)*, *gcy-18(nj38)*, and *gcy-23(oy150)* single mutants and wild-type nematodes at 20°C. Statistical analyses for panels (B–E, H, J–L) are in ***Supplementary file 2***.

The online version of this article includes the following source data and figure supplement(s) for figure 3:

**Source data 1.** Survival data for panels B–E, H, J–L.

**Source data 2.** Survival data for panels F, I.

**Figure supplement 1.** The AFD sensory neurons regulate *C. elegans* H$_2$O$_2$ resistance.

**Figure supplement 1—source data 1.** Survival data.

of caspase (***Chelur and Chalfie, 2007***; ***Glauser et al., 2011***) increased *C. elegans* peroxide resistance (***Schiffer et al., 2020***). Because the AFD neurons respond to temperature via TAX-4 channels to regulate diverse temperature-dependent behaviors (***Hedgecock and Russell, 1975***; ***Mori and Ohshima, 1995***), we speculated that they might also regulate peroxide resistance in response to temperature. To determine whether AFD neurons lowered the nematodes' peroxide resistance in a temperature-dependent manner, we measured the effects of AFD ablation at 20 and 25°C. Compared with wild-type nematodes, when grown and assayed at 20°C, the AFD-ablated nematodes exhibited an over twofold increase in resistance to tBuOOH (***Figure 3D***) and H$_2$O$_2$ (***Figure 3—figure supplement 1***), and were more likely to survive exposure to *E. faecium* liquid-culture supernatant (***Figure 3F***). At 25°C, AFD ablation increased resistance to tBuOOH by 31% (***Figure 3E***), a much smaller amount than at 20°C. Therefore, the AFD neurons normally lower *C. elegans* peroxide resistance in a temperature-dependent manner.

If the AFD neurons were blocking the induction of peroxide defenses, we hypothesized that ablation of both AFD neurons might result in induction of peroxide defenses at lower temperatures similar to those seen in unablated nematodes at the higher temperature. Therefore, we predicted that AFD-ablated nematodes grown at 25°C would not exhibit a further increase in peroxide resistance at 20°C, unlike wild-type nematodes. Consistent with that prediction, AFD-ablated nematodes grown at 25°C exhibited the same levels of resistance as wild-type nematodes grown at 25°C (***Figure 3G, H***). AFD-ablated nematodes grown continuously at 20°C exhibited the highest levels of peroxide resistance (***Figure 3H***). Similarly, in assays at 20°C measuring nematode survival after exposure to a supernatant derived from a liquid culture of *E. faecium*, AFD-ablated nematodes grown at either 20 or 25°C survived as well as wild-type nematodes grown at 25°C (***Figure 3I***). We propose that, in wild-type *C. elegans*, the extent to which the AFD neurons lower peroxide defenses is reduced in response to higher temperature.

Last, we determined whether previously identified mechanisms for temperature perception by the AFD neurons were required for the temperature-dependent regulation of peroxide resistance. The AFD neurons sense temperature using receptor guanylate cyclases, which catalyze cGMP production, leading to the opening of TAX-4 channels (***Goodman and Sengupta, 2019***). Three receptor guanylate cyclases are expressed exclusively in AFD neurons: GCY-8, GCY-18, and GCY-23 (***Inada et al., 2006***; ***Yu et al., 1997***) and are thought to act as temperature sensors (***Takeishi et al., 2016***). Triple mutants lacking *gcy-8, gcy-18*, and *gcy-23* function are behaviorally atactic on thermal gradients and fail to display changes in intracellular calcium or thermoreceptor current in the AFD neurons in response to temperature changes (***Inada et al., 2006***; ***Ramot et al., 2008***; ***Takeishi et al., 2016***; ***Wang et al., 2013***; ***Wasserman et al., 2011***). We found that when grown and assayed at 20°C, *gcy-23(oy150) gcy-8(oy44) gcy-18(nj38)* triple null mutants survived 43% longer in the presence of tBuOOH than wild-type controls (***Figure 3J***). In contrast, at 25°C, the *gcy-23 gcy-8 gcy-18* triple mutants showed a 12% decrease in peroxide resistance relative to wild-type controls (***Figure 3K***). Therefore, the three AFD-specific receptor guanylate cyclases influenced the temperature dependence of peroxide resistance, lowering peroxide resistance at 20°C and slightly increasing it at 25°C. At 20°C, the *gcy-8(oy44)*, *gcy-18(nj38)*, and *gcy-23(oy150)* single mutants increased peroxide resistance by 10%, 51%, and

21%, respectively, relative to wild-type controls (*Figure 3L*). Therefore, each of the three AFD-specific receptor guanylate cyclases regulates peroxide resistance, and their roles are not fully redundant. We conclude that temperature perception by AFD via GCY-8, GCY-18, and GCY-23 enables *C. elegans* to lower their peroxide resistance at the lower cultivation temperature. Other mechanisms within AFD likely contribute to the regulation of peroxide resistance, as AFD ablation caused a greater increase in peroxide resistance than the *gcy-23 gcy-8 gcy-18* triple mutant.

## Hydrogen peroxide defenses are induced by high cultivation temperature and by AFD ablation

To investigate whether the higher cultivation temperature and the ablation of the AFD sensory neurons increased *C. elegans* peroxide resistance through a common defense mechanism, we used mRNA sequencing (mRNA-seq) to compare the extent to which those interventions affected gene expression. Collecting mRNA from day 2 adults grown at 20 and 25°C and from AFD-ablated and unablated (wild-type) nematodes grown at 20°C, we identified differentially expressed transcripts. Relative to nematodes grown at 20°C, those grown at 25°C had lower expression of 2446 genes and higher expression of 809 genes, out of 18039 genes detected ($q$ value <0.01) (*Figure 4A* and *Figure 4—figure supplement 1A*). These changes in gene expression were consistent with previous studies comparing gene expression in nematodes grown at 20 and 25°C (*Gómez-Orte et al., 2018*; *Figure 4—figure supplement 2A, B*) and in nematodes shifted from 23 to 17°C (*Sugi et al., 2011*; *Figure 4—figure supplement 2D, E*). AFD ablation lowered the expression of 2077 genes and increased the expression of 2225 genes, out of 7912 genes detected ($q$ value <0.01) (*Figure 4B* and *Figure 4—figure supplement 1B*). Therefore, both the higher cultivation temperature and the ablation of the AFD sensory neurons induced broad changes in gene expression.

Next, we asked whether higher cultivation temperature and ablation of the AFD neurons altered gene expression for each transcript by the same amount and in the same direction. We found that both interventions induced changes in gene expression that were linearly correlated in a positive manner ($R^2$ = 9%, p < 0.0001, *Figure 4C*). We then asked whether this weak correlation was due to co-induction of upregulated genes, co-repression of downregulated genes, or both, using categorical analysis. We found that genes with either higher or lower expression in both wild-type nematodes at 25°C and AFD-ablated nematodes at 20°C were disproportionally enriched, and that almost no genes were upregulated at 25°C but downregulated in AFD-ablated nematodes at 20°C (*Figure 4D*). Therefore, we conclude that cultivation temperature and AFD ablation induced overlapping changes in gene expression.

We then determined whether genes previously shown to be regulated between various temperature ranges were co-regulated by growth at 25°C and by ablation of AFD at 20°C. Genes expressed at a higher level at 25°C than at 20°C (*Gómez-Orte et al., 2018*) were upregulated by ablation of AFD at 20°C and were also, as expected, upregulated by growth at 25°C (*Figure 4—figure supplement 2A*); however, genes expressed at a higher level at 15°C than at 20°C (*Gómez-Orte et al., 2018*) were downregulated by growth at 25°C but were upregulated by ablation of AFD at 20°C (*Figure 4—figure supplement 2B*). In addition, genes induced more than twofold when nematodes at 25°C were heat shocked by shifting them to 30°C (*McCarroll et al., 2004*) were upregulated by ablation of AFD at 20°C, but were unaffected by growth at 25°C (*Figure 4—figure supplement 2C*). We conclude that, in nematodes cultivated at 20°C, the AFD sensory neurons not only repressed genes induced at a higher cultivation temperature (25°C), but also repressed genes induced at a lower cultivation temperature (15°C) and in response to heat shock (30°C).

To identify processes that may be influenced by the transcriptomic changes induced by the higher cultivation temperature and by the ablation of the AFD neurons, we used Gene Ontology (GO) term enrichment analysis (*Angeles-Albores et al., 2016*; *Ashburner et al., 2000*) and clustered enriched GO terms based on semantic similarity (*Supek et al., 2011*), focusing on genes with more than a twofold increase or decrease in expression between wild-type nematodes at 25 and 20°C and between AFD-ablated and unablated nematodes at 20°C. We found that both higher emperature and AFD ablation downregulated genes associated with reproduction and with expression in the germline, and upregulated genes associated with defense and immune responses and with expression in the intestine (*Figure 4E, F* and *Supplementary file 4*). To expand this analysis, we determined the extent to which higher cultivation temperature and ablation of AFD co-regulated the expression of

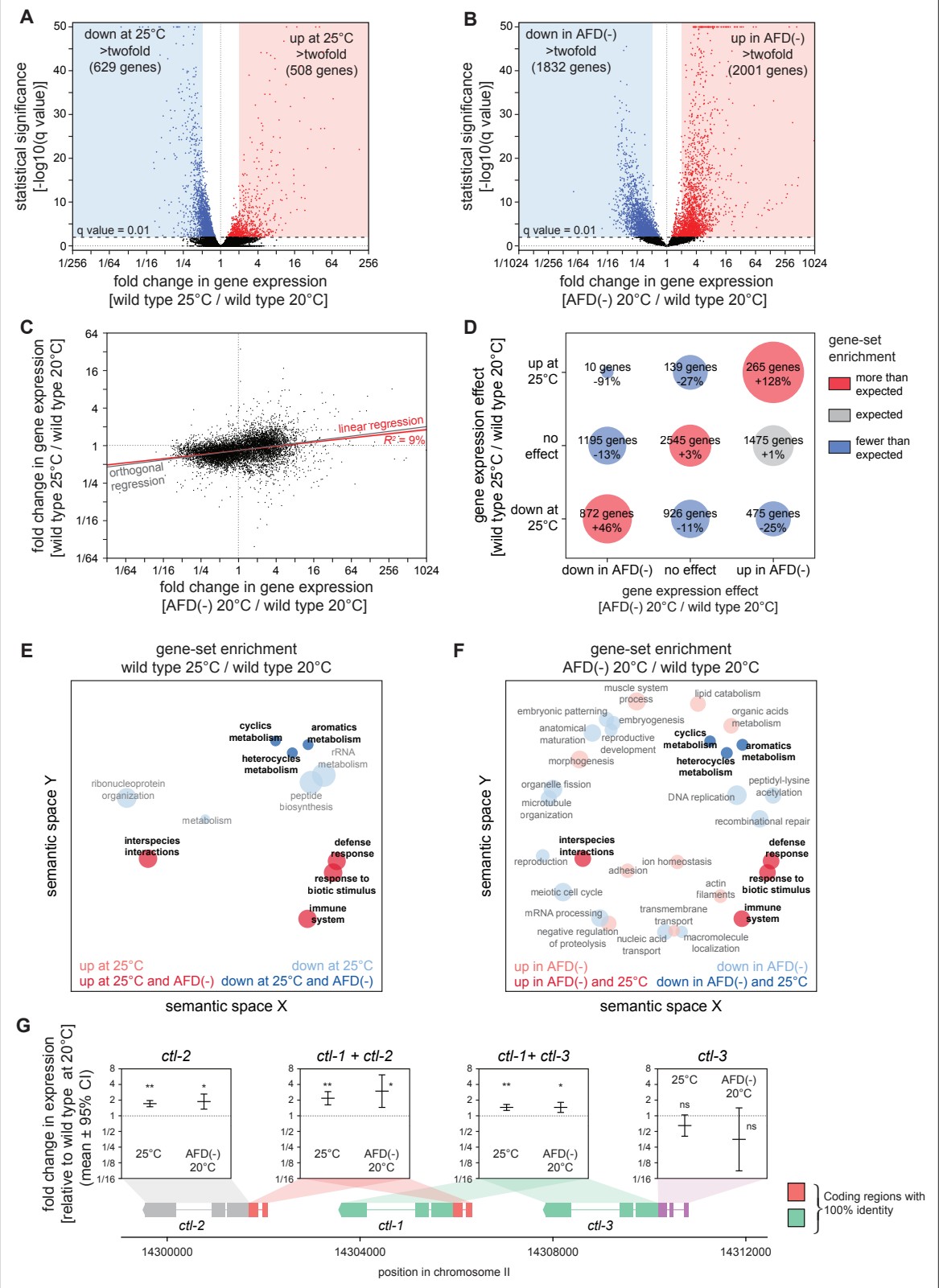

**Figure 4.** Hydrogen peroxide defenses are induced by high cultivation temperature and AFD ablation. Volcano plots showing the level and statistical significance of changes in gene expression induced (**A**) in wild-type nematodes by growth at 25°C relative to growth at 20°C and (**B**) by AFD ablation in nematodes grown at 20°C relative to wild-type (unablated) nematodes grown at 20°C. Genes up- and downregulated significantly (*q* value <0.01) are shown in red and blue, respectively. (**C**) Growth at 25°C and AFD ablation at 20°C induced correlated changes in gene expression. Linear regression fit is

*Figure 4 continued on next page*

*Figure 4 continued*

shown as a red line flanked by a red area marking the 95% confidence interval of the fit. The orthogonal regression fit (gray line) makes no assumptions about the dependence or independence of the variables. (**D**) Coregulation of genes up- and downregulated significantly (*q* value <0.01) by growth at 25°C and by AFD ablation at 20°C. Bubble size is proportional to gene-set enrichment (observed/expected). Gene sets with significantly more or fewer genes than expected (p < 0.001, cell chi-square test) are colored red and blue, respectively; gene sets of the expected size (p > 0.05) are colored gray. (**E, F**) Gene Ontology (GO) term enrichment analysis. (**E**) Biological processes associated with the set of 508 upregulated genes (red bubbles) and the set of 629 downregulated genes (blue bubbles) with a statistically significant and greater than twofold change in expression in wild-type nematodes grown at 25°C relative to those grown at 20°C. (**F**) Biological processes associated with the set of 2001 upregulated genes (red bubbles) and the set of 1832 downregulated genes (blue bubbles) with a statistically significant and greater than twofold change in expression in AFD-ablated nematodes grown at 20°C relative to wild-type (unablated) nematodes grown at 20°C. Bubble size is proportional to the statistical significance [−log$_{10}$(p value)] of enrichment. Biological processes that were induced or repressed by both interventions are bolded and shaded with darker red and blue colors, respectively. (**G**) Average changes in expression and 95% confidence intervals induced by growth at 25°C and AFD ablation at 20°C within intervals in the genomic region encoding the three *C. elegans* catalase genes. Gene models show the positions and splicing pattern of each catalase gene, intervals with 100% nucleotide identity are shown in orange (*ctl-1* and *ctl-2*) and green (*ctl-1* and *ctl-3*), and unique intervals are show in gray (*ctl-2*) and purple (*ctl-3*). The asterisks mark gene regions with significant fold-change in expression: **p < 0.001 and *p < 0.025, otherwise 'ns' indicates p > 0.1 (generalized linear model).

The online version of this article includes the following source data and figure supplement(s) for figure 4:

**Source data 1.** mRNA sequencing (mRNA-seq) analysis data.

**Source data 2.** mRNA sequencing (mRNA-seq) analysis data for the genomic region of the three catalase genes.

**Figure supplement 1.** Principal component analysis (PCA) of the sequenced samples.

**Figure supplement 2.** The AFD sensory neurons influence responses to noxious heat, high cultivation temperature, and low temperature.

**Figure supplement 3.** High cultivation temperature and AFD ablation induce positively correlated changes in the expression of gene sets affecting similar biological processes.

**Figure supplement 3—source data 1.** WormCat analysis data.

**Figure supplement 4.** High cultivation temperature and AFD ablation pre-induce genes induced by tert-butyl hydroperoxide.

**Figure supplement 5.** AFD ablation, but not high cultivation temperature, pre-induces genes induced by toxic organic compounds, toxic metals, and radiation.

gene sets affecting similar biological processes. We assigned each gene to a set of nested categories based on their physiological function and then their molecular function or cellular location using WormCat annotations (*Higgins et al., 2022*; *Holdorf et al., 2020*). Higher temperature and AFD ablation induced positively correlated changes in the average expression of those gene sets ($R^2$ = 24%, p < 0.0001, *Figure 4—figure supplement 3*). Therefore, the higher cultivation temperature and ablation of the AFD sensory neurons appeared to induce consistent changes in the expression of genes affecting similar biological processes.

We next determined whether genes induced when nematodes were exposed to tert-butyl hydroperoxide (*Oliveira et al., 2009*) were also induced by the higher cultivation temperature and by the ablation of the AFD neurons. Both growth at 25°C and ablation of AFD at 20°C increased the expression of those genes (*Figure 4—figure supplement 4*). Therefore, in the absence of peroxide exposure, genes induced by peroxides were pre-induced in both nematodes cultivated at the higher temperature and in AFD-ablated nematodes, suggesting those nematodes were better prepared to deal with peroxides in the environment.

To identify specific peroxide defenses induced by the higher cultivation temperature and by the ablation of the AFD neurons, we focused on the catalase genes, which encode enzymes that degrade hydrogen peroxide (*Loew, 1901*; *Nicholls, 2012*; *Togo et al., 2000*). The *C. elegans* genome contains three catalase genes in tandem—two newly duplicated cytosolic catalases, *ctl-1* and *ctl-3*, and a peroxisomal catalase, *ctl-2* (*Petriv and Rachubinski, 2004*)—that when overexpressed 10-fold increase *C. elegans* resistance to hydrogen peroxide 2.7-fold (*Schiffer et al., 2020*). *ctl-1* and *ctl-2* can increase *C. elegans* resistance to H$_2$O$_2$-dependent killing (*Chávez et al., 2007*; *Schiffer et al., 2020*). Previously, we found that *ctl-1* mRNA levels were 69% higher in *daf-1* Type 1 TGFβ receptor loss-of-function mutants, and that *ctl-1* function was required for a large part of the more than doubling of H$_2$O$_2$ resistance induced by those mutants (*Schiffer et al., 2020*). In our mRNA-seq analysis, wild-type nematodes grown at 25°C had 46% higher levels of *ctl-1* expression and 73% higher levels of *ctl-2* expression compared to nematodes grown at 20°C (*Figure 4G*), and ablation of the AFD neurons increased *ctl-1* expression by 46% and increased *ctl-2* expression by 89% (*Figure 4G*). Therefore,

the cultivation temperature and the AFD neurons regulated the expression of hydrogen peroxide defenses.

Last, we determined the extent to which the higher cultivation temperature and the ablation of the AFD neurons affected the expression of genes induced by toxic organic compounds, toxic metals, and radiation (*Eom et al., 2014*; *Greiss et al., 2008*; *Huffman et al., 2004*; *Lewis et al., 2009*; *Mueller et al., 2014*; *Sahu et al., 2013*; *Starnes et al., 2016*). Growth at 25°C did not increase the expression of genes induced by acrylamide, formaldehyde, benzene, silver, cadmium, arsenic, UVB rays, X rays, and gamma rays (*Figure 4—figure supplement 5A–I*), but ablation of AFD at 20°C induced all of those gene sets (*Figure 4—figure supplement 5A–I*). Therefore, ablation of the AFD sensory neurons induced genes normally induced by a wide variety of stressors in nematodes that were not exposed to those stressors, but the higher cultivation temperature only pre-induced a specific subset of genes that included hydrogen peroxide defenses and genes induced by peroxides.

## The high temperature-repressed INS-39 insulin/IGF1 hormone from the AFD sensory neurons lowers the nematode's peroxide resistance

To investigate how the AFD sensory neurons regulated the nematode's peroxide resistance, we took a candidate gene approach. We speculated that the AFD neurons signaled to target tissues via insulin/IGF1 peptide hormones because previous studies, including our own, showed that insulin/IGF1 signaling is a major determinant of peroxide resistance in *C. elegans* (*Schiffer et al., 2020*; *Tullet et al., 2008*). A recent single-neuron mRNA-seq study by the *C. elegans* Neuronal Gene Expression Map and Network consortium (CeNGEN) showed that AFD expresses many classes of peptide-hormone coding genes, including a subset of the 40 insulin/IGF1 genes in the genome: *ins-14*, *ins-15*, *ins-16*, *ins-39*, and *daf-28* (*Taylor et al., 2021*). We focused on the *ins-39* gene, which was highly expressed in AFD (Q. Ch'ng and J. Alcedo, personal communication) and was the only insulin/IGF1 gene with higher expression in AFD than in other neurons (*Taylor et al., 2021*).

To examine the expression of the *ins-39* gene in live nematodes, we used CRISPR/Cas9 genome editing to engineer a 'transcriptional' reporter that preserved the 5' and 3' cis-acting regulatory elements of the *ins-39* gene (*Tursun et al., 2009*) by inserting into the *ins-39* gene locus a SL2-spliced intercistronic region fused to the coding sequence of the green fluorescent protein (GFP) (*Figure 5A*). The *ins-39* reporter was expressed exclusively in the AFD neurons (*Figure 5B*). The level of *ins-39* gene expression in AFD was higher in nematodes grown at 20°C than in those grown at 25°C (*Figure 5C, D*). Therefore, temperature regulated *ins-39* gene expression in the AFD sensory neurons. Temperature perception by the AFD neurons requires TAX-4 cyclic GMP-gated channels, as the AFD neurons of *tax-4* mutants do not exhibit changes in calcium dynamics or thermoreceptor currents in response to warming or cooling (*Kimura et al., 2004*; *Ramot et al., 2008*). The temperature-dependent expression of the *ins-39* gene in the AFD neurons required the function of *tax-4*, as a *tax-4* null mutation nearly abolished *ins-39* gene expression in nematodes grown at either 20 or 25°C (*Figure 5E* and *Figure 5—figure supplement 1A*). Taken together, these findings suggested that the AFD neurons lowered the expression of the INS-39 insulin/IGF1 hormone in response to the cultivation temperature via a TAX-4-dependent process.

Next, we determined whether the INS-39 signal from AFD regulated the nematode's peroxide resistance. The *tm6467* null mutation in *ins-39* deletes 520 bases, removing almost all the *ins-39* coding sequence (*Figure 5A*), and inserts in that location 142-bases identical to an intervening sequence located between *ins-39* and its adjacent gene. In nematodes grown and assayed at 20°C, *ins-39(tm6467)* increased peroxide resistance by 26% relative to wild-type controls (*Figure 5F*). To determine whether *ins-39* gene expression in AFD was sufficient to lower peroxide resistance, we restored *ins-39(+)* expression only in the AFD neurons using the AFD-specific *gcy-8* promoter (*Inada et al., 2006*; *Yu et al., 1997*) in *ins-39(tm6467)* mutants. Expression of *ins-39(+)* only in AFD eliminated the increase in peroxide resistance of *ins-39(tm6467)* mutants (*Figure 5F*). Notably, the peroxide resistance of the two independent transgenic lines was 28% and 30% lower than that of wild-type controls, likely due to overexpression of the gene beyond wild-type levels. We conclude that the gene dose-dependent expression of *ins-39* in the AFD neurons regulated the nematode's peroxide resistance.

Last, we investigated whether cultivation temperature and INS-39 regulated *C. elegans* peroxide resistance via a common mechanism. In nematodes grown and assayed at 20°C, the *ins-39(tm6467)* null mutation increased peroxide resistance by 29% relative to wild-type controls (*Figure 5G*). In

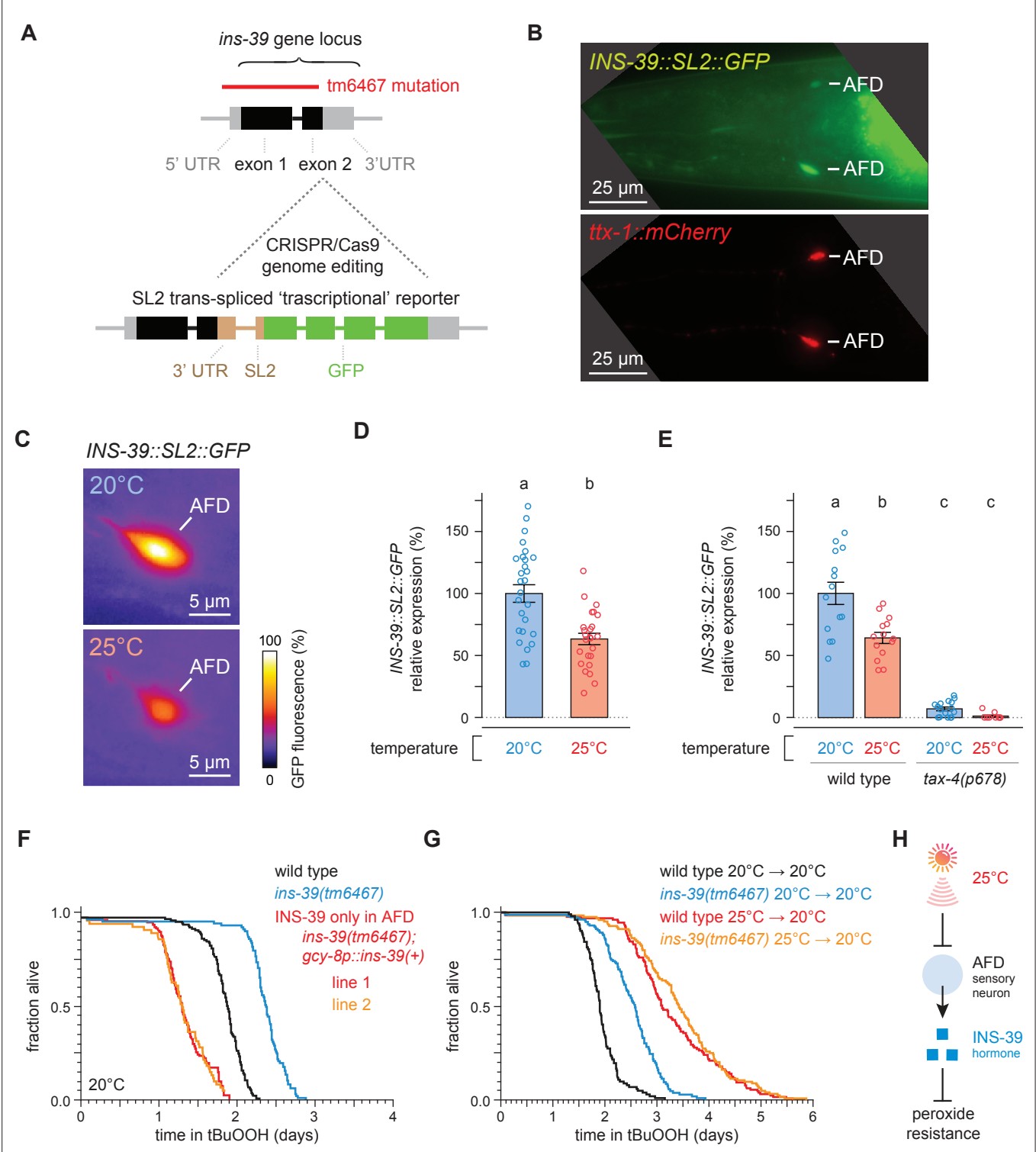

**Figure 5.** The high temperature-repressed INS-39 insulin/IGF1 hormone from the AFD sensory neurons lowers the nematode's peroxide resistance. (**A**) Schematic of the *ins-39* gene locus showing the CRISPR/Cas9 genome editing strategy used to engineer the *ins-39(oy167[ins-39::SL2::GFP])* 'transcriptional' reporter. The red line denotes the location of the *ins-39(tm6467)* deletion. (**B**) Example animal co-expressing the *ins-39(oy167[ins-39::SL2::GFP])* reporter (top panel) and the AFD-specific reporter *Ex[ttx-1p::TagRFP]* (bottom panel). The head region is shown and only the AFD neurons are detected. Lines indicate the AFD soma. Scale bar = 25 µm. (**C**) Representative images of the expression of the *ins-39(oy167[ins-39::SL2::GFP])* reporter in nematodes grown at 20°C (top panel) and 25°C (bottom panel) in one of the bilateral AFD neurons. Scale bar = 5 µm. (**D, E**) Quantification

*Figure 5 continued on next page*

*Figure 5 continued*

of the expression of the *ins-39(oy167[ins-39::SL2::GFP])* reporter. (**D**) Reporter expression was lower in nematodes grown at 20°C than at 25°C. Data are represented as mean ± s.e.m. Groups labeled with different letters exhibited significant differences ($n ≥ 25$ in both groups, $p < 0.0001$, analysis of variance [ANOVA]). (**E**) Reporter expression was nearly abolished in *tax-4(p678)* mutants. Data are represented as mean ± s.e.m. Groups labeled with different letters exhibited significant differences ($n ≥ 10$ in each group, $p < 0.0001$, Tukey HSD test) otherwise ($p > 0.05$). (**F**) Peroxide resistance of wild-type, *ins-39(tm6467)*, and *ins-39(tm6467)* with *ins-39(+)* reintroduced with the AFD-specific *gcy-8* promoter in nematodes grown and assayed at 20°C. (**G**) The *ins-39(tm6467)* mutation increased peroxide resistance in nematodes grown and assayed at 20°C, but did not further increase peroxide resistance in nematodes grown at 25°C and assayed at 20°C. (**H**) Sensory perception of the cultivation temperature regulates the nematodes' subsequent peroxide resistance. A high cultivation temperature lowers the expression of the AFD-specific INS-39 hormone, leading to the de-repression of the nematodes' peroxide defenses. Statistical analysis for panels (**F, G**) is in ***Supplementary file 5***.

The online version of this article includes the following source data and figure supplement(s) for figure 5:

**Source data 1.** Survival data for panels F, G.

**Source data 2.** Expression data for panels D, E.

**Figure supplement 1.** TAX-4 cyclic GMP-gated channels are required for *ins-39* gene expression in the AFD sensory neurons.

**Figure supplement 1—source data 1.** Survival data for panel B.

contrast, *ins-39(tm6467)* did not further increase peroxide resistance in nematodes grown at 25°C and assayed at 20°C (***Figure 5G***). Therefore, INS-39 lowered the nematodes' peroxide resistance in a manner dependent on the growth temperature history of the nematodes. We propose that at 20°C the AFD-specific hormone INS-39 represses the nematodes' peroxide defenses (***Figure 5H***). At 25°C, however, the AFD neurons express lower levels of INS-39, leading to the de-repression of the nematodes' peroxide defenses (***Figure 5H***).

The increase in peroxide resistance at 20°C caused by the *ins-39* null mutation was smaller than those caused by growth at 25°C in wild-type nematodes or by AFD ablation at 20°C. Therefore, in addition to INS-39, other AFD-derived signals likely regulated the induction of peroxide defenses in target tissues in response to growth at 25°C. We considered the possibility that the AFD neurons also regulated the nematodes' peroxide resistance through a process that required the neurotransmitter serotonin. Previous studies showed that serotonin is required for the induction of heat shock proteins in somatic tissues by AFD neurons in response to perception of a noxious 34°C heat shock (***Prahlad et al., 2008***; ***Tatum et al., 2015***), a much higher temperature than the 20 and 25°C cultivation temperatures we used in our studies. Serotonin biosynthesis requires the TPH-1 tryptophan hydroxylase (***Shivers et al., 2009***; ***Sze et al., 2000***). We found that the peroxide resistance of AFD-ablated nematodes was unaffected by the *tph-1(n4622)* null mutation (***Figure 5—figure supplement 1B***). Therefore, the AFD neurons regulated peroxide resistance in a serotonin-independent manner.

## DAF-16/FOXO functions in the intestine to increase the nematode's peroxide resistance in response to temperature-dependent signals from the AFD sensory neurons

To identify molecular determinants that might enable AFD to regulate the nematode's peroxide defenses via INS-39 in response to temperature, we investigated whether the changes in gene expression induced by temperature and AFD ablation mimicked those induced by specific transcription factors in response to reduced insulin/IGF1 signaling. The FOXO transcription factor DAF-16 is essential for the increase in peroxide resistance and most other phenotypes of mutants with reduced signaling by the insulin/IGF1 receptor, DAF-2 (***Kenyon et al., 1993***; ***Lin et al., 1997***; ***Ogg et al., 1997***; ***Schiffer et al., 2020***). Both higher temperature and AFD ablation increased the expression of genes directly upregulated by DAF-16 (***Kumar et al., 2015***; ***Figure 6A and B***) and increased the expression of genes upregulated in a *daf-16*-dependent manner in *daf-2(−)* mutants (***Murphy et al., 2003***; ***Figure 6—figure supplement 1A, B***). Genes directly upregulated by DAF-16 were disproportionately enriched among those upregulated significantly ($q$ value $<0.01$) by temperature and by AFD ablation, but not among those downregulated significantly by either intervention (***Supplementary file 6***). These findings were consistent with previous studies showing that the degree of nuclear localization of DAF-16 in the intestine increases from 20 to 25°C (***Wolf et al., 2008***). Together, these findings suggested that in response to cultivation temperature, reduced signaling by the AFD neurons might

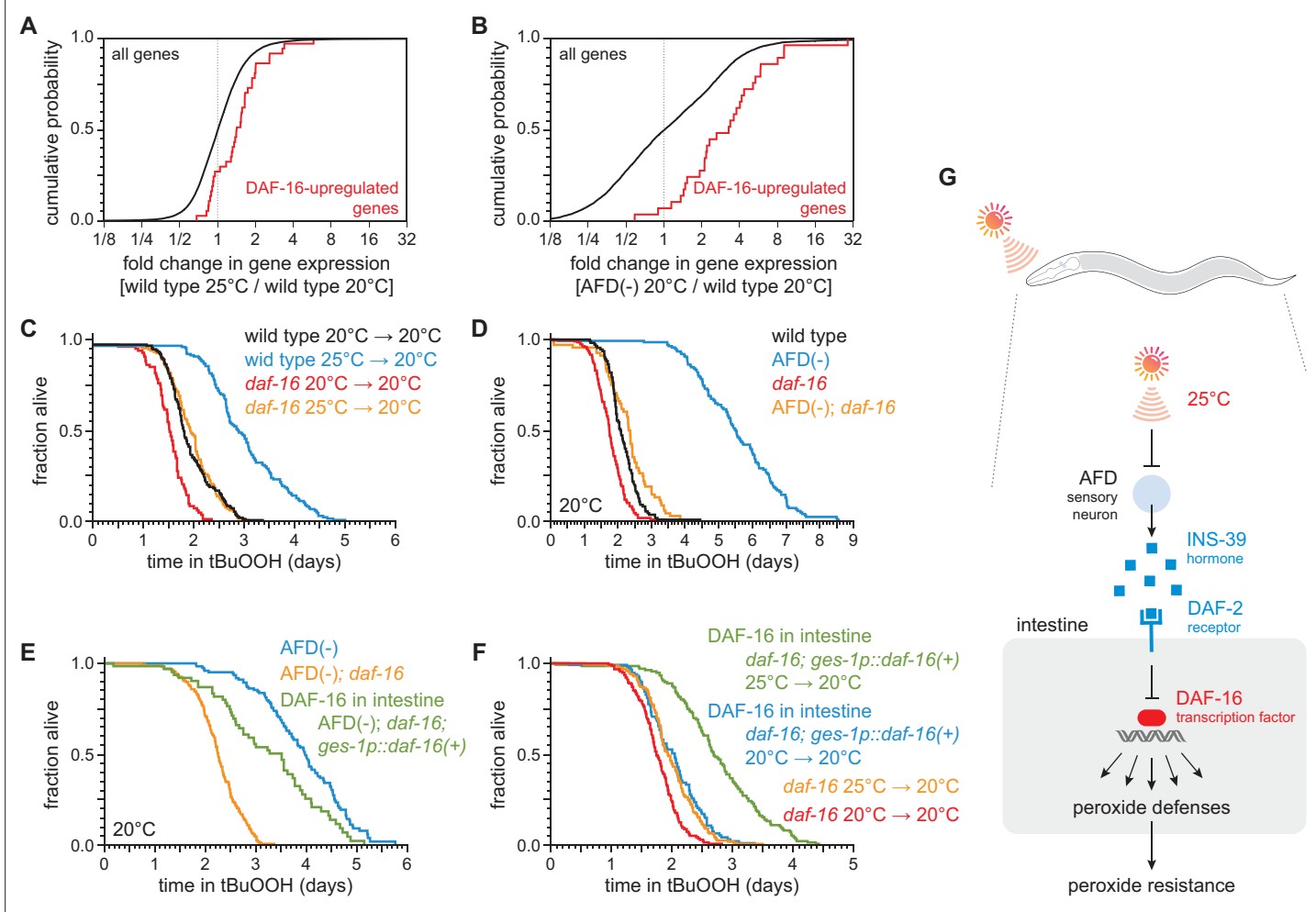

**Figure 6.** DAF-16/FOXO functions in the intestine to increase the nematode's peroxide resistance in response to temperature-dependent signals from the AFD sensory neurons. Genes directly upregulated by DAF-16 (***Kumar et al., 2015***) had higher expression (**A**) in nematodes grown at 25°C than in nematodes grown at 20°C and (**B**) in AFD-ablated nematodes grown at 20°C than in wild-type (unablated) nematodes grown at 20°C. *daf-16(mu86)* suppressed most of the increased peroxide resistance of (**C**) nematodes grown at 25°C and assayed at 20°C and (**D**) AFD-ablated nematodes grown at 20°C. (**E**) Peroxide resistance of AFD-ablated nematodes expressing *daf-16(+)* only in the intestine, AFD-ablated *daf-16(mu86)* controls, and AFD-ablated nematodes for reference. Nematodes were grown and assayed at 20°C. (**F**) Peroxide resistance of transgenic nematodes expressing *daf-16(+)* only in the intestine, and *daf-16(mu86)* controls. Nematodes were grown at the indicated temperatures and assayed at 20°C. (**G**) The AFD sensory neurons repress the expression of $H_2O_2$-protection services in the nematode's intestine via insulin/IGF1 signaling. AFD expresses high levels of INS-39 at the lower cultivation temperature (20°C), leading to repression of the CTL-1 and CTL-2 $H_2O_2$-degrading catalases and of other peroxide defenses. At the higher cultivation temperature (25°C), AFD lowers INS-39 expression, de-repressing the DAF-16/FOXO factor that increases the expression of peroxide defenses in the intestine. Statistical analyses for panels (**A, B**) are in ***Supplementary file 3*** and statistical analyses for panels (**C–F**) are in ***Supplementary file 7***.

The online version of this article includes the following source data and figure supplement(s) for figure 6:

**Source data 1.** Survival data for panels C–F.

**Figure supplement 1.** The AFD sensory neurons repress the expression of genes induced by DAF-16/FOXO.

**Figure supplement 1—source data 1.** Survival data for panel C.

induce the nematodes' peroxide defenses by increasing the activity of the DAF-16/FOXO transcription factor in target tissues.

To determine whether DAF-16 was required for the regulation of peroxide resistance by the AFD sensory neurons and by cultivation temperature, we examined the effects of a null mutation in *daf-16*. The *daf-16(mu86)* null mutation decreased peroxide resistance in nematodes grown at 25°C and assayed at 20°C by 35%, a greater extent than the 21% reduction in peroxide resistance induced by

that mutation in nematodes grown and assayed at 20°C (*Figure 6C*). Similarly, in nematodes grown and assayed at 20°C, the *daf-16(mu86)* null mutation decreased the peroxide resistance of AFD-ablated nematodes by 58% but caused only a 18% reduction in peroxide resistance in unablated (wild-type) nematodes (*Figure 6D*). Therefore, the regulation of peroxide resistance by the AFD sensory neurons and by cultivation temperature was, in part, dependent on the DAF-16/FOXO transcription factor.

Next, we set out to identify which target tissues were important for increasing *C. elegans* peroxide resistance via DAF-16 in response temperature-dependent signals from the AFD sensory neurons. First, we determined the extent to which restoring *daf-16(+)* expression in a specific tissue, using a tissue-specific promoter, increased peroxide resistance in AFD-ablated *daf-16* mutants. We speculated that *daf-16* might function in the intestine, because our transcriptomic analysis showed that both higher temperature and AFD ablation upregulated gene expression in the intestine (*Supplementary file 4*). Consistent with that prediction, in AFD-ablated *daf-16(mu86)* mutants grown and assayed at 20°C, restoring *daf-16(+)* expression only in the intestine was sufficient to partially rescue peroxide resistance to a level almost comparable to that of AFD-ablated *daf-16(+)* nematodes (*Figure 6E*). Therefore, *daf-16(+)* functioned in the intestine to increase peroxide resistance in AFD-ablated nematodes.

We followed a similar scheme to determine whether intestinal DAF-16 increased the nematode's peroxide resistance in response to cultivation temperature. In *daf-16(mu86)* mutants grown and assayed at 20°C, restoring *daf-16(+)* expression only in the intestine increased peroxide resistance by a small amount, 15% (*Figure 6F*), indicating that *daf-16(+)* function in the intestine was sufficient to increase peroxide resistance when the AFD neurons were present. Notably, in *daf-16(mu86)* mutants grown at 25°C and assayed at 20°C, restoring *daf-16(+)* expression only in the intestine increased peroxide resistance to a greater extent, 39%, than in nematodes grown and assayed at 20°C (*Figure 6F*). Therefore, temperature regulated the size of the increase in peroxide resistance induced by *daf-16(+)* function in the intestine.

Based on these observations, we propose that communication between AFD sensory neurons and the intestine via insulin/IGF1 signaling enables *C. elegans* to regulate their peroxide defenses in response to perception of the cultivation temperature (*Figure 6G*). At a higher cultivation temperature, lower INS-39 expression by AFD leads to a decrease in signaling by the DAF-2 receptor, which enables DAF-16/FOXO transcription factors to induce peroxide defenses to a greater extent than at the lower cultivation temperature.

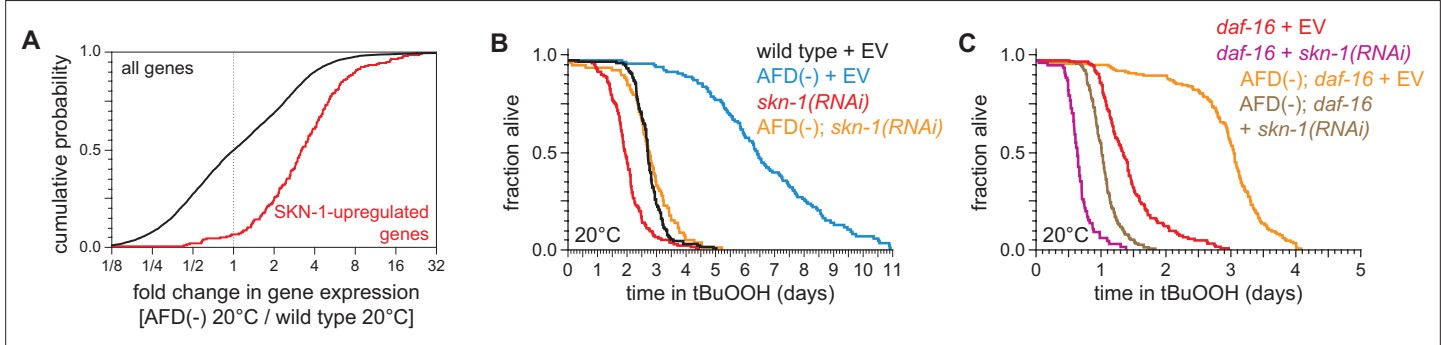

**Figure 7.** SKN-1/NRF and DAF-16/FOXO collaborate to increase the nematodes' peroxide resistance in response to signals from the AFD sensory neurons. (**A**) Genes upregulated by *skn-1(+)* in wild-type nematodes (*Oliveira et al., 2009*) had higher expression in AFD-ablated nematodes grown at 20°C than in wild-type (unablated) nematodes grown at 20°C. (**B**) *skn-1(RNAi)* suppressed most of the increased peroxide resistance of AFD-ablated nematodes grown at 20°C. Control RNAi consisted of feeding the nematodes the same bacteria but with the empty vector (EV) plasmid pL4440 instead of a plasmid targeting *skn-1*. (**C**) *skn-1(RNAi)* lowered peroxide resistance to a greater extent in AFD-ablated *daf-16(mu86)* mutants at 20°C than in (unablated) *daf-16(mu86)* mutants at 20°C. Statistical analysis for panel (**A**) is in *Supplementary file 3* and statistical analyses for panels (**B, C**) are in *Supplementary file 8*.

The online version of this article includes the following source data and figure supplement(s) for figure 7:

**Source data 1.** Survival data for panels B, C.

**Figure supplement 1.** The AFD sensory neurons repress the expression of genes induced by SKN-1/NRF.

## SKN-1/NRF and DAF-16/FOXO collaborate to increase the nematodes' peroxide resistance in response to AFD ablation

We next examined whether other transcription factors might act with DAF-16 to increase peroxide resistance in AFD-ablated nematodes at 20°C. The DAF-3/coSMAD transcription factor (*Patterson et al., 1997*) is required for almost all of the increase in peroxide resistance induced by lack of DAF-7/TGFβ signaling from the ASI sensory neurons (*Schiffer et al., 2020*). In contrast, the *daf-3(mgDf90)* null mutation did not affect the peroxide resistance of AFD-ablated nematodes (*Figure 6—figure supplement 1C*). Therefore, unlike the ASI neurons, the AFD neurons did not regulate the nematodes' peroxide resistance via DAF-3/coSMAD.

Like DAF-16, the NRF ortholog SKN-1 increases *C. elegans* peroxide resistance in response to reduced DAF-2 signaling (*Tullet et al., 2008*). The expression of genes upregulated by *skn-1(+)* in wild-type nematodes (*Oliveira et al., 2009*) and in *daf-2* loss-of-function mutants (*Ewald et al., 2015*) was increased by AFD ablation but was not increased by higher cultivation temperature (*Figure 7A* and *Figure 7—figure supplement 1A–C*), suggesting that SKN-1 might increase peroxide resistance in AFD-ablated nematodes. Knockdown of *skn-1* via RNA interference (RNAi) decreased the peroxide resistance of AFD-ablated nematodes by 58%, but caused a smaller, 27%, reduction in peroxide resistance in wild-type nematodes (*Figure 7B*). RNAi of *skn-1* also decreased the peroxide resistance of AFD-ablated *daf-16* mutants (*Figure 7C*). In addition, RNAi of *skn-1* caused a larger reduction in peroxide resistance in *daf-16* mutants when the AFD neurons were ablated than when those neurons were present (*Figure 7C*), suggesting that DAF-16 and SKN-1 had non-overlapping roles in promoting peroxide resistance in AFD-ablated nematodes.

We propose that when nematodes are cultured at 20°C, the AFD neurons promote signaling by the DAF-2 insulin/IGF1 receptor in target tissues, which subsequently lowers the nematode's peroxide resistance by repressing transcriptional activation by SKN-1/NRF and DAF-16/FOXO. However, this repression is not complete, because both *daf-16(mu86)* and *skn-1(RNAi)* lowered peroxide resistance at 20°C when the AFD neurons were present. It is also likely that DAF-16 and SKN-1 are not the only factors that contribute to peroxide resistance in AFD-ablated nematodes at 20°C, because AFD ablation increased peroxide resistance in *daf-16(mu86); skn-1(RNAi)* nematodes, albeit to a lesser extent than in *daf-16(+)* or *skn-1(+)* backgrounds.

## DAF-16/FOXO potentiates the changes in gene expression induced by the AFD sensory neurons

What role does the DAF-16/FOXO transcription factor play in regulating gene expression in response to signals from the AFD sensory neurons? In principle, DAF-16 could mediate all, some, or none of the changes in gene expression induced by AFD ablation. To distinguish between these possibilities, we used genome-wide epistasis analysis (*Angeles-Albores et al., 2018*) to compare the transcriptomes of unablated *daf-16(+)* [wild-type] nematodes, unablated *daf-16(mu86)* mutants, AFD-ablated *daf-16(+)* nematodes, and AFD-ablated *daf-16(mu86)* mutants, on day 2 of adulthood and grown at 20°C. This analysis quantified the extent to which DAF-16 affected gene expression differently in AFD-ablated and unablated nematodes (*Figure 8A* and *Figure 8—figure supplement 1A*).

We found that lack of *daf-16* gene function in unablated nematodes at 20°C significantly lowered the expression of just two genes and significantly increased the expression of none, out of 7387 genes detected (*q* value <0.01) (*Figure 8B*). Previous transcriptomic studies in *daf-2(−)* mutants—unlike our study, which was conducted in *daf-2(+)* nematodes—have identified thousands of genes whose expression was regulated by *daf-16* (*Kumar et al., 2015*; *Lin et al., 2018*; *Murphy et al., 2003*). We found that lack of *daf-16* gene function in AFD-unablated [*daf-2(+)*] nematodes lowered the expression of genes directly upregulated by DAF-16 (*Kumar et al., 2015*; *Figure 8D*) and lowered the expression of genes upregulated in a *daf-16*-dependent manner in *daf-2(−)* mutants (*Murphy et al., 2003*; *Figure 8E*). However, these effects were small, averaging to just a 10% decrease in expression. These small *daf-16*-dependent effects on gene expression suggest that DAF-16 function was almost fully repressed by DAF-2 at 20°C. While these transcriptional effects were small, they could nevertheless contribute to the peroxide resistance of wild-type nematodes at 20°C. DAF-16 may also play a larger role in gene expression in unablated nematodes after peroxide exposure.

In contrast, when the AFD neurons were ablated, lack of *daf-16* gene function induced much broader changes in gene expression, lowering the expression of 431 genes and increasing the

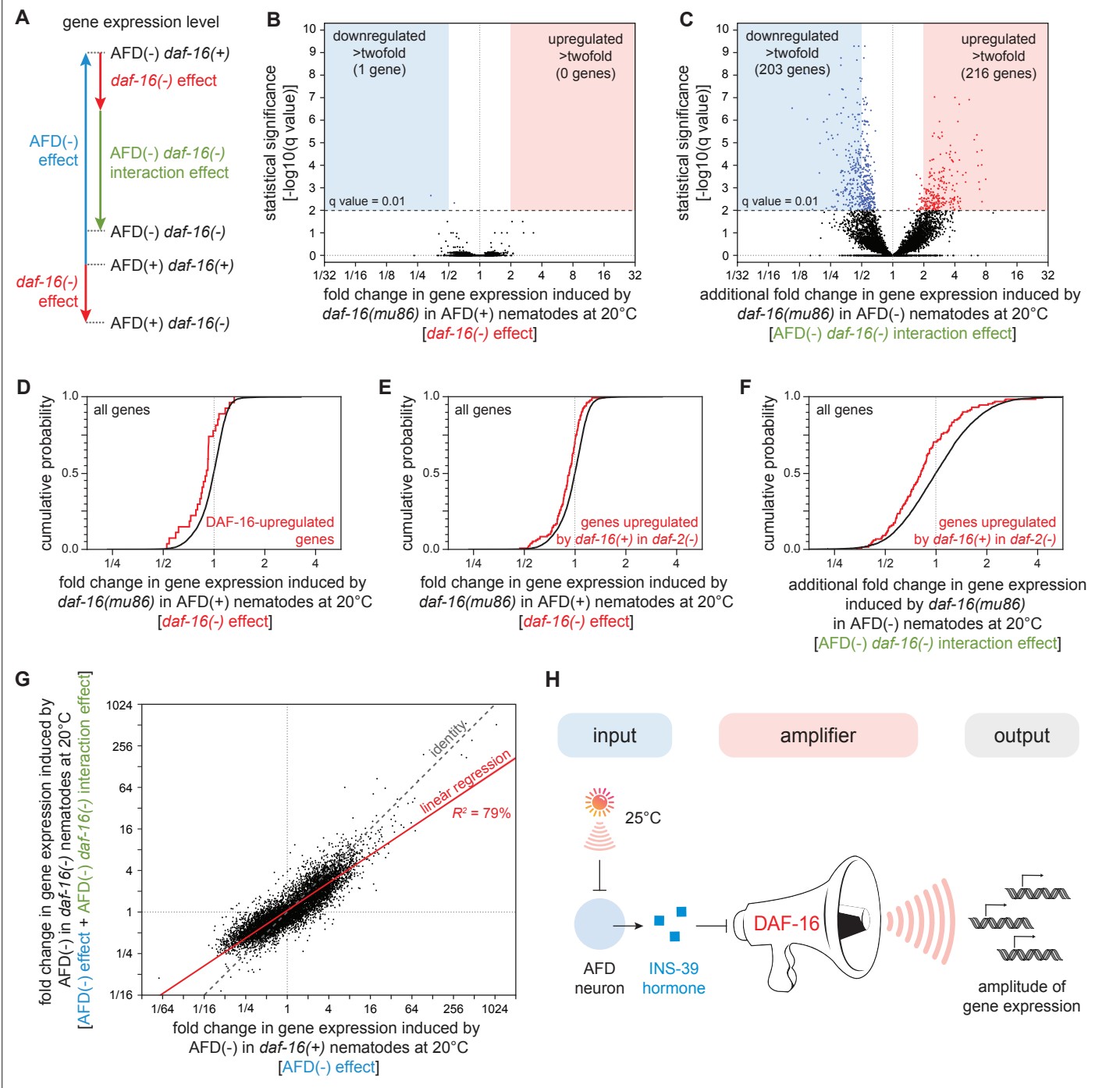

**Figure 8.** DAF-16/FOXO potentiates the changes in gene expression induced by the AFD sensory neurons. (**A**) We performed mRNA sequencing (mRNA-seq) on wild-type [AFD(+) *daf-16(+)*], *daf-16(mu86)* null mutants [AFD(+) *daf-16(–)*], AFD-ablated nematodes [AFD(−) *daf-16(+)*], and AFD-ablated *daf-16(mu86)* null mutants [AFD(−) *daf-16(−)*] grown at 20°C, and used an epistasis model to quantify the extent to which AFD ablation and *daf-16* mutation affected the expression of each gene, relative to wild-type, in terms of the independent effects induced by AFD ablation (blue arrow) and by lack *daf-16* gene function (red arrow), and the additional effect induced by the interaction between AFD ablation and lack *daf-16* gene function (green arrow). Volcano plots showing the level and statistical significance of (**B**) the changes in gene expression induced by lack *daf-16* gene function in unablated nematodes at 20°C and (**C**) the additional changes in gene expression induced by lack *daf-16* gene function in AFD-ablated nematodes at 20°C. Genes up- and downregulated significantly (*q* value < 0.01) are shown in red and blue, respectively. (**D, E**) Effect of lack *daf-16* gene function in unablated nematodes at 20°C on the expression of (**C**) genes directly upregulated by DAF-16 (***Kumar et al., 2015***) and (**D**) genes upregulated in a *daf-16*-dependent manner in *daf-2(−)* mutants (***Murphy et al., 2003***). (**F**) Effect of the additional changes in gene expression induced by lack *daf-16* gene function in AFD-ablated nematodes at 20°C on the expression of genes upregulated in a *daf-16*-dependent manner in *daf-2(−)* mutants (***Murphy***

*Figure 8 continued on next page*

*Figure 8 continued*

*et al., 2003*). (**G**) The effect of AFD ablation on gene expression at 20°C was systematically smaller in *daf-16(mu86)* mutants (*y*-axis) than in *daf-16(+)* nematodes (*x*-axis). Linear regression fit is shown as a red line flanked by a red area marking the 95% confidence interval of the fit. (**H**) The DAF-16/FOXO transcription factor amplifies the changes in gene expression induced by AFD ablation. This means that DAF-16 determines the gene-expression responsiveness, but not the response to signals from the AFD sensory neurons. Statistical analyses for panels (**D–F**) are in *Supplementary file 3*.

The online version of this article includes the following source data and figure supplement(s) for figure 8:

**Source data 1.** mRNA sequencing (mRNA-seq) analysis data.

**Figure supplement 1.** Gene expression is regulated by the interaction of the AFD sensory neurons and DAF-16/FOXO.

**Figure supplement 2.** DAF-16/FOXO potentiates the changes in gene-set expression induced by the AFD sensory neurons.

**Figure supplement 2—source data 1.** WormCat analysis data.

expression of 238 genes (*q* value < 0.01) (*Figure 8C*). In addition, in AFD-ablated nematodes, lack of *daf-16* lowered the expression of genes upregulated in a *daf-16*-dependent manner in *daf-2(−)* mutants (*Murphy et al., 2003*) to a greater degree than in unablated nematodes (*Figure 8F*). Taken together, these findings showed that DAF-16 played a larger role in regulating gene expression at 20°C when the AFD neurons were ablated.

Finally, we investigated how the gene-regulatory influence of the AFD neurons depended quantitatively on DAF-16. The AFD neurons and DAF-16 worked together to regulate gene expression, because the extent to which DAF-16 affected gene expression deviated systematically from the level of gene expression predicted if AFD and DAF-16 acted independently (*Figure 8—figure supplement 1B*). To examine how the AFD neurons and DAF-16 jointly regulated gene expression, we compared the extent to which ablation of the AFD neurons affected gene expression in *daf-16(m86)* mutants and in *daf-16(+)* nematodes. The effect of AFD ablation on gene expression was systematically smaller in *daf-16(mu86)* mutants than in *daf-16(+)* nematodes ($R^2$ = 79%, slope = 0.67, p < 0.0001, *Figure 8G*). Using simulations, we showed that effect was robust despite the uncertainty in our estimates of how much AFD and DAF-16 affected the expression of each gene (see Materials and methods). In addition, we found that the extent to which AFD ablation affected the average expression of sets of genes with related functions (*Higgins et al., 2022*; *Holdorf et al., 2020*) was systematically lower in *daf-16(mu86)* mutants than in *daf-16(+)* nematodes ($R^2$ = 86%, slope = 0.67, p < 0.0001, *Figure 8—figure supplement 2*). Therefore, the size of the effect of AFD ablation on gene expression was systematically smaller when the contribution of DAF-16 to gene expression was removed. We conclude that the DAF-16/FOXO transcription factor potentiates the changes in gene expression induced by ablation of the AFD sensory neurons (*Figure 8H*).

## Discussion

Across the tree of life, organisms face the lethal threat from hydrogen peroxide attack (*Avery and Morgan, 1924*; *Imlay, 2018*). This threat is inherently temperature dependent, because the reactivity of hydrogen peroxide increases with temperature (*Arrhenius, 1889*; *Evans and Polanyi, 1935*; *Eyring, 1935*). In this study, we found that *C. elegans* nematodes use temperature information to deal with the lethal threat of hydrogen peroxide produced by the pathogenic bacterium *E. faecium*: when a pair of the nematodes' neurons sensed a high cultivation temperature, they preemptively induced the nematodes' hydrogen peroxide defenses. To our knowledge, the findings described here provide the first evidence of a multicellular organism inducing their defenses to a chemical when they sense an inherent enhancer of the reactivity of that chemical.

### Temperature perception by sensory neurons regulates *C. elegans* hydrogen peroxide defenses

We show here that a small increase in temperature—within the range that *C. elegans* nematodes prefer in nature (*Crombie et al., 2019*)—increases the nematodes' sensitivity to killing by environmental peroxides and by hydrogen peroxide ($H_2O_2$) produced by the pathogenic bacterium *E. faecium*. These effects were not due to damage to the nematodes by the higher temperature but, instead, occurred despite the nematodes inducing protective defenses in response to experiencing the higher temperature before peroxide exposure.

We found that *C. elegans* deals with the enhanced threat posed by environmental peroxides at high cultivation temperature by coupling the induction of their $H_2O_2$ defenses to the perception of temperature by their AFD sensory neurons. These neurons have specialized sensory endings that are the primary thermoreceptors of the nematode, enabling them to adjust their behavior and heat defenses in response to temperature (*Goodman and Sengupta, 2019*; *Hedgecock and Russell, 1975*; *Prahlad et al., 2008*). The AFD sensory neurons used an INS-39 insulin/IGF1 hormone—which they expressed exclusively—to relay temperature information to the intestine, the tissue that provided $H_2O_2$-protection services to the nematode. At a low cultivation temperature, AFD expressed high levels of INS-39, leading to repression of the CTL-1 and CTL-2 catalases and of other peroxide-induced genes. However, at a high cultivation temperature, AFD lowered INS-39 expression, leading to the induction of peroxide defenses by the DAF-16/FOXO transcriptional activator.

What mechanisms regulate *ins-39* gene expression in the AFD neurons in response to cultivation temperature? On a short timescale of seconds to minutes, the AFD neurons respond to changes in temperature by transiently increasing intracellular $[Ca^{2+}]$ and changing thermoreceptor currents through a process dependent on TAX-4 cyclic GMP-gated channels (*Kimura et al., 2004*; *Ramot et al., 2008*). On a longer timescale of hours, changes in temperature can modulate gene expression within AFD through a process mediated in part by intracellular $[Ca^{2+}]$ via the calcium/calmodulin-dependent protein kinase CMK-1 (*Ippolito et al., 2021*; *Yu et al., 2014*). Interestingly, the baseline intracellular $[Ca^{2+}]$ in AFD was lower in nematodes grown continuously at 25°C than in those at 15°C, although levels at 20°C were not assessed in that work (*Ippolito et al., 2021*). Given that *tax-4* is essential for *ins-39* gene expression at both 20 and 25°C, it will be interesting to determine how cultivation temperature and TAX-4 act to regulate *ins-39* gene expression in AFD on different timescales.

The repression of peroxide-protection services by the AFD neurons at the lower cultivation temperature did not rely on the neurotransmitter serotonin, unlike the induction of heat defenses by these neurons in response to 34°C heat shock (*Tatum et al., 2015*). AFD ablation at 20°C also induced gene sets expressed at higher levels in response to low (15°C) cultivation temperature (*Gómez-Orte et al., 2018*) and gene sets induced by high heat (30°C) (*McCarroll et al., 2004*), but those gene sets were not induced by high cultivation temperature (25°C). Therefore, the AFD sensory neurons repressed gene sets regulated by noxious heat, high cultivation temperature, and low temperature. It is possible that the AFD neurons respond to different temperature ranges by regulating the expression of specific gene sets in target tissues via different signals. In addition to expressing INS-39, these neurons express other peptide hormones—including hormones in the insulin/IGF1, FMRFamide, pigment dispersal factor, and oxytocin–vasopressin families (*Barrios et al., 2012*; *Beets et al., 2012*; *Chen et al., 2016*; *Kim and Li, 2004*; *Taylor et al., 2021*). It is possible that the AFD neurons also regulate the expression of specific sets of genes through temperature-independent signals; these signals could either be constitutive or regulated by other inputs sensed by AFD, such as carbon dioxide (*Bretscher et al., 2011*) and magnetic fields (*Vidal-Gadea et al., 2015*). We conclude that the AFD thermosensory neurons play a central role in the regulation of distinct systemic responses to temperature.

## Target tissues control their responsiveness to sensory signals via DAF-16/FOXO

Using genome-wide epistasis analysis we showed that the DAF-16/FOXO transcription factor potentiated the changes in gene expression induced by AFD ablation at 20°C. This means that DAF-16 determined the responsiveness, but not the response, of gene expression at 20°C to signals from AFD. We reason that while *C. elegans* cells manage the challenge of deciding when to express specific genes by relinquishing control of that decision to signals from the AFD sensory neurons, they retain control of their responsiveness to those signals via the DAF-16/FOXO factor. Intestinal DAF-16 is also regulated by a wide variety of factors, including FLP-6 FMRFamide signals from the AFD neurons (*Chen et al., 2016*), insulin/IGF1 signals from sensory neurons other than AFD (*Artan et al., 2016*; *Zhang et al., 2018*), and signals from the germline (*Berman and Kenyon, 2006*; *Hsin and Kenyon, 1999*); and by signal-transduction pathways other than the insulin/IGF1 pathway, including the JNK pathway (*Oh et al., 2005*), AMPK pathway (*Greer et al., 2007*), TGFβ pathway (*Liu et al., 2004*; *Narasimhan et al., 2011*; *Shaw et al., 2007*), TOR pathway (*Robida-Stubbs et al., 2009*), and TRPA pathway (*Xiao et al., 2013*; *Zhang et al., 2015*). Therefore, we expect

those factors to determine via DAF-16 the gene-expression responsiveness to signals from the AFD neurons.

Multiple types of information appear to converge in a common mechanism to regulate the induction of intestinal peroxide defenses in *C. elegans*. Previously, we found that the SKN-1 and DAF-16 transcription factors collaborated to mediate the induction of peroxide defenses in response to information about food levels, sensed by the ASI neurons and communicated to the intestine via a TGFβ-insulin/IGF1 hormone relay (*Schiffer et al., 2020*). DAF-16 and SKN-1 functions in the intestine are also regulated by signals from the germline (*Berman and Kenyon, 2006*; *Ghazi et al., 2009*; *Steinbaugh et al., 2015*), and TRPA-1 channels in the intestine regulate DAF-16 function in that tissue at 15°C (*Xiao et al., 2013*). In insects and mammals, insulin/IGF1 signaling components also regulate cellular antioxidant defenses (*Brunet et al., 2004*; *Clancy et al., 2001*; *Holzenberger et al., 2003*; *Tatar et al., 2003*). It will be interesting to explore the extent to which sensory-neuronal and other signals collaborate to regulate hydrogen peroxide defenses via insulin/IGF1 signaling in all animals.

## Ideas and speculation: faithful assessment of a threat by sensing the enhancer of that threat

What does *C. elegans* accomplish by coupling the induction of hydrogen peroxide defenses to sensory perception of high cultivation temperature? To address this question, in this section we consider the specificity of the nematode's strategy for dealing with the threat of hydrogen peroxide and identify the strategy's unique features from a chemical-kinetic perspective.

How specific is the nematode's strategy? One possibility was that temperature defenses might induce cross-protection from the stress caused by $H_2O_2$. However, high cultivation temperature (25°C) increased the expression of the intestinal catalases CTL-1 and CTL-2, which are enzymes specialized for degrading $H_2O_2$ (*Amara et al., 2001*; *Loew, 1901*; *Mishra and Imlay, 2012*). A second possibility was that the high cultivation temperature induced a broad set of defense responses that included those triggered by peroxides. However, that was not the case, because gene sets induced by toxic metals, organic compounds, and high-energy radiation were not induced at 25°C relative to 20°C. Therefore, the defenses induced by high cultivation temperature were specialized and included those for coping with the stress induced by hydrogen peroxide.

What are the strategy's unique features? We considered the possibility that the strategy of using temperature information to preemptively induce hydrogen peroxide defenses is adaptive because when *C. elegans* nematodes encounter a higher cultivation temperature in their natural habitat, they are more likely to subsequently encounter $H_2O_2$ and, therefore, need $H_2O_2$ protection. Such a strategy, called adaptive prediction (*Mitchell et al., 2009*), is used by the bacteria *E. coli* and *Vibrio cholerae*, and by the yeasts *Saccharomyces cerevisiae* and *Candida albicans*, to sequentially induce specific defenses based on the typical order of stresses they encounter in their respective ecological settings (*Mitchell et al., 2009*; *Rodaki et al., 2009*; *Schild et al., 2007*; *Tagkopoulos et al., 2008*). Adaptive prediction would provide *C. elegans* with a guess of when to induce their $H_2O_2$ defenses based on how often and how quickly high temperature is followed by $H_2O_2$ exposure in *C. elegans'* ecological setting (*Levins, 1968*; *Mitchell and Pilpel, 2011*).

Adaptive prediction provides a plausible explanation for why *C. elegans* evolved a regulatory mechanism coupling temperature perception to $H_2O_2$ defense. However, in our opinion, that explanation is insufficient, because it does not incorporate a key feature of the nematode's strategy: the chemical constraint linking temperature and hydrogen peroxide reactivity removes guesswork from the strategy. Contrary to the expectation from adaptive prediction, *C. elegans* nematodes are not guessing that in their ecological setting increasing temperature leads to a higher $H_2O_2$ threat; instead, in all ecological settings the nematodes' proteins, nucleic acids, and lipids are inherently more likely to be damaged by $H_2O_2$ with increasing temperature because those chemical reactions necessarily run faster with increasing temperature (*Arrhenius, 1889*; *Evans and Polanyi, 1935*; *Eyring, 1935*). This chemical constraint means that by coupling the induction of $H_2O_2$ defenses to the perception of high temperature, the nematodes are not guessing; instead, they are assessing faithfully the threat that hydrogen peroxide poses. We refer to this distinct strategy as 'enhancer sensing' (*Figure 9*).

Enhancer sensing provides a new framework for understanding the adaptive value of strategies coupling the induction of defense responses to the perception of inputs that inherently modulate the need for those defenses. In a classical stress response, the strategy provides faithful information about

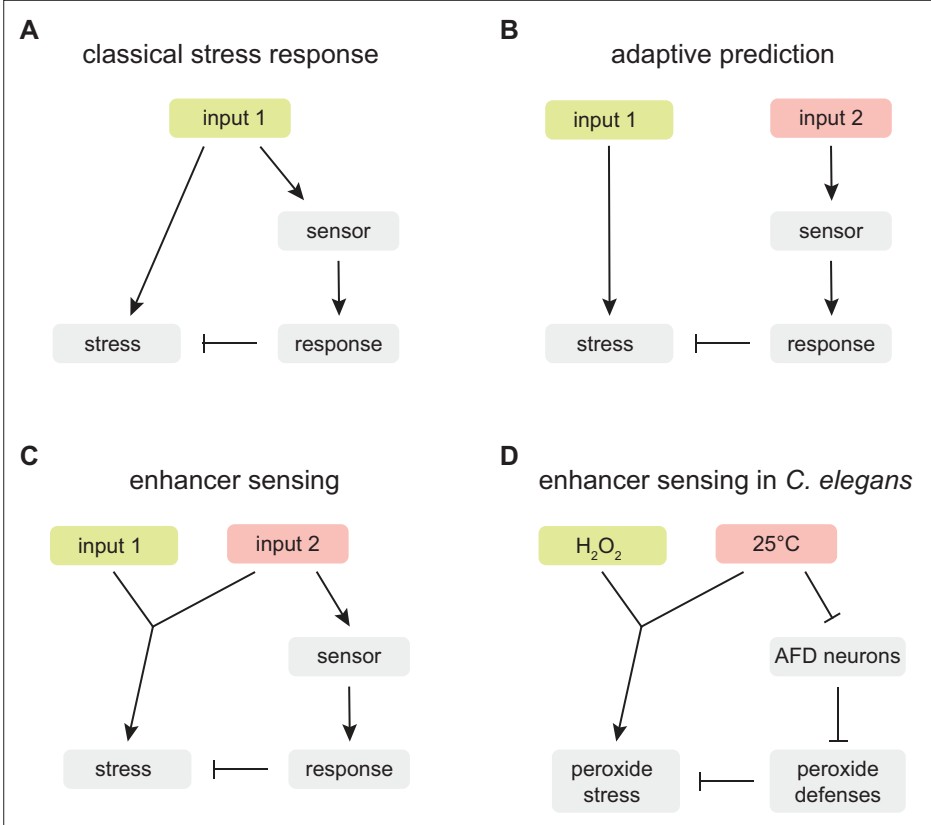

**Figure 9.** An enhancer sensing strategy enables *C.elegans* to assess faithfully the threat of hydrogen peroxide using temperature information. (**A**) Classical stress response: the strategy provides faithful information about the threat the organism faces because the response that enables the organism to cope with the stress induced by input 1 is coupled to the perception of input 1. (**B**) Adaptive prediction: the strategy provides a guess (whose predictive value matches the co-occurrence of inputs 1 and 2 in the ecological setting of the organism) but not necessarily faithful information about the threat the organism faces, because input 2 does not induce (nor affect the capacity of input 1 to induce) the stress that the organism attempts to cope with by inducing a response to input 2. (**C**) Enhancer sensing: the strategy provides faithful information about the threat the organism faces because the capacity of input 1 to induce a stress is modulated by input 2 and, therefore, perception of either input provides information about the threat posed by the interaction of those inputs. (**D**) The nematode *C. elegans* uses an enhancer sensing strategy that couples the de-repression of specific $H_2O_2$ defenses to the sensory perception of high temperature, an inherent enhancer of the reactivity of $H_2O_2$.

the threat the organism faces because the response that enables the organism to cope with the stress induced by an input is coupled to the perception of that input (*Figure 9A*). In enhancer sensing, an input's capacity to induce a stress is modulated by another input; the strategy provides faithful information about the threat the organism faces because perception of either input provides information about the threat posed by the interaction of those inputs (*Figure 9C*). In contrast, in adaptive prediction, the sensed input does not induce (nor modulate the capacity of another input to induce) the stress that the organism attempts to cope with by inducing a response to that input; as a result, the strategy does not necessarily provide faithful information about the threat the organism faces; instead the strategy provides a guess whose predictive value matches the co-occurrence in the ecological setting of the organism of the input that induces the stress and the input that is sensed (*Figure 9B*).

We show here that *C. elegans* couples $H_2O_2$ defense to the perception of high temperature. We expect this enhancer sensing strategy's output (the level of $H_2O_2$ defense) to provide *C. elegans* with an evolutionarily optimal strategy across ecologically relevant inputs (cultivation temperatures) (*Kussell and Leibler, 2005*; *Maynard, 1982*; *Wolf et al., 2005*). This strategy is implemented at the organismic level through the division of labor between the AFD neurons, which sense and broadcast temperature information, and the intestine, which responds to that information by providing

H$_2$O$_2$ defense (*Figure 9D*). Ascertaining that *C. elegans* uses this strategy does not depend on the temperature information broadcast by the AFD neurons exclusively regulating defense responses to temperature-dependent threats, because the regulation of defenses toward temperature-insensitive threats could affect the efficacy of defenses toward temperature-dependent threats; for example, suppressing defenses toward a temperature-insensitive threat would be beneficial if those defenses interfered with H$_2$O$_2$ defense or depleted energy resources contributing to H$_2$O$_2$ defense.

## Ideas and speculation: limitations and unanswered questions

Because the studies presented here are the first to identify an enhancer sensing strategy, we do not know the extent to which this type of strategy is common across organisms. However, many previous findings may be indicative of enhancer sensing. For example, in the case of the regulation of H$_2$O$_2$ defenses by temperature, previous studies have shown that H$_2$O$_2$ defenses are induced in response to high temperature in a wide variety of organisms, including bacteria (*Engelmann et al., 1995*; *Mossialos et al., 2006*), yeasts (*Deegenaars and Watson, 1997*; *Mitchell et al., 2009*; *Wieser et al., 1991*), plants (*Hu et al., 2021*; *Nishizawa et al., 2006*; *Panchuk et al., 2002*), cnidarians (*Dash and Phillips, 2012*), and human HeLa cells (*Pallepati and Averill-Bates, 2010*).

In addition, we do not know the extent to which enhancer sensing strategies couple temperature perception to multiple defense responses within any organism. In their natural habitat, *C. elegans* nematodes encounter many chemicals that, like H$_2$O$_2$, are inherently more reactive at higher temperatures. However, it is difficult to predict whether enhancer sensing strategies coupling temperature perception to defense toward those chemicals would be likely to provide a high adaptive value because we do not know the extent to which those chemicals are common, abundant, and reactive enough to cause consequential damage within the temperature range that *C. elegans* experience in their ecological setting. Because growth at 25°C did not induce gene sets induced by acrylamide, formaldehyde, benzene, silver, cadmium, and arsenic, we expect that resistance to lethal concentrations of these toxic chemicals will not increase with pre-exposure to 25°C. However, it is likely that other temperature ranges sensed by the AFD neurons might regulate resistance to those chemicals, because AFD ablation at 20°C induced the gene sets induced by each of those chemicals. In the future, we plan to determine the extent to which *C. elegans* uses enhancer sensing strategies to couple the perception of specific temperature ranges to the induction of defenses toward these and other toxic chemicals, and whether those strategies rely on temperature perception and broadcasting by AFD and other temperature-sensing neurons. More broadly, it will be interesting to determine the extent to which enhancer sensing strategies are used throughout the tree of life to couple specific defense responses to the perception of inputs that enhance the need for those defenses.

## *C. elegans* relies on a combination of strategies to deal with the threat of hydrogen peroxide

*C. elegans* decides when to induce behavioral and cellular H$_2$O$_2$ defenses by relying on many classes of sensory neurons (*Bhatla and Horvitz, 2015*; *Schiffer et al., 2020*; *Schiffer et al., 2021*). The function of these neurons can be understood in terms of their roles in different strategies enabling the nematode to deal with the lethal threat of environmental H$_2$O$_2$: enhancer sensing, adaptive prediction, and classical stress response. The AFD sensory neurons play a role in an enhancer sensing strategy by coupling the induction of specific H$_2$O$_2$ defenses to temperature perception. The ASI sensory neurons play a role in adaptive prediction by repressing H$_2$O$_2$ defenses in response to perception of *E. coli*, which protects the nematodes by depleting H$_2$O$_2$ from the nematode's environment (*Schiffer et al., 2020*), like most bacteria in the nematodes' ecological setting (*Schiffer et al., 2021*). The ASJ and I2 sensory neurons play a role in classical stress responses by triggering locomotory escape and feeding inhibition, respectively, in response to perception of H$_2$O$_2$ (*Bhatla and Horvitz, 2015*; *Schiffer et al., 2021*). We speculate that by relying on a combination of strategies, *C. elegans* nematodes can better manage the challenge of avoiding inducing costly H$_2$O$_2$ defenses that can cause undesirable side effects at inappropriate times.

# Materials and methods

## *C. elegans* culture, strains, and transgenes

Wild-type *C. elegans* were Bristol N2. *C. elegans* hermaphrodites were cultured at 20°C on NGM agar plates (Nematode Growth Medium, 17 g/l agar, 2.5 g/l Bacto Peptone, 3.0 g/l NaCl, 1 mM CaCl$_2$, 1 mM MgSO$_4$, 25 mM H$_2$KPO$_4$/HK$_2$PO4 pH 6.0, 5 mg/l cholesterol) seeded with *E. coli* OP50, unless noted otherwise. Double mutant worms were generated by standard genetic methods. The *Pgcy-8::ins-39(+)* (pFS1) plasmid was built via gene synthesis (Twist Bioscience) by inserting the *gcy-8* promoter (800 basepairs) followed by the *ins-39* open reading frame and 3′ untranslated region (600 basepairs) into a pTwist High Copy vector backbone. The plasmid was injected at 30 ng/µl into *ins-39(tm6467)* with 50 ng/µl *Punc-122::GFP* as co-injection marker. *ins-39(oy167[ins-39::SL2::GFP])* was made according to published protocols (*Ghanta et al., 2021*). Briefly, a dsDNA donor was made by amplifying SL2::GFP with 5′ SP9-modified oligos containing 35 bp overhangs homologous to the *ins-39* genomic locus for insertion immediately after the stop codon, using these primers: AGCAGGTC AAAGACGACTTCGTCACACTGCTCTGAGCTGTCTCATCCTACTTTCAC (forward) and ACTGGGCA AACGGAGAGTGAACGATGGAGCATTGACTATTTGTATAGTTCATCCATGCC (reverse). The genomic locus was cut with a crRNA targeting the sequence GATGGAGCATTGATCAGAGC. For a list of all bacterial and worm strains used in this study, see *Supplementary file 9* and *Supplementary file 10*, respectively. For a list of PCR genotyping primers and phenotypes used for strain construction, see *Supplementary file 11*.

## Survival assays

Automated survival assays were conducted using a *C. elegans* lifespan machine scanner cluster (*Stroustrup et al., 2013*) as described previously (*Servello and Apfeld, 2020*). This platform enables the acquisition of survival curves with very high temporal resolution and large population sizes. All chemicals were obtained from Sigma. For survival assays with 1 mM hydrogen peroxide and 6 mM tert-butyl hydroperoxide, the respective compound was added to molten agar immediately before pouring onto 50 mm NGM agar plates. Plates were dried (*Stroustrup et al., 2013*) and seeded with 100 µl of concentrated *E. coli* OP50 resuspended at an OD$_{600}$ of 20 (*Entchev et al., 2015*). For RNAi experiments, the appropriate *E. coli* HT115 (DE3) strain was used instead. For hydrogen peroxide assays, *E. coli* JI377 was used instead (*Seaver and Imlay, 2001*). Nematodes were cultured at the specified developmental temperature until the onset of adulthood, and then cultured at the specified adult temperature, in groups of up to 100, on plates with 10 µg/ml 5-fluoro-2′-deoxyuridine (FUDR), to avoid vulval rupture (*Leiser et al., 2016*) and eliminate live progeny. In a previous study (*Schiffer et al., 2020*), as an alternative to FUDR, we inhibited formation of the eggshell of fertilized *C. elegans* embryos with RNAi of *egg-5* (*Entchev et al., 2015*), with identical results in wild-type nematodes and *daf-7(−)* mutants, which increase peroxide resistance. Day 2 adults were transferred to lifespan machine assay plates. A typical experiment consisted of up to 4 genotypes or conditions, with 4 assay plates of each genotype or condition, each assay plate containing a maximum of 40 nematodes, and 16 assay plates housed in the same scanner. All experiments were repeated at least once, yielding the same results. Scanner temperature was calibrated to 20 or 25°C with a thermocouple (ThermoWorks USB-REF) on the bottom of an empty assay plate. Death times were automatically detected by the lifespan machine's image-analysis pipeline, with manual curation of each death time through visual inspection of all collected image data (*Stroustrup et al., 2013*), without knowledge of genotype or experimental condition.

## *E. faecium* survival assays

Nematodes were continuously fed *E. coli* JI377 for at least three generations and grown under the specified developmental temperature until the onset of adulthood, then cultivated at the specified adult temperature on plates with 10 µg/ml FUDR. *E. faecium* E007 was cultured in 500 ml of Brain Heart Infusion (BHI) medium overnight at 37°C without aeration and then aerated for 4 hr at 37°C prior to collecting the supernatant. The supernatant was stored at −20°C and incubated at the specified temperature before use. Nematodes were washed with M9 buffer with 0.01% Tween and transferred into 24-well plates containing either BHI or the appropriate amount of *E. faecium* E007 supernatant, *E. coli* JI377 at an OD$_{600}$ of 2, in a final volume of 2 ml. Plates were sealed with parafilm, incubated at the

specified temperature, and survival was scored after 16 hr by mixing the wells via pipette and scoring for movement using a dissection stereo microscope equipped with white-light transillumination.

## Transcriptomic analysis

mRNA for sequencing was extracted from day 2 adult animals. Worms were cultured on NGM agar plates seeded with *E. coli* OP50 and synchronized at the late L4 stage by transfer onto new NGM agar plates seeded with *E. coli* OP50 and supplemented with 10 µg/ml FUDR. Worms were cultured at 20°C except for the growth temperature assay for which worms were cultured at 25 or 20°C for four generations before sampling at the respective temperatures. We adapted a nematode lysis protocol (*Ly et al., 2015*) for bulk lysis to pool 30 individuals per sample in 120 µl of lysis buffer. cDNA preparation from mRNA was performed by SmartSeq2 as described (*Picelli et al., 2014*). cDNA was purified using an in-house paramagnetic bead-based DNA purification system mimicking Agencourt AMPure XP magnetic beads. Dual-barcoded Nextera sequencing libraries were prepared according to the manufacturer's protocol and purified twice with magnetic beads. Libraries were sequenced on an Illumina NextSeq 500 with a read length of 38 bases and approx. $2.0 \times 10^6$ paired-end reads per sample. RNA-seq reads were aligned to the *C. elegans* Wormbase reference genome (release WS265) using STAR version 2.6.0c (*Dobin et al., 2013*) and quantified using featureCounts version 2.0.0 (*Liao et al., 2014*), both using default settings. To quantify the expression within intervals in the genomic region encoding the three *C. elegans* catalase genes, we created a GTF that matches genomic positions defined previously (*Petriv and Rachubinski, 2004*). The reads count matrix was normalized using scran (*Lun et al., 2016*). Differential analysis was performed using a negative binomial generalized linear model as implemented by DESeq2 (*Love et al., 2014*). A batch replicate term was added to the regression equation to control for confounding. Batch-corrected counts were obtained by matching the quantiles of distributions of counts to the batch-free distributions as in the Combat-seq method (*Zhang et al., 2020*). Principal component analysis was performed on the batch-corrected and normalized log counts with a pseudo-count of one. To access the significance of the slope between the sum of the 'AFD(−) effect' and 'AFD(−) *daf-16(−)* interaction effect' coefficients and the 'AFD(−) effect' coefficient, we simulated 1000 sets of coefficients using a normal distribution with mean equal to the maximum-likelihood estimate of the coefficients and with standard deviation equal to the standard error of the estimates. We then fit a linear regression to each of the simulated coefficients and computed their coefficient of determination ($R^2$). These simulations showed that, after accounting for the level of uncertainty on our estimates of the values of the coefficients for 'AFD(−) effect' and 'AFD(−) *daf-16(−)* interaction effect', the average value of the regression's slope was 0.6826 (99% confidence interval [0.6825, 0.6827]) and the average $R^2$ was 0.699 (99% confidence interval [0.6989, 0.6991]). GO enrichments, tissue enrichment analysis, and phenotypic enrichment analysis were determined by using the WormBase Enrichment Suite (*Angeles-Albores et al., 2016*). We clustered and plotted GO terms with *q* value $<10^{-6}$ using REVIGO (*Supek et al., 2011*). Curated gene-expression datasets were obtained from WormExp (*Yang et al., 2016*). A curated hierarchical classification of genes into sets based on physiological function, molecular function, and cellular location was obtained from WormCat (*Higgins et al., 2022*; *Holdorf et al., 2020*).

## Microscopy

Nematodes carrying the *ins-39(oy167[ins-39::SL2::GFP])* allele were immobilized with 20 mM tetramisole, mounted on 10% agarose pads on slides, and imaged on a Zeiss Axio Imager M2 epifluorescent microscope with a ×63 objective. Exposure time was set to 300 ms and images were acquired with 2 × 2 binning. Detectable GFP expression was determined to be entirely restricted to AFD, in day 1 adults, by coexpression of the AFD-specific marker *ttx-1p::mCherry* (*Satterlee et al., 2001*). Images for GFP quantification were acquired with no red marker in the background. Images were processed in ImageJ, and expression was quantified from a maximum projected z-stack as corrected total cell fluorescence (CTCF) by the equation: CTCF = integrated density − (area of selected cell ROI × mean fluorescence of a nearby background ROI).

## RNA interference

*E. coli* HT115 (DE3) bacteria with plasmids expressing dsRNA targeting specific genes were obtained from the Ahringer and Vidal libraries (*Kamath et al., 2001*; *Rual et al., 2004*). Empty vector plasmid

pL4440 was used as control. Bacterial cultures were grown in LB broth with 100 μg/ml ampicillin at 37°C and seeded onto NGM agar plates containing 50 μg/ml carbenicillin and 2 mM IPTG. Nematodes were cultivated on *E. coli* OP50 until day 2 of adulthood, then transferred to RNAi plates. Their progeny was subsequently used for each assay.

## Kinetic modeling

The Eyring–Polanyi equation, derived from transition state theory, describes how chemical reactions depend on temperature (*Evans and Polanyi, 1935*; *Eyring, 1935*):

$$k = \frac{\kappa k_B T}{h} e^{\frac{\Delta S^{\ddagger}}{R}} e^{-\frac{\Delta H^{\ddagger}}{RT}} \tag{1}$$

where $k$ is the rate coefficient of the reaction, $T$ is the temperature, $\Delta H^{\ddagger}$ is the enthalpy of activation of the reaction, $\Delta S^{\ddagger}$ is the entropy of activation of the reaction, $R$ is the ideal gas constant, $k_B$ is the Boltzmann constant, $h$ is the Plank constant, and $\kappa$ is the transmission coefficient. The relative rate of a reaction at two different temperatures is the ratio of the reaction's rate coefficients at those temperatures:

$$\frac{k_1}{k_2} = \frac{T_1}{T_2} e^{-\frac{\Delta H^{\ddagger}}{R}\left(\frac{1}{T_1} - \frac{1}{T_2}\right)} \tag{2}$$

We obtained the $\Delta H^{\ddagger}$ values of specific reactions of $H_2O_2$ from published data: 46.6 kJ/mol for the Fenton reaction with DNA-bound Fe(II) at neutral pH in the presence of ATP (*Park et al., 2005*), 20 kJ/mol for the thiol oxidation of the highly reactive catalytic cysteine of alkyl hydroperoxide reductase E from *Mycobacterium tuberculosis* (*Zeida et al., 2014*), between 20 and 65 kJ/mol for the thiol oxidation of the reactive cysteine residue in (the non-hydroperoxidase) glyceraldehyde-3-phosphate dehydrogenase (GAPDH) (*Deponte, 2017*), above 100 kJ/mol for the thiol oxidation of regular cysteines in proteins (*Deponte, 2017*), and 68.5 kJ/mol for the thiol oxidation of free cysteine in aqueous solution (*Luo et al., 2005*).

## Statistical analysis

Statistical analyses were performed in JMP Pro version 15 (SAS). Survival curves were calculated using the Kaplan–Meier method. We used the log-rank test to determine if the survival functions of two or more groups were equal. We used analysis of variance (ANOVA) to determine whether the fold-change in gene expression of specific gene sets and of all genes were equal. We used ANOVA for GFP expression comparisons and, in cases where more than two groups were compared, used the Tukey HSD post hoc test to determine which pairs of groups in the sample differed. We used the Cell chi-square test to determine if a cell in a table differed from its expected value in the overall table. We used ordinal linear regression to determine whether the proportions of dead animals after treatment with *E. faecium* supernatant were equal across groups and to quantify interactions between groups using the following linear model: data = intercept + group 1 + group 2 + group 1 × group 2 + $\varepsilon$. The second to last term in this model quantifies the existence, magnitude, and type (synergistic or antagonistic) of interaction between groups. We used the Bonferroni correction to adjust p values when performing multiple comparisons.

## Materials availability

Further information and requests for resources and reagents should be directed to and will be fulfilled by the Lead Contact, Javier Apfeld (j.apfeld@northeastern.edu).

## Acknowledgements

We thank Erin Cram, Jodie Schiffer, and Yuyan Xu for detailed comments on our manuscript. Tobias Dansen, Marcel Deponte, James Imlay, and Christine Winterbourn for advice on hydrogen peroxide reaction rates. Joy Alcedo, Ryan Baugh, Danielle Garsin, and Yun Zhang kindly provided strains. Queelim Ch'ng and Joy Alcedo shared that *ins-39* is expressed in AFD. We benefitted from discussions with members of Javier Apfeld's and Erin Cram's labs. We derived some information from Wormbase, which is supported by the National Human Genome Research Institute at the NIH (grant #U41 HG002223), the UK Medical Research Council, and the UK Biotechnology and Biological Sciences

Research Council. Some strains were provided by the CGC, which is funded by NIH Office of Research Infrastructure Programs (P40 OD010440).

## Additional information

### Competing interests

Piali Sengupta: Senior editor, *eLife*. The other authors declare that no competing interests exist.

### Funding

| Funder | Grant reference number | Author |
|---|---|---|
| National Science Foundation | CAREER 1750065 | Javier Apfeld |
| Northeastern University | Tier 1 | Javier Apfeld |
| Generalitat de Catalunya | CERCA Programme | Nicholas Stroustrup |
| The Centro de Excelencia Severo Ochoa | CEX2020-001049-S | Nicholas Stroustrup |
| European Research Council | 852201 | Nicholas Stroustrup |
| National Institutes of Health | R35 GM122463 | Piali Sengupta |
| National Institutes of Health | F32 NS112453 | Nathan Harris |
| National Science Foundation | 1757443 | Nohelly Derosiers |
| Ministerio de Economía, Industria y Competitividad, Gobierno de España | Tio the EMBL partnership | Nicholas Stroustrup |
| Ministerio de Economía, Industria y Competitividad, Gobierno de España | MEIC Excelencia award PID2020-115189GB-I00 | Nicholas Stroustrup |
| The Centro de Excelencia Severo Ochoa | MCIN/AEI /10.13039/501100011033 | Nicholas Stroustrup |

The funders had no role in study design, data collection, and interpretation, or the decision to submit the work for publication.

### Author contributions

Francesco A Servello, Conceptualization, Data curation, Formal analysis, Supervision, Investigation, Visualization, Methodology, Writing – original draft; Rute Fernandes, Natasha Oswal, Anders Lindberg, Nohelly Derosiers, Investigation; Matthias Eder, Data curation, Formal analysis, Investigation, Methodology; Nathan Harris, Investigation, Methodology, Writing – review and editing; Olivier MF Martin, Data curation, Formal analysis, Investigation, Visualization, Methodology, Writing – review and editing; Piali Sengupta, Supervision, Funding acquisition, Writing – review and editing; Nicholas Stroustrup, Supervision, Funding acquisition, Project administration, Writing – review and editing; Javier Apfeld, Conceptualization, Data curation, Formal analysis, Supervision, Funding acquisition, Investigation, Visualization, Methodology, Writing – original draft, Project administration

### Author ORCIDs

Nathan Harris http://orcid.org/0000-0002-7856-520X
Natasha Oswal http://orcid.org/0000-0003-1478-8356
Piali Sengupta http://orcid.org/0000-0001-7468-0035
Nicholas Stroustrup http://orcid.org/0000-0001-9530-7301
Javier Apfeld http://orcid.org/0000-0001-9897-5671

Decision letter and Author response
Decision letter https://doi.org/10.7554/eLife.78941.sa1
Author response https://doi.org/10.7554/eLife.78941.sa2

## Additional files

### Supplementary files

• Supplementary file 1. Statistical analysis for *Figure 2* and *Figure 2—figure supplement 1*.

• Supplementary file 2. Statistical analysis for *Figure 3* and *Figure 3—figure supplement 1*.

• Supplementary file 3. Statistical analysis of gene-set expression for *Figures 6–8*; *Figure 4—figure supplements 2 and 4*, and 5; *Figure 6—figure supplement 1*; and *Figure 7—figure supplement 1*.

• Supplementary file 4. Gene Ontology analysis for *Figure 4*.

• Supplementary file 5. Statistical analysis for *Figure 5* and *Figure 5—figure supplement 1*.

• Supplementary file 6. Regulation of DAF-16 target genes among genes up- and down-regulated significantly (*q* value <0.01) by growth at 25°C and by AFD ablation at 20°C.

• Supplementary file 7. Statistical analysis for *Figure 6* and *Figure 6—figure supplement 1*.

• Supplementary file 8. Statistical analysis for *Figure 7*.

• Supplementary file 9. Bacterial strains.

• Supplementary file 10. Nematode strains.

• Supplementary file 11. PCR genotyping primers and phenotypes used for strain construction.

• MDAR checklist

### Data availability

Raw mRNA-seq read files are available under Bioproject PRJNA822361 (https://www.ncbi.nlm.nih.gov/bioproject/PRJNA822361). All data generated or analyzed during this study are included in the manuscript and supporting files.

The following dataset was generated:

| Author(s) | Year | Dataset title | Dataset URL | Database and Identifier |
|---|---|---|---|---|
| Apfeld J | 2022 | Neuronal temperature perception induces specific defenses that enable *C. elegans* to cope with the enhanced reactivity of hydrogen peroxide at high temperature | https://www.ncbi.nlm.nih.gov/bioproject/PRJNA822361 | NCBI BioProject, PRJNA822361 |

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
