## [Editor Report]

For any organism, tailoring defenses to the most pressing threats has high adaptive value – this paper makes the important finding that the nematode *C. elegans* pre-emptively augments its defenses against hydrogen peroxide when temperature increases, a condition that enhances the damage this compound causes. The authors describe this strategy as ‘enhancer sensing,’ whereby the perception of an environmental stimulus leads to the induction of defenses against a distinct (but mechanistically linked) threat. Convincing mechanistic studies reveal a role for a key thermosensory neuron and insulin-like signaling in this phenomenon. Because of its interdisciplinary outlook, this work will be of interest to readers in the fields of sensory neuroscience, stress physiology, and evolutionary biology.

---

## [Decision Letter]

**Decision letter after peer review:**

Thank you for submitting your article "Neuronal temperature perception induces specific defenses that enable *C. elegans* to cope with the enhanced reactivity of hydrogen peroxide at high temperature" for consideration by *eLife*. Your article has been reviewed by 3 peer reviewers, including Douglas Portman as Reviewing Editor and Reviewer #1, and the evaluation has been overseen by Claude Desplan as the Senior Editor.

Essential revisions:

1) Your model proposes but does not directly test the idea that the requirement for AFD in 25C-induced peroxide resistance reflects a role for thermosensation by AFD in this process. In your revision, please test this directly (for example, using gcy-18 gcy-8 gcy-23 triple mutants in which AFD thermosensation is abolished) or, at a minimum, determine whether AFD activity is required (for example, by silencing AFD or by rescuing tax-4 in AFD).

2) Your model does not directly address whether AFD is the relevant site of ins-39 function. In your revision, please test this directly by AFD-specific rescue and/or disruption of ins-39.

3) In several places in the paper, the reviewers feel that your paper oversimplifies the role of daf-16 in temperature-induced peroxide resistance. More detail about these concerns can be found in the individual reviews below. Please edit the manuscript to more clearly address AFD-independent roles of daf-16 as well as daf-16/skn-1-independent roles of AFD.

4) The reviewers feel that the "enhancer sensing" idea, while intriguing, is not supported strongly enough by the paper's results. Please carry out additional experiments (as suggested by reviewers 1 and 3) and/or tone down the writing to make it clear that this idea is speculative. Please also consider Reviewer 3's concern about potential confusion caused by the use of the term "enhancer sensing".

Each reviewer has additional comments and suggestions below. While we ask that you consider all of these, it is not necessary for your revision to include new experiments to address these points.

*Reviewer #1 (Recommendations for the authors):*

Here, Servello et al., explore the role of temperature and the temperature-sensing neuron AFD in promoting protection against peroxide damage. Unlike many other environmental threats, peroxide toxicity is expected to be temperature-dependent, since its chemical reactivity should be enhanced by higher temperatures. The authors convincingly and rigorously show that transient exposure to 25C, a condition of mild heat stress in *C. elegans*, activates animals' defenses against peroxides but potentially not other agents. Interestingly, this response requires the temperature-sensing AFD neurons, though whether temperature-dependent AFD activity is itself involved in this regulation is not explored. Further, the authors find that temperature regulates AFD's expression of the insulin ins-39 and provide evidence supporting the idea that repression of ins-39 at 25C contributes to enhanced peroxide defense. The authors use transcriptomic approaches to explore gene expression changes in animals in which AFD neurons are ablated, providing evidence that the FoxO-family transcription factor DAF-16 potentiates AFD signaling. However, because AFD ablation triggers effects broader than transient 25C exposure, the significance of these findings for temperature-dependent peroxide defense is somewhat unclear. Additionally, the possibility that DAF-16 (as well as another protective factor, SKN-1) function in parallel to temperature stress is consistent with many of the results shown but is not as thoroughly considered. Together, these studies identify a fascinating example of pre-emptive threat response triggered by the detection of a potentiator of that threat, a phenomenon they term "enhancer sensing." While some predictions of the specificity of this phenomenon remain untested, the paper provides intriguing insight into the potential mechanisms by which it may occur.

The dependence of the enhancer-sensing phenomenon on AFD leads the authors to conclude that the 25C stimulus is sensed by AFD itself, but this needs to be directly tested. To do this, they could ask whether tax-4 function is required in AFD, or use mutants in which AFD's thermosensory function is compromised.

The enhancer-sensing model is fascinating, but as it stands it is somewhat oversold. The authors could tone down the writing, indicating that this model is suggested rather than shown. Alternatively, they could more carefully test some of its predictions – for example by exploring the response to other threats (e.g. some of the toxicants described in Figure S5) at 20C and 25C in WT and AFD-ablated animals.

The role of ins-39 remains somewhat speculative. Figure 4F shows that ins-39 mutants have a reduced induction of peroxide defense, but it seems that this could be the result of a ceiling effect. The authors' model predicts that overexpression of ins-39, particularly at 25C, should sensitize animals to peroxide damage, a prediction that should be tested directly. Further, the authors seem to assume that AFD is the relevant site of ins-39 function, but this needs to be better supported.

Most of the daf-16 and skn-1 experiments are carried out in AFD-ablated animals, making the relevance of these findings for the 25C-dependent induction of peroxide defense somewhat unclear. As the authors show, AFD ablation causes much more extensive changes than transient 25C exposure, clearly seen in slope of the line in 3C. Further, unlike 25C exposure, AFD ablation is a chronic and non-physiological state. It would be useful for the authors to be cautious in their interpretation of these findings and to be clearer about how strongly they can connect them to the "enhancer sensing" phenomenon. Along these lines, the potentiation idea could be toned down a bit. Much of the data is consistent with parallel function for daf-16 (and skn-1) – for example, Figure 5C indicates additive effects of daf-16 and 25C exposure; 6C shows that AFD ablation still has a clear effect on peroxide sensitivity in the absence of both daf-16 and skn-1; and Figure S8a shows that much of the transcriptional response to AFD ablation (along PC1) is intact in daf-16 animals.

Based on theory or data, it would be useful for the authors to be more specific about the extent to which a 5-degree rise in temperature would be expected to enhance peroxide damage. The idea itself is solid, but whether the size of the effect is large enough to be biologically meaningful isn't addressed.

FUdR is used in all of the peroxide-sensitivity assays. Is there any reason to be concerned about this? It would be useful for the authors to comment on the reasons why they don't (or do?) expect disruption of germline proliferation to influence responses to peroxide/temperature/AFD ablation.

In a number of places, I think the writing could be toned down a bit. Some examples:

Line 124 – authors use the word multiple, but unless I've missed something, here this means two. "Multiple" is not inaccurate but is a bit misleading.

Line 154-55 – these are empirical observations, so the idea that the worm is changing its physiology "to prepare for" something is speculative.

Lines 181-2 – to me, these results *suggest* that neuronal sensory transduction by tax-4 channels is important.

Line 294 – because AFD ablation isn't a physiological state, I'd suggest avoiding the suggestion that defenses are "pre-induced"

One stylistic comment that the authors should feel free to ignore and do not need to address in their rebuttal: I find the repeated use of "the nematodes" in the text to be a little strange. Referring instead to "*C. elegans*" or "animals" would be a more standard approach.

*Reviewer #2 (Recommendations for the authors):*

In this study, Servello and the colleagues characterize how a temperature sensing neuron AFD regulates increased resistance to hydrogen peroxide in worms cultivated at a higher temperature. They show that loss of AFD and the insulin-like peptide INS-39 produced by AFD increase H2O2 resistance similarly as high temperature growth. To understand the molecular basis, they use mRNA-seq and analysis of gene expression at the whole-genome scale and transgenic lines to show that AFD ablation and high cultivation temperature generate overlapping changes in gene expression via the function of the FOXO transcription factor DAF-16 in the intestine.

This study is built on their previous work that established *C. elegans* as a model to study mechanisms for sensing and resistance of H2O2, an important environmental chemical threat for living organisms. Here, the authors uncover the neuronal and molecular basis for H2O2 resistance induced by high cultivation temperature. The authors use multiple approaches, including genetics, transgenics, whole-genome gene expression analysis, to characterize "enhancer sensing" that they discovered in this study. The experiments are well designed with appropriate controls. The data analysis is comprehensive and revealing. The findings are novel and explain a common and interesting phenomenon. The new understanding generated in this study will appeal to the readers in the fields of sensory biology, signaling transduction and physiology. The implications or conclusions of a few results presented here could be further discussed or clarified in the context of several previous studies.

1. My main question is about the link between AFD's response to higher temperature, its activity, and ins-39 expression. Previous studies show that increasing temperature from 20 to 25 degree activates AFD measured by intracellular calcium imaging. These results together with the findings in this paper would suggest that increased AFD activity reduces ins-39 expression. It will be helpful for the authors to discuss about these implications more clearly. In this paper, the authors seem to suggest that higher temperature at 25 inhibits AFD to reduce ins-39 expression. This may lead to the prediction that in the tax-4 mutant, in which AFD is not active in response to temperature, ins-39 expression is higher than wild type. This is different from the results in Figure 4E. It is possible that the effect of growing at 25 degree on AFD is different from acute sensing of the higher temperature. It will be much helpful for authors to discuss and clarify these points clearly.

2. Figure 1. Is H2O2 similarly stable at 20 degree and 25 degree?

3. Fig2H shows that AFD ablation has a stronger effect in increasing resistance than growing at 25c -- the additional effect caused by AFD ablation could use some discussion.

4. The rationale for the authors to focus on ins-39 needs to be better clarified, since multiple INSs are found in AFD.

5. Figure 4F. When growing at 25c, no further increase in resistance is seen in the ins-39 mutant compared to wild type, indicating a full effect; but at 20c, the ins-39 mutant does not fully mimic the resistance in worms grown at 25c. How would the authors explain this partial effect?

6. Figure 4G. CenGEN also identified the expression of ins-39 in several other neurons including ASK and ASJ. Therefore, the AFD-specific function of ins-39 in regulating H2O2 resistance should be further clarified.

7. Figure 3G. The authors suggest that increased ctl-1 and ctl-2 expression in AFD(-) worms confers increased resistance, because overexpression of these genes increases H2O2 resistance as shown in previous studies. This analysis of ctl genes provide evidence for the mechanistic basis for increased H2O2 resistance in AFD(-) worms. Is the increased expression of ctl-1 and ctl-2 in AFD(-) worms comparable to those generated by overexpression (ie does it confer higher resistance)? Some level of support or clarification will be very helpful.

8. Figure 7. It is much easier to read if most of the analysis in FigS8 is included in Fig7.

*Reviewer #3 (Recommendations for the authors):*

This paper offers novel mechanistic insights into how pre-exposure to warm temperature increases the resistance of *C. elegans* to peroxides, which are more toxic at warmer temperature. The temperature range tested in this study lies within the animal's living conditions and is much lower than that of heat shock. Therefore, this study expands our understanding of how past thermosensory experience shapes physiological fitness under chemical stress. The paper is technically sound with most experiments or analyses carried out rigorously, and therefore the conclusions are solid. However, it challenges our current understanding of the role of the *C. elegans* thermosensory system in coping with stress. The traditional view is that the AFD thermosensory neuron is activated upon sensing temperature rise, and that temperature sensation through AFD positively regulates systemic heat shock response and promotes longevity in *C. elegans*. Thus, it is quite unexpected that AFD ablation activates DAF-16 and improves peroxide resistance. It also appears counterintuitive that genes upregulated at 25 degrees overlap extensively with those upregulated by AFD ablation at 20 degrees. I feel that it is premature to coin the term "enhancer sensing" for such a phenomenon, as their work does not rule out the possibility that AFD ablation increases resistance to other stresses that are independent of temperature regarding their toxicity or magnitude of hazard. Additional work is necessary to clarify these issues.

1. Whether the role of AFD in inhibiting peroxide resistance is related to AFD activity needs further clarification. AFD activity depends on the animal's thermosensory experience. As animals in this study are maintained at 20 degrees unless indicated specifically, the AFD displays activities starting around 17 degrees and peaks around 20 degrees. Under such condition, the AFD displays little or no activity to thermal stimuli around 15 degrees. It will be important to test whether cultivation of animals at 20 degrees improves peroxide resistance at 15 degrees, compared to 15 degrees-cultivation/15 degrees peroxide testing. The authors should also test whether AFD ablation further improves survival under peroxides at 15 degrees for animals grown at 20 degrees, whose AFD should show little or no activities at 15 degrees.

2. The importance of the thermosensory function of AFD should be verified. In the current study, the tax-4 mutation was used to infer AFD activity, but tax-4 is expressed in sensory neurons other than AFD. In addition to AFD, AWC can sense temperature and it also expresses tax-4. Therefore, influence on AFD from other tax-4-expressing neurons cannot be excluded. On the other hand, ablation of AFD removes all AFD functions, including those that are constitutive and temperature-independent. Therefore, the authors should test the gcy-18 gcy-8 gcy-23 triple mutant, in which the AFD neurons are fully differentiated but completely insensitive to thermal stimuli. These three thermosensor genes are exclusively expressed in AFD. Compared to the tax-4 mutant that is broadly defective in multiple sensory modalities, this triple gcy mutant shows defects specifically in thermosensation. They should see whether results obtained from the AFD ablated animals could be reproduced by experiments using the gcy-18 gcy-8 gcy-23 triple mutant. The authors are also recommended to investigate ins-39 expression in AFD and profile gene expression patterns in the gcy-18 gcy-8 gcy-23 triple mutant.

3. The literature suggests that AFD promotes longevity likely in part through daf-16 (Chen at al., 2016) or independent of daf-16 (Lee and Kenyon, 2009). Whatever it is, various studies show that activation of AFD and daf-16 promote a normal lifespan at higher temperature, and AFD ablation shortens lifespan at either 20 or 25 degrees. Therefore, the finding that DAF-16-upregulated genes overlap extensively with those upregulated by AFD ablation is quite unexpected (Figure 5B). The authors should perform further gene ontology (GO) analysis to identify subsets of genes co-regulated by DAF-16 and AFD ablation, whether these genes are reported to be involved in longevity regulation, immunity, stress response, etc.

4. I feel that "enhancer sensing" is an overstatement, or at least a premature term that is not sufficiently supported without further investigations. The authors should explore whether AFD ablation or pre-exposure to warm temperature specifically enhances resistance to a stressor the toxicity of which is increased at higher temperature, but does not affect the resistance to other temperature-insensitive threats.

5. The authors need to provide data of all survival assays as supplemental tables, and indicate the sets of data that are selected to be presented in the figures. This is standard practice for *C. elegans* lifespan or survival assays and is important for the readers to understand the reproducibility of the experiments. This applies to Figures 1A, 1C, 1D, 2B-E, 2H, 4F, 5C-F, 6B, 6C, S1A, S1B, S2, S6B and S7C.

6. Page 17, Line 437-438: Loss of daf-16 in AFD-intact animals results in changes in the expression of merely two genes, which is quite surprising given the critical role of daf-16 in lifespan and stress resistance. Although the authors explained this in Discussion saying that fold changes in most genes were mild in the daf-16(mu86) mutant, it might be helpful to point out that in previous studies, many daf-16-dependent genes were characterized by comparing gene expressions in the daf-2 and daf-16; daf-2 double mutants. Given the critical importance of daf-16 in fitness, stress resistance and longevity, this apparent discrepancy with the existing literature must be explained in sufficient detail.

7. DAF-16 and SKN-1 nuclear localization should be investigated at warm temperature, in the absence of AFD and in the ins-39 mutant.

8. Figure 3G: The fold change in the expression levels of ctl-1, ctl-2 and ctl-3 at 25 degrees (v.s. 20 degrees) and in AFD-ablated animals should be confirmed by RT-qPCR. The data from RNA-seq in their current form do not allow for statistical analysis to determine whether the expression of these ctl genes are significantly upregulated by high cultivation temperature or AFD ablation.

9. It would be interesting to characterize gene expression changes in the ins-39 mutant and make comparison to those induced by temperature shift and AFD ablation. However, I understand that this is a substantial amount of work and it is OK if the authors decide not to do it at this moment.

10. Please provide statistics in all the Figure panels instead of using designations such as "a, b, c".

11. For esthetical reasons, the authors should remove the black portions in Figure 4B that were used to rectify the borders of the rotated fluorescent images. However, I think there is still room for careful cropping of the fluorescent images to make them rectangular without having to add another layer of black background.

12. Page 13, Line 327: please describe the molecular lesion of ins-39(tm6467).

13. Please indicate sample sizes for experiments in Figure 4C and 4D.

14. Remove the "A" label for Figure S2.

[Editors’ note: further revisions were suggested prior to acceptance, as described below.]

Thank you for resubmitting your work entitled "Neuronal temperature perception induces specific defenses that enable *C. elegans* to cope with the enhanced reactivity of hydrogen peroxide at high temperature" for further consideration by *eLife*. Your revised article has been evaluated by Claude Desplan (Senior Editor) and a Reviewing Editor.

All of the reviewers feel that your manuscript has been substantially improved. However, in their discussion, the reviewers continued to express concerns about the validity of the "enhancer sensing" model. Before publication, we ask that you revise the Abstract and Discussion to present a more balanced view. Any other changes that you may wish to make in response to the comments below are optional.

*Reviewer #1 (Recommendations for the authors):*

In their revised manuscript, Servello et al., have nicely addressed most of the concerns I raised earlier. I find the manuscript to be significantly improved. Before the paper is ready for publication, I think it would be worthwhile for the authors to consider addressing the following issues by editing the text.

– The demonstration that the gcy triple mutant has a peroxide-resistant phenotype is nice, but it would be useful for the authors to speculate about why the phenotype of this mutant is so much weaker than ADF ablation. The phenotypes of the single gcy mutants are also a little unexpected; this should be noted somewhere.

– I continue to feel that the "enhancer sensing" model is pushed too hard. In particular, I find the argument in the Discussion, lines 657 to 688, to be unconvincing. Adaptive sensing provides a mechanism by which species can "learn" to associate one piece of information with another, allowing detection of a particular environmental signal to become predictively coupled to an apparently distinct response. Here, the authors have shown (quite nicely) that *C. elegans* has "learned" that an increase in temperature predicts an increased susceptibility to peroxide damage. To me, this fits quite well with the adaptive sensing paradigm. I'm ok with the idea that what they are seeing could be considered a variant form of adaptive sensing, but it seems to me to be excessive to decide that a new term is needed for what amounts to essentially the same principle. In a way, even that is overkill: because the hazard is the thermodynamic effect of temperature itself, one could argue that increasing peroxide defense *is* an aspect of temperature defense, in the same way that increasing expression of protein chaperones is. As the authors note, it's well demonstrated that heat detection by the *C. elegans* nervous system can trigger the systemic activation of the heat shock response. The activation of peroxide defenses seems to me to be, in principle, an equivalent phenomenon.

– The authors write in their rebuttal that it is incorrect to predict that the response to non-H2O2 stresses should not be enhanced by 25C pre-exposure. In principle, I agree – one can't necessarily predict that augmenting defenses against these stressors wouldn't also be adaptive at 25C – but isn't the point of lines 340-342 to indicate that pre-exposure to 25C does indeed have some selectivity in terms of the stress responses it enhances? If so, then there seems to be a strong prediction that animals' responses to acrylamide, formaldehyde, etc, would not be changed by 25C pre-exposure.

*Reviewer #3 (Recommendations for the authors):*

The authors addressed most of the points that I raised in the initial review of the study. Although I am still not completely convinced by the idea of enhancer sensing and some of the interpretations of the role of AFD on gene expression and physiology, in particular its interaction with daf-16, I think those issues are better addressed in a separate study. Regarding "enhancer censing," for such a new term to be coined, multiple examples are necessary. Therefore, I suggest that the authors tone down this claim in the Discussion/Abstract, and also discuss the potential limitations of their study in justifying such a claim. In short, a more balanced view is needed.

---

## [Author Response]

Essential revisions:1) Your model proposes but does not directly test the idea that the requirement for AFD in 25C-induced peroxide resistance reflects a role for thermosensation by AFD in this process. In your revision, please test this directly (for example, using gcy-18 gcy-8 gcy-23 triple mutants in which AFD thermosensation is abolished) or, at a minimum, determine whether AFD activity is required (for example, by silencing AFD or by rescuing tax-4 in AFD).

As requested, we determined whether previously identified mechanisms for temperature perception by the AFD neurons were required for the temperature-dependent regulation of peroxide resistance using *gcy-18 gcy-8 gcy-23* triple mutants and the respective single mutants. The findings from the new experiments lead us to conclude that temperature perception by AFD via GCY-8, GCY-18, and GCY-23 enables *C. elegans* to lower their peroxide resistance at the lower cultivation temperature.

2) Your model does not directly address whether AFD is the relevant site of ins-39 function. In your revision, please test this directly by AFD-specific rescue and/or disruption of ins-39.

As requested, we determined whether *ins-39* gene expression in the AFD neurons was sufficient to lower peroxide resistance by restoring *ins-39(+)* gene expression only in the AFD neurons using the AFD-specific *gcy-8* promoter. The findings from this experiment lead us to conclude that expression of *ins-39* in the AFD neurons was sufficient to regulate *C. elegans* peroxide resistance.

3) In several places in the paper, the reviewers feel that your paper oversimplifies the role of daf-16 in temperature-induced peroxide resistance. More detail about these concerns can be found in the individual reviews below. Please edit the manuscript to more clearly address AFD-independent roles of daf-16 as well as daf-16/skn-1-independent roles of AFD.

We have edited the manuscript to more clearly address the roles of DAF-16 in the response of peroxide resistance to cultivation temperature, and the roles of DAF-16 and SKN-1 when the AFD neurons are present and when they are ablated, as detailed in our responses to reviewer #1’s point 4 and to reviewer #3’s point 3 and additional issue 2.

4) The reviewers feel that the "enhancer sensing" idea, while intriguing, is not supported strongly enough by the paper's results. Please carry out additional experiments (as suggested by reviewers 1 and 3) and/or tone down the writing to make it clear that this idea is speculative. Please also consider Reviewer 3's concern about potential confusion caused by the use of the term "enhancer sensing".

To address these concerns, we edited the manuscript and expanded the manuscript’s discussion, as detailed in our responses to reviewer #1’s point 2 and to reviewer #3’s point 4. The revised manuscript now makes a clear distinction between (a) enhancer sensing as a strategy for the regulation of defense responses (a conceptual advance we introduce in this paper) and (b) the specific instance of enhancer sensing we found is used by *C. elegans* to couple the induction of H2O2 defenses to the perception of temperature. Additionally, we now (c) discuss the evolutionary contexts that may favor or constrain the evolution of enhancer sensing strategies coupling temperature perception to the induction of defenses towards reactive chemicals other than hydrogen peroxide.

Reviewer #1 (Recommendations for the authors):Here, Servello et al., explore the role of temperature and the temperature-sensing neuron AFD in promoting protection against peroxide damage. Unlike many other environmental threats, peroxide toxicity is expected to be temperature-dependent, since its chemical reactivity should be enhanced by higher temperatures. The authors convincingly and rigorously show that transient exposure to 25C, a condition of mild heat stress in *C. elegans*, activates animals' defenses against peroxides but potentially not other agents. Interestingly, this response requires the temperature-sensing AFD neurons, though whether temperature-dependent AFD activity is itself involved in this regulation is not explored. Further, the authors find that temperature regulates AFD's expression of the insulin ins-39 and provide evidence supporting the idea that repression of ins-39 at 25C contributes to enhanced peroxide defense. The authors use transcriptomic approaches to explore gene expression changes in animals in which AFD neurons are ablated, providing evidence that the FoxO-family transcription factor DAF-16 potentiates AFD signaling. However, because AFD ablation triggers effects broader than transient 25C exposure, the significance of these findings for temperature-dependent peroxide defense is somewhat unclear. Additionally, the possibility that DAF-16 (as well as another protective factor, SKN-1) function in parallel to temperature stress is consistent with many of the results shown but is not as thoroughly considered. Together, these studies identify a fascinating example of pre-emptive threat response triggered by the detection of a potentiator of that threat, a phenomenon they term "enhancer sensing." While some predictions of the specificity of this phenomenon remain untested, the paper provides intriguing insight into the potential mechanisms by which it may occur.The dependence of the enhancer-sensing phenomenon on AFD leads the authors to conclude that the 25C stimulus is sensed by AFD itself, but this needs to be directly tested. To do this, they could ask whether tax-4 function is required in AFD, or use mutants in which AFD's thermosensory function is compromised.

We thank the reviewer for suggesting these experiments. As requested, we determined whether previously identified mechanisms for temperature perception by the AFD neurons were required for the temperature-dependent regulation of peroxide resistance using *gcy-18 gcy-8 gcy-23* triple mutants and the respective single mutants. The findings from the new experiments lead us to conclude that temperature perception by AFD via the GCY-8, GCY-18, and GCY-23 receptor guanylate cyclases, which are exclusively expressed in the AFD neurons, contributes to the temperature-dependent regulation of peroxide resistance in *C. elegans*. These experiments are detailed in the following new paragraph in the Results section:

“Last, we determined whether previously identified mechanisms for temperature perception by the AFD neurons were required for the temperature-dependent regulation of peroxide resistance. The AFD neurons sense temperature using receptor guanylate cyclases, which catalyze cGMP production, leading to the opening of TAX-4 channels (Goodman and Sengupta, 2019). Three receptor guanylate cyclases are expressed exclusively in AFD neurons: GCY-8, GCY-18, and GCY-23 (Inada et al., 2006; Yu et al., 1997) and are thought to act as temperature sensors (Takeishi et al., 2016). Triple mutants lacking *gcy-8, gcy-18*, and *gcy-23* function are behaviorally atactic on thermal gradients and fail to display changes in intracellular calcium or thermoreceptor current in the AFD neurons in response to temperature changes (Inada *et al.*, 2006; Ramot et al., 2008; Takeishi *et al.*, 2016; Wang et al., 2013; Wasserman et al., 2011). We found that when grown and assayed at 20°C, *gcy-23(oy150) gcy-8(oy44) gcy-18(nj38)* triple null mutants survived 43% longer in the presence of tBuOOH than wild-type controls (Figure 3J). In contrast, at 25°C, the *gcy-23 gcy-8 gcy-18* triple mutants showed a 12% decrease in peroxide resistance relative to wild-type controls (Figure 3K). Therefore, the three AFD-specific receptor guanylate cyclases influenced the temperature dependence of peroxide resistance, lowering peroxide resistance at 20°C and slightly increasing it at 25°C. At 20°C, the *gcy-8(oy44)*, *gcy-18(nj38)*, and *gcy-23(oy150)* single mutants increased peroxide resistance by 10%, 51%, and 21%, respectively, relative to wild-type controls (Figure 3L). Therefore, each of the three AFD-specific receptor guanylate cyclases regulates peroxide resistance. We conclude that temperature perception by AFD via GCY-8, GCY-18, and GCY-23 enables *C. elegans* to lower their peroxide resistance at the lower cultivation temperature.”

The enhancer-sensing model is fascinating, but as it stands it is somewhat oversold. The authors could tone down the writing, indicating that this model is suggested rather than shown. Alternatively, they could more carefully test some of its predictions – for example by exploring the response to other threats (e.g. some of the toxicants described in Figure S5) at 20C and 25C in WT and AFD-ablated animals.

We edited the manuscript and expanded the manuscript’s discussion to address these concerns as well as similar concerns from reviewer #3. In the paper we show that the regulation of the induction of H2O2 defenses in *C. elegans* is coupled to the perception of temperature (an inherent enhancer of the reactivity of H2O2). To understand the significance of this finding in an evolutionary context, and to explain why such a regulatory system would evolve, we introduced in the discussion a new conceptual framework, “enhancer sensing,” and devoted a section of the discussion to demonstrating that the phenomenon that we observed could not be adequately explained by existing frameworks used to understand the evolutionary origins of the regulatory systems for defense responses.

We now realize that we did not sufficiently and clearly explain the scope for the criterion for establishing a phenomenon represents enhancer sensing, leading to incorrect predictions by reviewer’s 1 and 3 about (a) whether what we observed in *C. elegans* is an instance of enhancer sensing (or more proof is needed) and (b) what the enhancer sensing model for the coupling of temperature perception to H2O2 defense would predict about how temperature and the AFD neurons would affect resilience to other chemicals. We regret failing to adequately explain the model’s scope and predictions and believe that we have now explicitly addressed the scope of what constitutes enhancer sensing and the predictions of the model. In particular, we previously did not spell out (a) the distinction between the enhancer sensing strategy and the mechanistic implementation of that strategy; and, importantly, (b) we did not discuss what the enhancer sensing strategy coupling temperature perception to H2O2 defense in *C. elegans* predicted (and did not predict) about whether a similar strategy would be expected to be used by *C. elegans* to deal with other temperature-dependent threats. We now address these issues in two new paragraphs in the discussion that read:

“We show here that *C. elegans* uses an enhancer sensing strategy that couples H2O2 defense to the perception of high temperature. We expect this strategy’s output (the level of H2O2 defense) to provide the nematodes with an evolutionarily optimal strategy across ecologically relevant inputs (cultivation temperatures) (Kussell and Leibler, 2005; Maynard Smith, 1982; Wolf et al., 2005). This strategy is implemented at the organismic level through the division of labor between the AFD neurons, which sense and broadcast temperature information, and the intestine, which responds to that information by providing H2O2 defense (Figure 9D). Ascertaining that *C. elegans* relies on this enhancer sensing strategy does not depend on the temperature information broadcast by AFD exclusively regulating defense responses to temperature-dependent threats, because the regulation of defenses towards temperature-insensitive threats could affect defenses towards temperature-dependent threats; for example, suppressing defenses towards a temperature-insensitive threat would be beneficial if those defenses interfered with H2O2 defense or depleted energy resources contributing to H2O2 defense.

As with any sensing strategy, enhancer sensing strategies are more likely to evolve when sensing is informative and responding is beneficial. In their natural habitat, *C. elegans* encounter many environmental chemicals that, like H2O2, are inherently more reactive at higher temperatures. It will be interesting to determine the extent to which *C. elegans* uses enhancer sensing strategies coupling temperature perception to the induction of defenses towards those chemicals, and whether those strategies rely on temperature perception and broadcasting by the AFD neurons. We expect that sensing strategies regulating defense towards those chemicals would be more likely to evolve when those chemicals are common, reactive, and cause consequential damage.”

We note that our ability to predict survival to other toxicants, such as those that trigger specific gene-expression responses that are AFD-dependent but are unaffected between 20C and 25C (as proposed by the reviewer), is limited not only by our lack of knowledge about the specific mechanisms that protect worms from those toxicants, but also by our lack of knowledge about whether defense towards hydrogen peroxide interferes (or synergizes) with defense towards each of those toxicants and whether defense towards those toxicants interferes (or synergizes) with H2O2 defense. We therefore think that those experiments would be better addressed in future studies.

The role of ins-39 remains somewhat speculative. Figure 4F shows that ins-39 mutants have a reduced induction of peroxide defense, but it seems that this could be the result of a ceiling effect. The authors' model predicts that overexpression of ins-39, particularly at 25C, should sensitize animals to peroxide damage, a prediction that should be tested directly. Further, the authors seem to assume that AFD is the relevant site of ins-39 function, but this needs to be better supported.

As requested by all three reviewers, we determined whether *ins-39* gene expression in AFD was sufficient to lower peroxide resistance by restoring *ins-39(+)* gene expression only in the AFD neurons using the AFD-specific *gcy-8* promoter. As predicted by the reviewer, these worms were more sensitive to peroxide than wild-type worms. The findings from this experiment lead us to conclude that expression of *ins-39* in the AFD neurons was sufficient to regulate the nematode’s peroxide resistance. The new section reads:

“Next, we determined whether the INS-39 signal from AFD regulated the nematode’s peroxide resistance. The *tm6467* null mutation in *ins-39* deletes 520 bases, removing almost all the *ins-39* coding sequence (Figure 5A), and inserts in that location 142-bases identical to an intervening sequence located between *ins-39* and its adjacent gene. In nematodes grown and assayed at 20°C, *ins-39(tm6467)* increased peroxide resistance by 26% relative to wild-type controls (Figure 5F). To determine whether *ins-39* gene expression in AFD was sufficient to lower peroxide resistance, we restored *ins-39(+)* expression only in the AFD neurons using the AFD-specific *gcy-8* promoter (Inada *et al.*, 2006; Yu *et al.*, 1997) in *ins-39(tm6467)* mutants. Expression of *ins-39(+)* only in AFD eliminated the increase in peroxide resistance of *ins-39(tm6467)* mutants (Figure 5F). Notably, the peroxide resistance of the two independent transgenic lines was 28% and 30% lower than that of wild-type controls, likely due to overexpression of the gene beyond wild-type levels. We conclude that the gene dose-dependent expression of *ins-39* in the AFD neurons regulated the nematode’s peroxide resistance.”

The temperature-shift experiments in figure 5G (formerly 4F) indicated that the effect on peroxide resistance at 20C of growth at 25C and of the *ins-39* mutation were non additive. We interpreted this epistatic interaction to be due to action in a common pathway. It is possible that while growth at 25C increases the subsequent peroxide resistance at 20C, it could limit the nematodes’ subsequent peroxide resistance at 20C (beyond those peroxide-resistance increasing effects) when in combination with another intervention, even if those interventions acted via parallel mechanisms—a ceiling effect, as proposed by the reviewer. We favor the alternative interpretation, that the mechanisms act sequentially, because of our findings that *ins-39* gene expression within AFD was lower at 25C than at 20C, leading us to propose the sequential model in figure 5H (formerly 4G).

Most of the daf-16 and skn-1 experiments are carried out in AFD-ablated animals, making the relevance of these findings for the 25C-dependent induction of peroxide defense somewhat unclear. As the authors show, AFD ablation causes much more extensive changes than transient 25C exposure, clearly seen in slope of the line in 3C. Further, unlike 25C exposure, AFD ablation is a chronic and non-physiological state. It would be useful for the authors to be cautious in their interpretation of these findings and to be clearer about how strongly they can connect them to the "enhancer sensing" phenomenon. Along these lines, the potentiation idea could be toned down a bit. Much of the data is consistent with parallel function for daf-16 (and skn-1) – for example, Figure 5C indicates additive effects of daf-16 and 25C exposure; 6C shows that AFD ablation still has a clear effect on peroxide sensitivity in the absence of both daf-16 and skn-1; and Figure S8a shows that much of the transcriptional response to AFD ablation (along PC1) is intact in daf-16 animals.

We have made several adjustments in the text to address these concerns. As the reviewer noted, the experiments with *skn-1* were performed only in AFD ablated worms. We have renamed the section heading to “SKN-1/NRF and DAF-16/FOXO collaborate to increase the nematodes’ peroxide resistance in response to AFD ablation” to make that clear.

In contrast, the peroxide resistance experiments with *daf-16* were done also in worms grown at 25C and then shifted to 20C during the peroxide resistance assay. The connection of *daf-16* with the temperature dependent regulation of peroxide resistance was established in temperature shifts experiments in *daf-16* single mutants (Figure 6C, formerly 5C) and in transgenic worms rescuing the *daf-16* mutant only in the intestine (Figure 6F). In the revised text we make it clearer that the effect of the *daf-16* mutation is bigger when the nematodes are shifted from 25C to 20C: “The *daf-16(mu86)* null mutation decreased peroxide resistance in nematodes grown at 25°C and assayed at 20°C by 35%, a greater extent than the 21% reduction in peroxide resistance induced by that mutation in nematodes grown and assayed at 20°C (Figure 6C).”

As the reviewer noted, *daf-16* and *skn-1* have a role in peroxide resistance when the AFD neurons are not ablated (albeit a smaller one than when those neurons are ablated). We have made several changes and additions to the text to make that explicit. Most notably, the revised last paragraph of the SKN-1 section now reads: “We propose that when nematodes are cultured at 20°C, the AFD neurons promote signaling by the DAF-2/insulin/IGF1 receptor in target tissues, which subsequently lowers the nematode’s peroxide resistance by repressing transcriptional activation by SKN-1/NRF and DAF-16/FOXO. However, this repression is not complete, because both *daf-16(mu86)* and *skn-1(RNAi)* lowered peroxide resistance at 20°C when the AFD neurons were present. It is also likely that DAF-16 and SKN-1 are not the only factors that contribute to peroxide resistance in AFD-ablated nematodes at 20°C, because AFD ablation increased peroxide resistance in *daf-16(mu86); skn-1(RNAi)* nematodes, albeit to a lesser extent than in *daf-16(+)* or *skn-1(+)* backgrounds.”

The potentiation idea was specific to the effects of DAF-16 on gene expression. As the reviewer noted, much of the transcriptional response to AFD ablation is intact (albeit reduced in magnitude) in AFD-ablated *daf-16* mutants, leading to a shift in the PC1 score for the mutant. At the level of the expression of individual genes, we quantified those effects in Figure 8G (formerly 7D). When we did the RNAseq experiments we had expected that lack of *daf-16* would eliminate either all the changes in gene expression induced by AFD ablation or eliminate those changes for a subset of genes. Instead, what we found was much more subtle, and unexpected: the size of the gene expression change induced by AFD ablation was reduced by the *daf-16* mutation, and that reduction was systematic. Specifically, we found that the bigger the change in gene expression induced by AFD ablation, the bigger the effect of *daf-16* in the AFD ablated animals (that is, potentiation), leading to a change in the slope in the regression line in Figure 8G. We revised the paper to ensure we only used the word potentiation in this context (gene expression), even though formally DAF-16 also potentiated the effects of AFD ablation (and temperature shift from 25C to 20C) on peroxide resistance.

Based on theory or data, it would be useful for the authors to be more specific about the extent to which a 5-degree rise in temperature would be expected to enhance peroxide damage. The idea itself is solid, but whether the size of the effect is large enough to be biologically meaningful isn't addressed.

We are grateful to the reviewer for this request. When we wrote the paper, we only had an intuition, based on our knowledge of chemical kinetics, that temperature would increase the rates of the reactions of H2O2. To address the reviewer’s request, we have done mathematical modeling to (a) formalize that intuition, and (b) predict how much faster would H2O2 react with biologically relevant molecules. This has, in our opinion, made the paper much better. This new analysis is detailed in a new figure (Figure 1), a respective methods section, and the following text in the Results section:

“Previous studies showed that H2O2 kills *C. elegans* in a dose-dependent manner at environmental concentrations above 0.1 mM (Bolm *et al.*, 2004; Jansen *et al.*, 2002; Moy *et al.*, 2004). We expected that higher temperatures would make the same concentration of H2O2 more lethal to *C. elegans*, because the reaction rates of the chemical reactions of H2O2 increase exponentially with temperature (Arrhenius, 1889; Evans and Polanyi, 1935; Eyring, 1935). The exact molecular mechanisms by which H2O2 kills *C. elegans*, or any organism, remain unknown but are thought to involve the reactions of H2O2 with biologically important molecules, including proteins and DNA (Khademian and Imlay, 2021). Using chemical kinetics, we modeled how an increase in temperature from 20°C to 25°C would affect the rates of the chemical reactions of H2O2 with those biomolecules (Figure 1). Because these rate differences depend on the enthalpy of activation of the specific chemical reaction, they can vary widely between reactions. The Fenton reaction of H2O2 with DNA-bound Fe(II), which leads to DNA damage, was predicted to be 40% faster at 25°C than at 20°C (Figure 1). For the oxidation of the thiol groups of cysteines, reaction rates with H2O2 were predicted to be more than two-fold faster for regular cysteines in proteins, 62% faster for free cysteines, up to 56% faster for very reactive cysteines such as the redox-sensitive cysteine residue of GAPDH, and 17% faster for the most reactive cysteines of hydroperoxidases (Figure 1). These predicted increases in H2O2's reactivity towards specific biomolecules at 25°C, compared to 20°C, are similar to the ones that would occur at 20°C if H2O2 concentration were increased substantially—from 17% to more than 100%, depending on the specific reaction.”

FUdR is used in all of the peroxide-sensitivity assays. Is there any reason to be concerned about this? It would be useful for the authors to comment on the reasons why they don't (or do?) expect disruption of germline proliferation to influence responses to peroxide/temperature/AFD ablation.

Because our assays are automated, we cannot manually remove progeny during the assay. We used FUDR to prevent the production of viable progeny that could prevent us from measuring only the survival of their parents. We have previously used other methods that prevent progeny production in these assays, with results identical to those with FUDR. We have added a note right after the FUDR methods section to describe and reference those findings: “In a previous study (Schiffer *et al.*, 2020), as an alternative to FUDR, we inhibited formation of the eggshell of fertilized *C. elegans* embryos with RNAi of *egg-5* (Entchev et al., 2015), with identical results in wild-type nematodes and *daf-7* mutants, which increase peroxide resistance.”

In a number of places, I think the writing could be toned down a bit. Some examples:

We thank the reviewer for this comment. We have addressed these issues as noted below.

Line 124 – authors use the word multiple, but unless I've missed something, here this means two. "Multiple" is not inaccurate but is a bit misleading.

We replaced “multiple” with “two”.

Line 154-55 – these are empirical observations, so the idea that the worm is changing its physiology "to prepare for" something is speculative.

We made it explicit that we were speculating. The revised sentence reads: “Based on these findings, we speculated that *C. elegans* nematodes induced their peroxide defenses when grown at the higher temperature to prepare for the increased lethal threat posed by peroxides at high temperature (Figure 2F).”

Lines 181-2 – to me, these results suggest that neuronal sensory transduction by tax-4 channels is important.

We replaced “Therefore …” with “These findings suggested that neuronal sensory transduction by TAX-4 channels normally lowers the nematodes’ peroxide resistance to a lesser extent at high cultivation temperature.

Line 294 – because AFD ablation isn't a physiological state, I'd suggest avoiding the suggestion that defenses are "pre-induced"

We have re-written the sentence to explain more clearly what we meant. The revised sentence now reads:

“Therefore, ablation of the AFD sensory neurons induced genes normally induced by a wide variety of stressors in nematodes that were not exposed to those stressors, but the higher cultivation temperature only pre-induced a specific subset of genes that included hydrogen peroxide defenses and genes induced by peroxides.”

One stylistic comment that the authors should feel free to ignore and do not need to address in their rebuttal: I find the repeated use of "the nematodes" in the text to be a little strange. Referring instead to "*C. elegans*" or "animals" would be a more standard approach.

We changed many instances of “nematodes” to “*C. elegans*”.

Reviewer #2 (Recommendations for the authors):In this study, Servello and the colleagues characterize how a temperature sensing neuron AFD regulates increased resistance to hydrogen peroxide in worms cultivated at a higher temperature. They show that loss of AFD and the insulin-like peptide INS-39 produced by AFD increase H2O2 resistance similarly as high temperature growth. To understand the molecular basis, they use mRNA-seq and analysis of gene expression at the whole-genome scale and transgenic lines to show that AFD ablation and high cultivation temperature generate overlapping changes in gene expression via the function of the FOXO transcription factor DAF-16 in the intestine.This study is built on their previous work that established *C. elegans* as a model to study mechanisms for sensing and resistance of H2O2, an important environmental chemical threat for living organisms. Here, the authors uncover the neuronal and molecular basis for H2O2 resistance induced by high cultivation temperature. The authors use multiple approaches, including genetics, transgenics, whole-genome gene expression analysis, to characterize "enhancer sensing" that they discovered in this study. The experiments are well designed with appropriate controls. The data analysis is comprehensive and revealing. The findings are novel and explain a common and interesting phenomenon. The new understanding generated in this study will appeal to the readers in the fields of sensory biology, signaling transduction and physiology. The implications or conclusions of a few results presented here could be further discussed or clarified in the context of several previous studies.1. My main question is about the link between AFD's response to higher temperature, its activity, and ins-39 expression. Previous studies show that increasing temperature from 20 to 25 degree activates AFD measured by intracellular calcium imaging. These results together with the findings in this paper would suggest that increased AFD activity reduces ins-39 expression. It will be helpful for the authors to discuss about these implications more clearly. In this paper, the authors seem to suggest that higher temperature at 25 inhibits AFD to reduce ins-39 expression. This may lead to the prediction that in the tax-4 mutant, in which AFD is not active in response to temperature, ins-39 expression is higher than wild type. This is different from the results in Figure 4E. It is possible that the effect of growing at 25 degree on AFD is different from acute sensing of the higher temperature. It will be much helpful for authors to discuss and clarify these points clearly.

We thank the reviewer for this insightful comment. We have made changes to the Results section and added a new paragraph to the discussion to address the origin of the TAX-4 dependence of *ins-39* gene expression. The rewritten text at the end of the paragraph about *ins-*39 gene expression in the results now reads: “Taken together, these findings suggested that the AFD neurons lowered the expression of the INS-39 insulin/IGF1 hormone in response to the cultivation temperature via a TAX-4-dependent process.”

The new discussion paragraph reads:

“What mechanisms regulate *ins-39* gene expression in the AFD neurons in response to cultivation temperature? On a short timescale of seconds to minutes, the AFD neurons respond to changes in temperature by transiently increasing intracellular [Ca^+2^] and changing thermoreceptor currents through a process dependent on TAX-4 cyclic GMP-gated channels (Kimura *et al.*, 2004; Ramot *et al.*, 2008). On a longer timescale of hours, changes in temperature can modulate gene expression within AFD through a process mediated in part by intracellular [Ca^+2^] via the calcium/calmodulin-dependent protein kinase CMK-1 (Ippolito et al., 2021; Yu et al., 2014). Interestingly, the baseline intracellular [Ca^+2^] in AFD was lower in nematodes grown continuously at 25°C than in those at 15°C, although levels at 20°C were not assessed in that work (Ippolito *et al.*, 2021). Given that *tax-4* is essential for *ins-39* gene expression at both 20°C and 25°C, it will be interesting to determine how cultivation temperature and TAX-4 act to regulate *ins-39* gene expression in AFD on different timescales.”

2. Figure 1. Is H2O2 similarly stable at 20 degree and 25 degree?

H2O2 is predicted to be more reactive at 25C than at 20C, as quantified by the mathematical modeling shown in the new Figure 1. We interpret this question to refer to whether this compound is stable in our nematode survival assays. We have not determined empirically whether the rate of H2O2 degradation at any temperature is sufficiently high to make a difference during the assays we used. However, our assays were designed to minimize H2O2 degradation and maximize consistency. Specifically, in all our assays with H2O2 in this paper, the nematodes were (a) fed *E. coli* JI377, a *katG katE ahpCF* triple null mutant strain which cannot degrade environmental H2O2 (unlike the parental strain MG1655 and the commonly used *E. coli* strain OP50, which rapidly degrade H2O2), and (b) were exposed to the same batch of environments with H2O2 (survival plates with H2O2 or supernatants from *E. faecium* liquid culture). Despite the concern that H2O2 may degrade more rapidly at 25C than at 20C, in the assay in Figure 2B, a lower proportion of the worms grown and assayed at 25C survived exposure to the supernatant from *E. faecium* liquid culture than of worms grown and assayed at 20C. All the other assays with H2O2 were conducted at the same temperature, therefore, at each time point the worms were likely exposed to the same concentration of H2O2.

3. Fig2H shows that AFD ablation has a stronger effect in increasing resistance than growing at 25c -- the additional effect caused by AFD ablation could use some discussion.

We do not know what gave rise to these 21% differences in peroxide resistance. One possibility is that this small peroxide-resistance differences are due to differences in development: worms grown at 20C develop more slowly than those at 25C, and perhaps the additional developmental time enables AFD ablated worms to induce higher defense levels during adulthood at 20C than at 25C, leading to the increased survival of the worms grown at 20C.

4. The rationale for the authors to focus on ins-39 needs to be better clarified, since multiple INSs are found in AFD.

We have rewritten the paragraph to better explain the rationale for selecting *ins-39* in our candidate gene approach. Out of the many insulin/IGF1-like genes expressed in AFD, *ins-39* was the most highly expressed in AFD according to CeNGEN, and it was the insulin/IGF1-like gene with the highest expression in AFD relative to all neurons combined. The paragraph now reads:

“To investigate how the AFD sensory neurons regulated the nematode’s peroxide resistance, we took a candidate gene approach. We speculated that the AFD neurons signaled to target tissues via insulin/IGF1 peptide hormones because previous studies, including our own, showed that insulin/IGF1 signaling is a major determinant of peroxide resistance in *C. elegans* (Schiffer *et al.*, 2020; Tullet et al., 2008). A recent single-neuron mRNA-seq study by the *C. elegans* Neuronal Gene Expression Map and Network consortium (CeNGEN) showed that AFD expresses many classes of peptide-hormone coding genes, including a subset of the 40 insulin/IGF1 genes in the genome: *ins^-1^4*, *ins^-1^5*, *ins^-1^6*, *ins-39*, and *daf-28* (Taylor et al., 2021). We focused on the *ins-39* gene, which was highly expressed in AFD (Q. Ch'ng and J. Alcedo, personal communication) and was the only insulin/IGF1 gene with higher expression in AFD than in other neurons (Taylor *et al.*, 2021).”

5. Figure 4F. When growing at 25c, no further increase in resistance is seen in the ins-39 mutant compared to wild type, indicating a full effect; but at 20c, the ins-39 mutant does not fully mimic the resistance in worms grown at 25c. How would the authors explain this partial effect?

We thank the reviewer for noting this important point, which we had not explained. In the revised manuscript we explicitly address that point in this statement in the Results section:

“The increase in peroxide resistance at 20°C caused by the *ins-39* null mutation was smaller than those caused by growth at 25°C in wild-type nematodes or by AFD ablation at 20°C. Therefore, in addition to INS-39, other AFD-derived signals likely regulated the induction of peroxide defenses in target tissues in response to growth at 25°C.”

6. Figure 4G. CenGEN also identified the expression of ins-39 in several other neurons including ASK and ASJ. Therefore, the AFD-specific function of ins-39 in regulating H2O2 resistance should be further clarified.

We thank the reviewer for noting that *ins-39* was expressed also in ASK and ASJ neurons in the CeNGEN dataset. Our studies using the SL2-based *ins-39* transcriptional reporter (made via gene-editing of the *ins-39* endogenous locus) showed that *ins-39* was expressed exclusively in AFD. It is plausible that differences in growth conditions could account for these differences in *ins-39* expression in ASK and ASJ.

As described in our response to Reviewer #1’s point 3, we determined whether *ins-39* gene expression in AFD was sufficient to lower peroxide resistance by restoring *ins-39(+)* gene expression only in the AFD neurons using the AFD-specific *gcy-8* promoter. The findings from that experiment lead us to conclude that expression of *ins-39* in the AFD neurons was sufficient to regulate the nematode’s peroxide resistance.

7. Figure 3G. The authors suggest that increased ctl-1 and ctl-2 expression in AFD(-) worms confers increased resistance, because overexpression of these genes increases H2O2 resistance as shown in previous studies. This analysis of ctl genes provide evidence for the mechanistic basis for increased H2O2 resistance in AFD(-) worms. Is the increased expression of ctl-1 and ctl-2 in AFD(-) worms comparable to those generated by overexpression (ie does it confer higher resistance)? Some level of support or clarification will be very helpful.

The reviewer raises an interesting point about how the levels of catalase gene expression relate to the H2O2 resistance of the nematodes. The catalase overexpressing strain has 10-fold higher catalase activity, and increases H2O2 resistance 2.7 fold. To develop a better intuition of how consequential the ~45-90% changes in ctl-gene expression we observed are likely to be, we now compare how those increases in catalase-gene expression in AFD ablated worms and in wild-type worms at 25C, relative to wild type worms at 20C, compare with those we observed, in a previous study, in H2O2-resistant TGFbeta pathway mutants. The revised paragraph now reads:

“The *C. elegans* genome contains three catalase genes in tandem—two newly duplicated cytosolic catalases, *ctl-1* and *ctl-3*, and a peroxisomal catalase, *ctl-2* (Petriv and Rachubinski, 2004)—that when overexpressed 10-fold increase *C. elegans* resistance to hydrogen peroxide 2.7-fold (Schiffer *et al.*, 2020). *ctl-1* and *ctl-2* can increase *C. elegans* resistance to H2O2-dependent killing (Chavez *et al.*, 2007; Schiffer *et al.*, 2020). Previously, we found that *ctl-1* mRNA levels were 69% higher in *daf-1* Type 1 TGFβ receptor loss-of-function mutants, and that *ctl-1* function was required for a large part of the more than doubling of H2O2 resistance induced by those mutants (Schiffer *et al.*, 2020). In our mRNA-seq analysis, wild-type nematodes grown at 25°C had 46% higher levels of *ctl-1* expression and 73% higher levels of *ctl-2* expression compared to nematodes grown at 20°C (Figure 4G), and ablation of the AFD neurons increased *ctl-1* expression by 46% and increased *ctl-2* expression by 89% (Figure 4G). Therefore, the cultivation temperature and the AFD neurons regulated the expression of hydrogen peroxide defenses.”

8. Figure 7. It is much easier to read if most of the analysis in FigS8 is included in Fig7.

To address this issue, we moved the three panels quantifying the expression of DAF-16 targets and *daf-16*-regulated genes from the supplement to the main figure. We also split the remaining panels in the supplement into two figure supplements, to make them easier to access.

Reviewer #3 (Recommendations for the authors):This paper offers novel mechanistic insights into how pre-exposure to warm temperature increases the resistance of *C. elegans* to peroxides, which are more toxic at warmer temperature. The temperature range tested in this study lies within the animal's living conditions and is much lower than that of heat shock. Therefore, this study expands our understanding of how past thermosensory experience shapes physiological fitness under chemical stress. The paper is technically sound with most experiments or analyses carried out rigorously, and therefore the conclusions are solid. However, it challenges our current understanding of the role of the *C. elegans* thermosensory system in coping with stress. The traditional view is that the AFD thermosensory neuron is activated upon sensing temperature rise, and that temperature sensation through AFD positively regulates systemic heat shock response and promotes longevity in *C. elegans*. Thus, it is quite unexpected that AFD ablation activates DAF-16 and improves peroxide resistance. It also appears counterintuitive that genes upregulated at 25 degrees overlap extensively with those upregulated by AFD ablation at 20 degrees. I feel that it is premature to coin the term "enhancer sensing" for such a phenomenon, as their work does not rule out the possibility that AFD ablation increases resistance to other stresses that are independent of temperature regarding their toxicity or magnitude of hazard. Additional work is necessary to clarify these issues.1. Whether the role of AFD in inhibiting peroxide resistance is related to AFD activity needs further clarification. AFD activity depends on the animal's thermosensory experience. As animals in this study are maintained at 20 degrees unless indicated specifically, the AFD displays activities starting around 17 degrees and peaks around 20 degrees. Under such condition, the AFD displays little or no activity to thermal stimuli around 15 degrees. It will be important to test whether cultivation of animals at 20 degrees improves peroxide resistance at 15 degrees, compared to 15 degrees-cultivation/15 degrees peroxide testing. The authors should also test whether AFD ablation further improves survival under peroxides at 15 degrees for animals grown at 20 degrees, whose AFD should show little or no activities at 15 degrees.

The reviewer raises an interesting point about the relation between the mechanisms that determine AFD activity in response to temperature and those that enable AFD to regulate peroxide resistance. In the revised manuscript we tested whether known mechanisms enabling AFD to sense changes in temperature acutely (receptor guanylate cyclases GCY-8, GCY-18, and GCY-23) played a role in the temperature dependence of peroxide resistance. We found that they did, as detailed in our response to reviewer #1’s point 1.

As noted by reviewer #2 in their point 1, and in our reply to that comment (and in a new discussion paragraph in the revised manuscript), the relationship between the known mechanisms the acutely regulate the activity of AFD in response to temperature and the mechanisms by which constant cultivation temperature regulates gene expression in AFD (and therefore the expression of peroxide resistance regulating signals like INS-39) is not well understood. Therefore, it is difficult to predict which temperatures will cause induction of peroxide defenses via AFD-dependent mechanisms, or via other mechanisms. While we agree with the reviewer that it will be interesting to characterize the extent to which other cultivation temperatures besides 25C lead to increased peroxide resistance at lower temperatures (including the proposed shifts from 20C to 15C), we think that those questions will be better addressed in future studies.

2. The importance of the thermosensory function of AFD should be verified. In the current study, the tax-4 mutation was used to infer AFD activity, but tax-4 is expressed in sensory neurons other than AFD. In addition to AFD, AWC can sense temperature and it also expresses tax-4. Therefore, influence on AFD from other tax-4-expressing neurons cannot be excluded. On the other hand, ablation of AFD removes all AFD functions, including those that are constitutive and temperature-independent. Therefore, the authors should test the gcy-18 gcy-8 gcy-23 triple mutant, in which the AFD neurons are fully differentiated but completely insensitive to thermal stimuli. These three thermosensor genes are exclusively expressed in AFD. Compared to the tax-4 mutant that is broadly defective in multiple sensory modalities, this triple gcy mutant shows defects specifically in thermosensation. They should see whether results obtained from the AFD ablated animals could be reproduced by experiments using the gcy-18 gcy-8 gcy-23 triple mutant. The authors are also recommended to investigate ins-39 expression in AFD and profile gene expression patterns in the gcy-18 gcy-8 gcy-23 triple mutant.

We thank the reviewer for this suggestion. We have performed the requested experiments, as detailed in our response to reviewer #1’s point 1. Briefly, we determined found that *gcy-18 gcy-8 gcy-23* triple mutants increased peroxide resistance at 20C but not at 25C, and found that the respective gcy single mutants affected peroxide resistance at 20C. In light of these findings, we concluded that temperature perception by AFD via GCY-8, GCY-18, and GCY-23 enables *C. elegans* to lower their peroxide defenses at the lower cultivation temperature.

3. The literature suggests that AFD promotes longevity likely in part through daf-16 (Chen at al., 2016) or independent of daf-16 (Lee and Kenyon, 2009). Whatever it is, various studies show that activation of AFD and daf-16 promote a normal lifespan at higher temperature, and AFD ablation shortens lifespan at either 20 or 25 degrees. Therefore, the finding that DAF-16-upregulated genes overlap extensively with those upregulated by AFD ablation is quite unexpected (Figure 5B). The authors should perform further gene ontology (GO) analysis to identify subsets of genes co-regulated by DAF-16 and AFD ablation, whether these genes are reported to be involved in longevity regulation, immunity, stress response, etc.

We thank the reviewer for this interesting comment about the complex mechanisms by which AFD regulates longevity. We note that AFD also has additional temperature-dependent roles in lifespan regulation, as Murphy et al. 2003 found that RNAi of *gcy-18* increased lifespan in wild-type worms at 20C but not at 25C. Therefore, AFD-specific interventions can also be lifespan extending at 20C.

We performed WormCat analysis, which is similar to gene ontology, in Figure 8—figure supplement 2 (formerly Figure S8G), which we described in the Results section: “we found that the extent to which AFD ablation affected the average expression of sets of genes with related functions (Higgins *et al.*, 2022; Holdorf *et al.*, 2020) was systematically lower in *daf-16(mu86)* mutants than in *daf-16(+)* nematodes (*R*^2^ = 86%, slope = 0.67, *P* < 0.0001, Figure 8—figure supplement 2).” Visual inspection of the plot and the very high coefficient of determination of 86% indicate that the size of the effect of AFD ablation on gene expression was systematically smaller when the contribution of DAF-16 to gene expression was removed.

In the revised manuscript we also moved the three panels quantifying the expression of DAF-16 targets and *daf-16*-regulated genes from the supplement to the main figure. One of those panels (Figure 8F) shows that genes upregulated by *daf-16(+)* in *daf-2* mutants were disproportionally affected by lack of *daf-16* in AFD-ablated worms, as we described in the Results section: “In addition, in AFD ablated nematodes, lack of *daf-16* lowered the expression of genes upregulated in a *daf-16*-dependent manner in *daf-2(-)* mutants (Murphy et al., 2003) to a greater degree than in unablated nematodes (Figure 8F).”

4. I feel that "enhancer sensing" is an overstatement, or at least a premature term that is not sufficiently supported without further investigations. The authors should explore whether AFD ablation or pre-exposure to warm temperature specifically enhances resistance to a stressor the toxicity of which is increased at higher temperature, but does not affect the resistance to other temperature-insensitive threats.

We edited the manuscript and expanded the manuscript’s discussion to address these concerns as well as similar concerns from reviewer #1. For clarity, we repeat much of our response to reviewer #1’s point 2 here, with the last paragraph of this response specific to this reviewer’s comment.

In the paper we show that in *C. elegans* the regulation of the induction of H2O2 defenses is coupled to the perception of temperature (an inherent enhancer of the reactivity of H2O2). To understand the significance of this finding in an evolutionary context, and to explain why such a regulatory system would evolve, we introduced in the discussion a new conceptual framework, “enhancer sensing,” and devoted a section of the discussion to demonstrating that the phenomenon that we observed could not be adequately explained by existing frameworks used to understand the evolutionary origins of the regulatory systems for defense responses.

We now realize that we did not sufficiently and clearly explain the scope for the criterion for establishing a phenomenon represents enhancer sensing, leading to incorrect predictions by reviewer’s 1 and 3 about (a) whether what we observed in *C. elegans* is an instance of enhancer sensing (or more proof is needed) and (b) what the enhancer sensing model for the coupling of temperature perception to H2O2 defense would predict about how temperature and the AFD neurons would affect resilience to other chemicals. We regret failing to adequately explain the model’s scope and predictions and believe that we have now explicitly addressed the scope of what constitutes enhancer sensing and the predictions of the model. In particular, we previously did not spell out (a) the distinction between the enhancer sensing strategy and the mechanistic implementation of that strategy; and, importantly, (b) we did not discuss what the enhancer sensing strategy coupling temperature perception to H2O2 defense in *C. elegans* predicted (and did not predict) about whether a similar strategy would be expected to be used by *C. elegans* to deal with other temperature-dependent threats. We now address these issues in two new paragraphs in the discussion that read:

“We show here that *C. elegans* uses an enhancer sensing strategy that couples H2O2 defense to the perception of high temperature. We expect this strategy’s output (the level of H2O2 defense) to provide the nematodes with an evolutionarily optimal strategy across ecologically relevant inputs (cultivation temperatures) (Kussell and Leibler, 2005; Maynard Smith, 1982; Wolf et al., 2005). This strategy is implemented at the organismic level through the division of labor between the AFD neurons, which sense and broadcast temperature information, and the intestine, which responds to that information by providing H2O2 defense (Figure 9D). Ascertaining that *C. elegans* relies on this enhancer sensing strategy does not depend on the temperature information broadcast by AFD exclusively regulating defense responses to temperature-dependent threats, because the regulation of defense towards temperature-insensitive threats could affect defenses towards temperature-dependent threats; for example, suppressing defenses towards a temperature-insensitive threat would be beneficial if those defenses interfered with H2O2 defense or depleted energy resources contributing to H2O2 defense.

As with any sensing strategy, enhancer sensing strategies are more likely to evolve when sensing is informative and responding is beneficial. In their natural habitat, *C. elegans* encounter many environmental chemicals that, like H2O2, are inherently more reactive at higher temperatures. It will be interesting to determine the extent to which *C. elegans* uses enhancer sensing strategies coupling temperature perception to the induction of defenses towards those chemicals, and whether those strategies rely on temperature perception and broadcasting by the AFD neurons. We expect that sensing strategies regulating defense towards those chemicals would be more likely to evolve when those chemicals are common, reactive, and cause consequential damage.”

We note, in the first of the new discussion paragraphs, that the existence of an enhancer sensing strategy is not contingent on whether the AFD neurons (that implement the temperature sensing and temperature-information broadcasting functions regulating peroxide defenses) also do not regulate defense responses to temperature-insensitive threats. For example, it may be beneficial to an animal facing high concentrations of environmental peroxides to suppress defense against a temperature-insensitive threat when those defenses are detrimental towards defense towards hydrogen peroxide. This could occur, for example, because there is an energetic trade off when mounting multiple defense responses, or because specific defenses towards temperature-insensitive threats interfere with peroxide defense. As we noted in our response to reviewer #1’s point 2, our ability to predict survival to threats other than H2O2 (including temperature-independent threats) is limited not only by our lack of knowledge about the specific mechanisms that protect worms from those threats, but also by our inability to predict the extent to which defenses towards different threats operate independently, constructively, or destructively with those that provide hydrogen peroxide defense. We therefore think that those experiments would be better addressed in future studies.

5. The authors need to provide data of all survival assays as supplemental tables, and indicate the sets of data that are selected to be presented in the figures. This is standard practice for *C. elegans* lifespan or survival assays and is important for the readers to understand the reproducibility of the experiments. This applies to Figures 1A, 1C, 1D, 2B-E, 2H, 4F, 5C-F, 6B, 6C, S1A, S1B, S2, S6B and S7C.

We have updated the relevant tables in response to this request. As detailed in the methods section, “a typical experiment consisted of up to four genotypes or conditions, with 4 assay plates of each genotype or condition, each assay plate containing a maximum of 40 nematodes, and 16 assay plates housed in the same scanner.” The updated tables now include the average survival of the nematodes in each of the assay plates.

6. Page 17, Line 437-438: Loss of daf-16 in AFD-intact animals results in changes in the expression of merely two genes, which is quite surprising given the critical role of daf-16 in lifespan and stress resistance. Although the authors explained this in Discussion saying that fold changes in most genes were mild in the daf-16(mu86) mutant, it might be helpful to point out that in previous studies, many daf-16-dependent genes were characterized by comparing gene expressions in the daf-2 and daf-16; daf-2 double mutants. Given the critical importance of daf-16 in fitness, stress resistance and longevity, this apparent discrepancy with the existing literature must be explained in sufficient detail.

We agree with the reviewer that these findings could be surprising. We have updated that paragraph with two statements to (a) make explicit that our finds are in contrast with those previous transcriptomic studies in *daf-2(-)* mutants, and (b) discuss whether the effects of the *daf-16* mutant on peroxide resistance at 20C could also be due to effects on transcription after the nematodes are transferred to peroxide. The updated paragraph now reads:

“We found that lack of *daf-16* gene function in unablated nematodes at 20°C significantly lowered the expression of just 2 genes and significantly increased the expression of none, out of 7,387 genes detected (q value < 0.01) (Figure 8B). Previous transcriptomic studies in *daf-2(-)* mutants—unlike our study, which was conducted in *daf-2(+)* nematodes—have identified thousands of genes whose expression was regulated by *daf-16* (Kumar *et al.*, 2015; Lin et al., 2018; Murphy *et al.*, 2003). We found that lack of *daf-16* gene function in AFD-unablated [*daf-2(+)*] nematodes lowered the expression of genes directly upregulated by DAF-16 (Kumar *et al.*, 2015) (Figure 8D) and lowered the expression of genes upregulated in a *daf-16-*dependent manner in *daf-2(-)* mutants (Murphy *et al.*, 2003) (Figure 8E). However, these effects were small, averaging to just a 10% decrease in expression. These small *daf-16-*dependent effects on gene expression suggest that DAF-16 function was almost fully repressed by DAF-2 at 20°C. While these transcriptional effects are small, they could nevertheless contribute to the peroxide resistance of wild-type nematodes at 20°C. DAF-16 may also play a larger role in gene expression in unablated nematodes after peroxide exposure.”

7. DAF-16 and SKN-1 nuclear localization should be investigated at warm temperature, in the absence of AFD and in the ins-39 mutant.

We agree with the reviewer that it would be interesting to determine how cultivation temperature and AFD ablation affect the function of the DAF-16 and SKN-1 transcription factors. We mentioned in the manuscript that a previous study showed that DAF-16 nuclear localization increased at 25C relative to 20C (Wolf et al., 2008). Those studies examined intestinal DAF-16 location, which we now say in the manuscript. We note that the activity of *daf-16* and *skn-1* could be regulated by temperature and by the AFD neurons at multiple levels in addition to nuclear location; therefore, we think that those studies are beyond the scope of this paper.

8. Figure 3G: The fold change in the expression levels of ctl-1, ctl-2 and ctl-3 at 25 degrees (v.s. 20 degrees) and in AFD-ablated animals should be confirmed by RT-qPCR. The data from RNA-seq in their current form do not allow for statistical analysis to determine whether the expression of these ctl genes are significantly upregulated by high cultivation temperature or AFD ablation.

We apologize for not including the statistical analysis. We have updated the figure to include the statistical analysis demonstrating that the expression of *ctl-1* and *ctl-2* (but not *ctl-*3) was increased at 25C and in AFD-ablated worms. RNA-seq is a fairly unbiased approach to assess transcript quantity across the whole genome and doesn’t suffer from probe bias that can be experienced with qPCR primers, especially in duplicated and rearranged regions of the genome like the one that includes the three catalase genes that contain multiple regions of perfect and nearly perfect sequence identity. In a previous study we used both RNAseq and qPCR to examine *ctl-1* mRNA levels in wild type and *daf-7* mutants (which cause a similar induction in *ctl-1* gene expression as AFD ablation and growth at 25C), with identical results (Schiffer et al., 2020), so it is unlikely that qPCR would not confirm the RNAseq findings from this study.

9. It would be interesting to characterize gene expression changes in the ins-39 mutant and make comparison to those induced by temperature shift and AFD ablation. However, I understand that this is a substantial amount of work and it is OK if the authors decide not to do it at this moment.

Although we agree that this is an interesting future direction, we think those studies are beyond the scope of this study.

10. Please provide statistics in all the Figure panels instead of using designations such as "a, b, c".

We comparing multiple groups to one another, and correcting for multiple comparisons, we used the “a, b, …” labels to mark sets of conditions that are not significantly different from each other (same letter) and conditions that are significantly different from each other (different letter). This is a common statistical practice that, in our opinion, helps the reader identify which sets of groups are different and which ones are not, especially when there are more than two such sets, as in Figures 2B,E, 3F, and 5E. The figure captions note the level of statistical significance that those letters represent. One figure panel that was not providing sample sizes was updated (Figure 5D,E). The updated (and representative) figure legend now reads: “Groups labeled with different letters exhibited significant differences (n ≥ 10 in each group, *P* < 0.0001, Tukey HSD test) otherwise (*P* > 0.05).

11. For esthetical reasons, the authors should remove the black portions in Figure 4B that were used to rectify the borders of the rotated fluorescent images. However, I think there is still room for careful cropping of the fluorescent images to make them rectangular without having to add another layer of black background.

As requested, we updated the figure panel to make clear what is the boundary of the picture, using gray boxes behind the pictures instead of black boxes. We agree with the reviewer that this is a better practice.

12. Page 13, Line 327: please describe the molecular lesion of ins-39(tm6467).

We updated the text to describe the *ins-39* mutant allele and updated Figure 5A to show the location of the *tm6467* deletion. The updated paragraph now reads: “The *tm6467* null mutation in *ins-39* deletes 520 bases, removing almost all the *ins-39* coding sequence (Figure 5A), and inserts in that location 142-bases identical to an intervening sequence located between *ins-39* and its adjacent gene.”

13. Please indicate sample sizes for experiments in Figure 4C and 4D.

We had forgotten to include the sample sizes in the quantification of *ins-39* gene expression in Figure 5D,E, which we now include.

14. Remove the "A" label for Figure S2.

We removed the “A” label, which was unnecessary.

[Editors’ note: further revisions were suggested prior to acceptance, as described below.]

All of the reviewers feel that your manuscript has been substantially improved. However, in their discussion, the reviewers continued to express concerns about the validity of the "enhancer sensing" model. Before publication, we ask that you revise the Abstract and Discussion to present a more balanced view. Any other changes that you may wish to make in response to the comments below are optional.

As requested, we revised the abstract and discussion to address the concern about the need for coining the “enhancer sensing” term, and to present a more balanced view that explicitly acknowledges the limitations of our study. In the revised manuscript:

– We removed all mentions of enhancer sensing from the Abstract and Introduction.

– We put all discussion about enhancer sensing in two “Ideas and speculation” subsections within the Discussion, in line with *eLife*’s policy of encouraging authors to speculate about the implications of data and results (https://elifesciences.org/inside-eLife/e3e52a93/eLife-latest-including-ideas-and-speculation-in-eLife-papers).

– In the first of these subsections, we consider the specificity of the nematode’s strategy and identify the strategy’s unique features from a chemical-kinetic perspective. Most importantly, we changed the objective of the section from understanding “why the strategy evolved” to understanding “what the strategy accomplishes.”

– In the second of these subsections, we discuss limitations of our study and current knowledge, and propose future directions.

– We made several changes to explicitly spell out why adaptive prediction provides an insufficient explanation of the nematode’s strategy and why introducing the enhancer sensing concept is warranted, in the first “Ideas and speculation” subsection of the Discussion.

– We rephrased several sentences to improve clarity, and to use a more consistent and precise language.

Reviewer #1 (Recommendations for the authors):In their revised manuscript, Servello et al., have nicely addressed most of the concerns I raised earlier. I find the manuscript to be significantly improved. Before the paper is ready for publication, I think it would be worthwhile for the authors to consider addressing the following issues by editing the text.

We thank the reviewer for these very nice comments and appreciate the favorable assessment of our revised manuscript.

– The demonstration that the gcy triple mutant has a peroxide-resistant phenotype is nice, but it would be useful for the authors to speculate about why the phenotype of this mutant is so much weaker than ADF ablation. The phenotypes of the single gcy mutants are also a little unexpected; this should be noted somewhere.

We thank the reviewer for raising these very interesting points, which we incorporated into the manuscript. We now state that the AFD-specific receptor guanylate cyclase genes are not fully redundant in the regulation of peroxide resistance. We also contrast the size of the peroxide resistance phenotypes of the AFD ablation and the *gcy-23 gcy-8 gcy-18* triple mutant. The updated Results paragraph now reads (additions in blue):

“Therefore, each of the three AFD-specific receptor guanylate cyclases regulates peroxide resistance, and their roles are not fully redundant. We conclude that temperature perception by AFD via GCY-8, GCY-18, and GCY-23 enables *C. elegans* to lower their peroxide resistance at the lower cultivation temperature. Other mechanisms within AFD likely contribute to the regulation of peroxide resistance, as AFD ablation caused a greater increase in peroxide resistance than the *gcy-23 gcy-8 gcy-18* triple mutant.”

– I continue to feel that the "enhancer sensing" model is pushed too hard. In particular, I find the argument in the Discussion, lines 657 to 688, to be unconvincing. Adaptive sensing provides a mechanism by which species can "learn" to associate one piece of information with another, allowing detection of a particular environmental signal to become predictively coupled to an apparently distinct response. Here, the authors have shown (quite nicely) that *C. elegans* has "learned" that an increase in temperature predicts an increased susceptibility to peroxide damage. To me, this fits quite well with the adaptive sensing paradigm.

We address the relationship between “enhancer sensing” and "adaptive sensing" below, in the next section (2b). Before addressing that issue, here we want to note that, as requested by Reviewers 1 and 3, we reorganized the Discussion section to present a more balanced discussion that does not push the enhancer sensing model too hard and explicitly discusses the potential limitations of our study. We now distinguish the first part of the discussion, which contains a more conventional enumeration of the immediate implications of our study, from the more speculative Discussion section where we introduce and develop the concept of enhancer sensing. The discussion about enhancer sensing is now in two “Ideas and speculation” subsections within the Discussion, in line with *eLife*’s policy of encouraging authors to speculate about the implications of data and results.

Most importantly, we have reframed the objective of the section presenting the enhancer sensing model. Previously, this section was framed in terms of answering “Why did *C. elegans* evolve a mechanism that couples the induction of hydrogen peroxide defenses to sensory perception of high cultivation temperature?” Focusing on “why did this thing evolve?” was an incorrect way to frame the discussion. Instead, we now focus on answering “What does *C. elegans* accomplish by coupling the induction of hydrogen peroxide defenses to sensory perception of high cultivation temperature?” We think that shifting the focus to “what does this thing do?” provides a more appropriate objective to this section of the discussion.

In the first of these “Ideas and speculation” subsections, we considered the specificity of the nematode’s strategy and identified the strategy’s unique features from a chemical-kinetic perspective. The first subsection title and the text explaining the objective that subsection reads:

“Ideas and speculation: faithful assessment of a threat by sensing the enhancer of that threat

What does *C. elegans* accomplish by coupling the induction of hydrogen peroxide defenses to sensory perception of high cultivation temperature? To address this question, in this section we consider the specificity of the nematode’s strategy for dealing with the threat of hydrogen peroxide and identify the strategy’s unique features from a chemical-kinetic perspective.

In the second of those subsections, we discuss limitations of our study and current knowledge, and propose future directions. The second subsection reads:

“Ideas and speculation: limitations and unanswered questions

Because the studies presented here are the first to identify an enhancer sensing strategy, we do not know the extent to which this type of strategy is common across organisms. However, many previous findings may be indicative of enhancer sensing. For example, in the case of the regulation of H2O2 defenses by temperature, previous studies have shown that H2O2 defenses are induced in response to high temperature in a wide variety of organisms, including bacteria (Engelmann et al., 1995; Mossialos et al., 2006), yeasts (Deegenaars and Watson, 1997; Mitchell *et al.*, 2009; Wieser et al., 1991), plants (Hu et al., 2021; Nishizawa et al., 2006; Panchuk et al., 2002), cnidarians (Dash and Phillips, 2012), and human HeLa cells (Pallepati and Averill-Bates, 2010).

In addition, we do not know the extent to which enhancer sensing strategies couple temperature perception to multiple defense responses within any organism. In their natural habitat, *C. elegans* nematodes encounter many chemicals that, like H2O2, are inherently more reactive at higher temperatures. However, it is difficult to predict whether enhancer sensing strategies coupling temperature perception to defense towards those chemicals would be likely to provide a high adaptive value because we do not know the extent to which those chemicals are common, abundant, and reactive enough to cause consequential damage within the temperature range that *C. elegans* experience in their ecological setting. Because growth at 25°C did not induce gene sets induced by acrylamide, formaldehyde, benzene, silver, cadmium, and arsenic, we expect that resistance to lethal concentrations of these toxic chemicals will not increase with pre-exposure to 25°C. However, it is likely that other temperature ranges sensed by the AFD neurons might regulate resistance to those chemicals, because AFD ablation at 20°C induced the gene sets induced by each of those chemicals. In the future, we plan to determine the extent to which *C. elegans* uses enhancer sensing strategies to couple the perception of specific temperature ranges to the induction of defenses towards these and other toxic chemicals, and whether those strategies rely on temperature perception and broadcasting by AFD and other temperature-sensing neurons. More broadly, it will be interesting to determine the extent to which enhancer sensing strategies are used throughout the tree of life to couple specific defense responses to the perception of inputs that enhance the need for those defenses.”

I'm ok with the idea that what they are seeing could be considered a variant form of adaptive sensing, but it seems to me to be excessive to decide that a new term is needed for what amounts to essentially the same principle.

We think that the distinction between these different stress response strategies is an important contribution of our manuscript. This distinction highlights the difference between strategies based on guessing and strategies based on knowing. To make this distinction more explicit and clearer, we rewrote the two paragraphs linking to the model in Figure 9 (and the caption for that Figure) to explicitly state the limitations of the “adaptive sensing” strategy (now referred as “adaptive prediction”, in line with the name coined by Mitchell et al., 2009). In the revised manuscript, we now explicitly name the key feature of the nematode’s strategy missing from the adaptive prediction strategy but present in the enhancer sensing strategy: the chemical constraint linking temperature and hydrogen peroxide reactivity. This chemical constraint distinguishes the adaptive prediction, a strategy based on guessing when to induce the nematode defenses, from strategies like enhancer sensing and classical stress responses, which are based on knowing when to do so. Making this useful distinction requires naming the new strategy to differentiate it from the other ones. The revised paragraph now reads:

“Adaptive prediction provides a plausible explanation for why *C. elegans* evolved a regulatory mechanism coupling temperature perception to H2O2 defense. However, in our opinion, that explanation is insufficient, because it does not incorporate a key feature of the nematode’s strategy: the chemical constraint linking temperature and hydrogen peroxide reactivity removes guesswork from the strategy. Contrary to the expectation from adaptive prediction, *C. elegans* nematodes are not guessing that in their ecological setting increasing temperature leads to a higher H2O2 threat; instead, in all ecological settings the nematodes’ proteins, nucleic acids, and lipids are inherently more likely to be damaged by H2O2 with increasing temperature because those chemical reactions necessarily run faster with increasing temperature (Arrhenius, 1889; Evans and Polanyi, 1935; Eyring, 1935). This chemical constraint means that by coupling the induction of H2O2 defenses to the perception of high temperature, the nematodes are not guessing; instead, they are assessing faithfully the threat that hydrogen peroxide poses. We refer to this distinct strategy as “enhancer sensing” (Figure 9).”

We also rephased the second of those two paragraphs to explain the differences between classical stress response, adaptive prediction, and enhancer strategies more clearly and with a more precise and consistent language. The rewritten paragraph now reads:

“Enhancer sensing provides a new framework for understanding the adaptive value of strategies coupling the induction of defense responses to the perception of inputs that inherently modulate the need for those defenses. In a classical stress response, the strategy provides faithful information about the threat the organism faces because the response that enables the organism to cope with the stress induced by an input is coupled to the perception of that input (Figure 9A). In enhancer sensing, an input’s capacity to induce a stress is modulated by another input; the strategy provides faithful information about the threat the organism faces because perception of either input provides information about the threat posed by the interaction of those inputs (Figure 9C). In contrast, in adaptive prediction, the sensed input does not induce (nor modulate the capacity of another input to induce) the stress that the organism attempts to cope with by inducing a response to that input; as a result, the strategy does not necessarily provide faithful information about the threat the organism faces; instead the strategy provides a guess whose predictive value matches the co-occurrence in the ecological setting of the organism of the input that induces the stress and the input that is sensed (Figure 9B).”

In a way, even that is overkill: because the hazard is the thermodynamic effect of temperature itself, one could argue that increasing peroxide defense is an aspect of temperature defense, in the same way that increasing expression of protein chaperones is. As the authors note, it's well demonstrated that heat detection by the *C. elegans* nervous system can trigger the systemic activation of the heat shock response. The activation of peroxide defenses seems to me to be, in principle, an equivalent phenomenon.

We agree that at the molecular level there are many similarities between the intercellular signaling mechanisms linking the perception of distinct temperature ranges by the AFD sensory neurons to the induction of distinct gene sets in target tissues. However, we think that it is worthwhile to make some distinctions about the separate role that temperature plays in inducing different types of stresses and different types of stress responses.

The stress induced by the unfolding of proteins at high (e.g. >30C) temperatures is specific to temperature and is not induced by (or modulated by) H2O2. The stress induced by the formation of oxidized proteins, lipids, and nucleic acids by the chemical reactions of H2O2 with macromolecules is specific to H2O2, and is not induced by temperature alone in the absence of H2O2. However, temperature does modulate the rates of the reactions of H2O2 in a manner predicted by chemical kinetics theory (as we show in Figure 1). Those rates are a function of both the concentration of H2O2 and temperature.

The strategies (Figure 9) that we discuss in the manuscript take account of the relationships between the input (or inputs) that induces a specific stress and the input whose perception regulates the induction of specific defenses that enable organisms to prevent, repair, or cope with that specific stress. In some cases, like in the heat shock response, one input (high temperature) causes protein-unfolding stress and the perception of that same input by the AFD neurons induces the expression of heat shock proteins that help the worm repair and cope with that protein-unfolding stress. This strategy maps to the classical stress response diagramed in Figure 9A. In contrast, the induction of intestinal catalases in response to temperature perception by AFD that we identified and characterized in our manuscript, does not map to the strategy diagramed in Figure 9A. Instead, it maps to the strategy we call enhancer sensing diagramed in Figure 9C because both inputs cause the stress (temperature and H2O2 jointly control the rates of the reactions that cause the peroxide stress) but the input (25C) sensed by AFD to induce catalase gene expression is different from the one that controls the specificity of the reactions that cause the stress. In contrast in guessing strategies (like adaptive prediction) separate inputs induce the stress and the stress response, as diagrammed in Figure 9B.

– The authors write in their rebuttal that it is incorrect to predict that the response to non-H2O2 stresses should not be enhanced by 25C pre-exposure. In principle, I agree – one can't necessarily predict that augmenting defenses against these stressors wouldn't also be adaptive at 25C – but isn't the point of lines 340-342 to indicate that pre-exposure to 25C does indeed have some selectivity in terms of the stress responses it enhances? If so, then there seems to be a strong prediction that animals' responses to acrylamide, formaldehyde, etc, would not be changed by 25C pre-exposure.

The reviewer makes an excellent point. In our original response we focused only on predictions of the enhancer sensing model. We apologize for answering in such a narrow sense. As noted by the reviewer, the selectivity of the effects of 25C on gene expression relative to 20C leads to as strong prediction (assuming that defense responses are specific and do not interfere with each other). We have incorporated the reviewer’s point in the “Ideas and speculation: limitations and unanswered questions” subsection of the discussion. The new text reads:

“In addition, we do not know the extent to which enhancer sensing strategies couple temperature perception to multiple defense responses within any organism. In their natural habitat, *C. elegans* nematodes encounter many chemicals that, like H2O2, are inherently more reactive at higher temperatures. However, it is difficult to predict whether enhancer sensing strategies coupling temperature perception to defense towards those chemicals would be likely to provide a high adaptive value because we do not know the extent to which those chemicals are common, abundant, and reactive enough to cause consequential damage within the temperature range that *C. elegans* experience in their ecological setting. Because growth at 25°C did not induce gene sets induced by acrylamide, formaldehyde, benzene, silver, cadmium, and arsenic, we expect that resistance to lethal concentrations of these toxic chemicals will not increase with pre-exposure to 25°C. However, it is likely that other temperature ranges sensed by the AFD neurons might regulate resistance to those chemicals, because AFD ablation at 20°C induced the gene sets induced by each of those chemicals. In the future, we plan to determine the extent to which *C. elegans* uses enhancer sensing strategies to couple the perception of specific temperature ranges to the induction of defenses towards these and other toxic chemicals, and whether those strategies rely on temperature perception and broadcasting by AFD and other temperature-sensing neurons.”

Reviewer #3 (Recommendations for the authors):The authors addressed most of the points that I raised in the initial review of the study. Although I am still not completely convinced by the idea of enhancer sensing and some of the interpretations of the role of AFD on gene expression and physiology, in particular its interaction with daf-16, I think those issues are better addressed in a separate study. Regarding "enhancer censing," for such a new term to be coined, multiple examples are necessary. Therefore, I suggest that the authors tone down this claim in the Discussion/Abstract, and also discuss the potential limitations of their study in justifying such a claim. In short, a more balanced view is needed.

We have made several changes to address these issues, including removing mentioning enhancer sensing in the Abstract and Introduction, and reorganizing the Discussion section to present a more balanced discussion that does not push the enhancer sensing model too hard and explicitly discusses the potential limitations of our study. We now distinguish the first part of the discussion, which contains a more conventional enumeration of the immediate implications of our study, from the more speculative Discussion section where we introduce and develop the concept of enhancer sensing. The discussion about enhancer sensing is now in two “Ideas and speculation” subsections within the Discussion: “Ideas and speculation: faithful assessment of a threat by sensing the enhancer of that threat” and “Ideas and speculation: limitations and unanswered questions”, in line with *eLife*’s policy of encouraging authors to speculate about the implications of data and results.

Most importantly, we have reframed the objective of the section presenting the enhancer sensing model. Previously, this section was framed in terms of answering “Why did *C. elegans* evolve a mechanism that couples the induction of hydrogen peroxide defenses to sensory perception of high cultivation temperature?” Focusing on “why did this thing evolve?” was an incorrect way to frame the discussion. Instead, we now focus on answering “What does *C. elegans* accomplish by coupling the induction of hydrogen peroxide defenses to sensory perception of high cultivation temperature?” We think that shifting the focus to “what does this thing do?” provides a more appropriate objective to this section of the discussion. Please refer to our response to Reviewer 1’s point 2a for a detailed description of these changes.

We think that coining the new term “enhancer sensing” is necessary in order to make the distinction between the different stress response strategies detailed in Figure 9. In our opinion, this is an important contribution to the paper. In the new “Ideas and speculation: limitations and unanswered questions” subsection, we begin the first and second paragraphs by acknowledging that we do not know how common this strategy is: “Because the studies presented here are the first to identify an enhancer sensing strategy, we do not know the extent to which this type of strategy is common across organisms” and ”In addition, we do not know the extent to which enhancer sensing strategies couple temperature perception to multiple defense responses within any organism.” Please refer to our response to Reviewer 1’s points 2b and 2c for a detailed explanation of our rationale for introducing the enhancer sensing concept and examples of how this concept can be useful.